# MITRA: Mixed Synthetic Priors for Enhancing Tabular Foundation Models

**Xiyuan Zhang**
Amazon

**Danielle C. Maddix**
Amazon

**Junming Yin**
Amazon

**Nick Erickson**
Amazon

**Abdul Fatir Ansari**
Amazon

**Boran Han**
Amazon

**Shuai Zhang**
Amazon

**Leman Akoglu**
Amazon and CMU

**Christos Faloutsos**
Amazon and CMU

**Michael W. Mahoney**
Amazon

**Cuixiong Hu**
Amazon

**Huzefa Rangwala**
Amazon

**George Karypis**
Amazon

**Bernie Wang**
Amazon

## Abstract

Since the seminal work of TabPFN [16], research on tabular foundation models (TFMs) based on in-context learning (ICL) has challenged long-standing paradigms in machine learning. Without seeing any real-world data, models pretrained on purely synthetic datasets generalize remarkably well across diverse datasets, often using only a moderate number of in-context examples. This shifts the focus in tabular machine learning from model architecture design to the design of synthetic datasets, or, more precisely, to the prior distributions that generate them. Yet the guiding principles for prior design remain poorly understood. This work marks the first attempt to address the gap. We systematically investigate and identify key properties of synthetic priors that allow pretrained TFMs to generalize well. Based on these insights, we introduce MITRA [1], a TFM trained on a curated mixture of synthetic priors selected for their diversity, distinctiveness, and performance on real-world tabular data. MITRA consistently outperforms state-of-the-art TFMs, such as TabPFNv2 [17] and TabICL [29], across both classification and regression benchmarks, with better sample efficiency.

## 1 Introduction

Tabular data lie at the core of many real-world applications, including healthcare, finance, e-commerce, and the sciences [32]. Predictive modeling on tabular data is central to statistical data analysis and underpins decision-making systems across these diverse domains [14]. Tree-based models, such as random forests, gradient boosting, and ensemble methods [3, 7], have long dominated tabular predictions, due to their strong empirical performance and ease of use. However, these methods are typically tailored to individual datasets, and they exhibit limited ability to transfer across different distributions. Despite advances in transfer learning, a truly general-purpose approach to tabular prediction has remained elusive, until the introduction of TabPFN [16].

Inspired by the success of large language models (LLMs), TabPFN and its follow-up works have introduced the notion of tabular foundation models (TFMs) based on in-context learning (ICL) [16, 17, 29, 2, 5]. These models are pretrained on synthetic tabular tasks, and they make predictions on

---

[1]We released both classifier (autogluon/mitra-classifier) and regressor (autogluon/mitra-regressor) model weights on HuggingFace.

39th Conference on Neural Information Processing Systems (NeurIPS 2025).

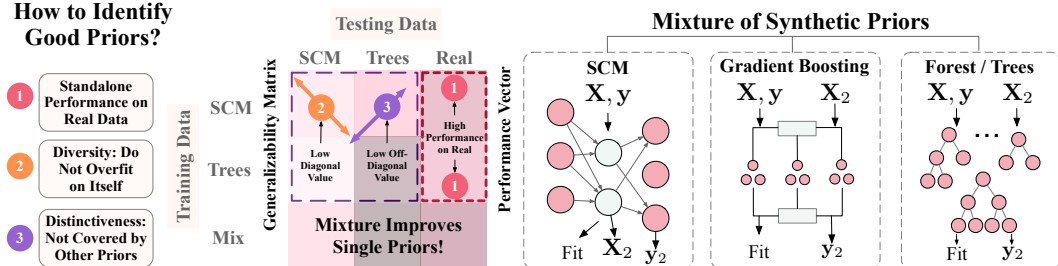

Figure 1: Mixture of priors in MITRA, Generalizability Matrix **G** and Performance Vector **P**. We consider three factors for good priors: (1) the performance of a pretrained TFM using data generated solely from that prior on real tabular data (Point 1 and **P**); (2) its diversity and (3) distinctiveness when included in a mixture (Points 2 and 3, captured by the diagonal and off-diagonal entries of **G**). This leads to our mixture comprising SCM, gradient boosting, random forest and decision tree priors.

real downstream tasks by conditioning on labeled training samples from the downstream tasks as in-context examples. Synthetic data, generated on-the-fly during pretraining, provides broad task coverage and enables adaptation, without the need for large amounts of real-world downstream data.

While most previous TFM efforts focus on architectural innovations [24, 9, 29], we advocate that greater attention should instead be on the design of the data priors—the distributions used to generate the synthetic datasets. Existing work typically relies on fixed or heuristic data priors [29, 16, 17, 2, 5], leaving fundamental questions open. For example: *What makes a synthetic data prior effective? How should one construct a mixture of priors for better generalization?*

In this paper, we investigate these questions, with the goal of identifying key properties of synthetic priors used for pretraining TFMs. Our findings sharpen a vague rule of thumb that "diversity of the prior is important." We show that the effectiveness of a synthetic prior depends on: (1) the performance of a TFM pretrained *solely on data generated from that prior*, when evaluated on real tabular data; (2) its *diversity*, i.e., how difficult it is for a TFM pretrained on this prior to overfit on its own distribution; and (3) *distinctiveness within a mixture of priors*, i.e., how hard it is for data generated from this prior to be predicted by TFMs pretrained on other priors. See Figure 1 for a simplified illustration.

Based on our insights, we construct a mixture of synthetic priors that consists of *structural causal models* (SCM) [16] and *tree-based priors* (TBP), including gradient boosting, random forest, decision tree, and extra tree models. We choose SCMs because we show that they are diverse and achieve the best standalone performance on real tabular datasets. We choose TBPs because we identify that TFMs pretrained with data from SCMs do not always generalize well to all types of data generated from TBPs, showing the distinctiveness property of TBPs.

Our mixture of priors enables effective coverage of the diverse distributions in real-world tabular data. Notably, our priors are *model-agnostic*, and they consistently demonstrate performance improvement for both row-based 1D attention [16, 2] and more advanced element-based 2D attention architectures [17, 5]. Building on the 2D attention architecture, we propose MITRA, a TFM pretrained with our mixture of priors that obtains state-of-the-art (SOTA) results. MITRA outperforms existing TFMs, e.g., TabPFNv2 [17], TabICL [29], and other strong baselines, on both classification and regression tasks. Additionally, MITRA models pretrained with our prior mixture demonstrate better sample efficiency, i.e., they consistently have stronger performance with fewer in-context examples. This highlights the benefits of our principled prior mixture analysis.

We summarize our major contributions as follows:

- **Characterize Key Factors for Good Priors.** We conduct the first principled analysis of how to combine various synthetic data priors for TFM pretraining, and we identify three key factors for good priors: performance, diversity and distinctiveness.
- **Construct Effective Mixture of Priors.** Following our analysis, we construct a novel and diverse mixture of synthetic priors that is effective and model-agnostic.
- **Build SOTA TFM with Prior Mixture.** We propose MITRA, a TFM pretrained using our mixture of priors. MITRA sets a new SOTA on both classification and regression tasks. MITRA outperforms strong baselines across three major benchmarks—TabRepo [30], TabZilla [25] and AMLB [11]— and it demonstrates better sample efficiency, given fewer in-context examples.

## 2 Related Work

**Traditional and Deep Learning-Based Tabular Models.** Historically, the tabular domain has been mainly dominated by traditional statistical methods, most notably gradient boosting (GB) decision trees, e.g., XGBoost [3], LightGBM [20], and CatBoost [28]. These methods are widely adopted in practice due to their strong performance, robustness, and interpretability. To further improve generalization and automation, ensemble-based systems (e.g., AutoGluon [7]) combine multiple base learners and automatically optimize hyperparameters and stacking strategies. More recently, deep learning-based approaches have been proposed to model complex and rich interactions in tabular data [31, 19, 13, 12]. For example, RealMLP [18] introduces an optimized multilayer perceptron (MLP) architecture, tuned over a broad set of meta-benchmark datasets. While both traditional statistical methods and neural networks remain competitive on many real-world benchmarks, they require retraining from scratch for each new dataset, and they struggle to generalize across different distributions. This need for repeated per-data *re*training poses scalability challenges and limits model reuse in real-world applications.

**TFMs: Semantically-Rich Models.** Recent work has explored adapting LLMs to structured data by serializing tables into text. TabLLM [15], GTL [34], and Tabula [10] enable zero-shot or few-shot inference by formatting rows and tasks into promptable inputs. TP-BERTa [36] propose a pretrained language model tailored for tabular prediction tasks, with relative magnitude tokenization and intra-feature attention mechanism. Beyond text serialization, several approaches focus on non-textual pretraining by leveraging cross-table techniques. These include independent featurizers [39], modular encodings with multi-task masked reconstruction [37], and prompt masked table modeling [38]. CARTE [21] leverages pretraining on knowledge graphs and column-level metadata to facilitate downstream tabular tasks. These methods often rely heavily on semantic metadata (e.g., column names or textual descriptions), curated schemas, or large-scale real corpora. This dependence limits their applicability in settings where this auxiliary information is unavailable or unreliable, and it may require expensive inference costs in practical deployments due to their reliance on LLMs.

**TFMs: ICL-Based Models.** A complementary line of work frames tabular prediction as an ICL task, where models are pretrained on synthetic or real datasets and downstream datasets are used as in-context examples. These efforts primarily focus on two directions: (1) designing better data priors; and (2) improving model architectures. In the first direction, TabPFN [16] introduces this paradigm by pretraining a Transformer [33] on SCM-generated synthetic data to simulate ICL for tabular classification problems. TabDPT [23] extends this approach by incorporating real datasets during pretraining. Several subsequent models—including TabForestPFN [2], Attic [5], and TabICL [29]—combine tree-based priors with SCMs for synthetic data generation. Specifically, TabForestPFN and Attic mix SCMs with decision trees, and TabICL combines SCMs with XGBoost. These approaches adopt tree-based priors in a heuristic manner, without explaining why incorporating these priors is beneficial. In the second direction, Attic [5] introduces an element-based attention mechanism, which treats each cell (rather than every row or column) in a table as a separate token in the Transformer. TabPFNv2 [17] adopts a similar element-wise Transformer architecture, and it further improves scalability and generalization by refining the synthetic prior distributions. More recently, TabICL [29] proposes a two-stage architecture to first build fixed-dimensional embeddings of rows, followed by an efficient attention mechanism for ICL. Recent efforts have also explored how to scale ICL through compressing prompts, selecting informative contexts [9, 35, 24] and hypernetworks [26, 1].

## 3 MITRA: TFM Pretrained on a Mixture of Priors

In this section, we describe how we characterize effective priors with the following three criteria: (1) strong performance on real datasets; (2) diversity; and (3) distinctiveness within the mixture. These characteristics together lead to the development of a new SOTA TFM, MITRA.

### 3.1 Data-Generating Priors

To pretrain a TFM on purely synthetic data in the supervised tabular learning setting, a data-generating prior $\mathcal{G}_i$ takes as input uniformly randomly generated hyper-parameters, e.g., feature size, number of samples, class count (for classification tasks), and categorical feature count; and it outputs a dataset $\mathcal{D}^{(i)} = \{(\mathbf{x}_n^{(i)}, y_n^{(i)})\}_{n=1}^{N_i}$. This dataset consists of $N_i$ feature-label pairs, where $\mathbf{x}_n^{(i)} \in \mathbb{R}^d$ denotes a $d$-dimensional feature vector with continuous or categorical attributes; and $y_n^{(i)} \in \mathcal{Y} \subset \mathbb{Z}$ denotes the corresponding target label for a classification task, or $y_n^{(i)} \in \mathcal{Y} \subset \mathbb{R}$ the target value for a regression

Table 1: Three factors of prior importance. Each entry $\mathbf{G}_{ij}$ in the Generalizability Matrix represents the AUC of a TFM pretained on data generated from $\mathcal{G}_i$ and evaluated on data from $\mathcal{G}_j$. Each element $\mathbf{P}_i$ in the Performance Vector corresponds to the AUC of a TFM pretained on data from $\mathcal{G}_i$ and evaluated on a real-world dataset. We use a color gradient to visually indicate the relative quality of each prior. The best/worst-performing priors are shown in the darkest green/red.

| Train | Test | | | | | | P | |
| | SCM | ET | GB | DT | RF | DSRF | TabRepo | |
|---|---|---|---|---|---|---|---|---|
| SCM | $0.830_{(0.201)}$ | $0.751_{(0.163)}$ | $0.892_{(0.102)}$ | $0.876_{(0.108)}$ | $0.708_{(0.140)}$ | $0.960_{(0.061)}$ | $0.857_{(0.142)}$ | |
| ET | $0.805_{(0.195)}$ | $0.871_{(0.115)}$ | $0.938_{(0.072)}$ | $0.948_{(0.059)}$ | $0.766_{(0.136)}$ | $0.975_{(0.049)}$ | $0.847_{(0.140)}$ | |
| GB | $0.805_{(0.201)}$ | $0.800_{(0.150)}$ | $0.971_{(0.034)}$ | $0.916_{(0.089)}$ | $0.745_{(0.143)}$ | $0.978_{(0.043)}$ | $0.843_{(0.147)}$ | |
| DT | $0.796_{(0.204)}$ | $0.866_{(0.120)}$ | $0.940_{(0.068)}$ | $0.951_{(0.055)}$ | $0.768_{(0.135)}$ | $0.974_{(0.050)}$ | $0.839_{(0.148)}$ | |
| RF | $0.804_{(0.198)}$ | $0.814_{(0.134)}$ | $0.917_{(0.080)}$ | $0.905_{(0.090)}$ | $0.761_{(0.129)}$ | $0.965_{(0.052)}$ | $0.831_{(0.147)}$ | |
| DSRF | $0.814_{(0.209)}$ | $0.839_{(0.133)}$ | $0.944_{(0.081)}$ | $0.937_{(0.069)}$ | $0.767_{(0.135)}$ | $0.984_{(0.034)}$ | $0.849_{(0.144)}$ | |

Generalizability Matrix **G** · **P**

Prior Importance — High / Low

task. The entire dataset can be represented as a two-dimensional matrix/table $D^{(i)} \in \mathbb{R}^{N_i \times (d+1)}$. (See Appendix A.1 for details on problem settings and preliminaries.)

In MITRA, we propose to use a mixture of $M$ data-generating priors $\{\mathcal{G}_i\}_{i=1}^{M}$. Concretely, we include the following types of priors: structural causal models (SCMs); and tree-based priors (TBPs).

**SCM.** The data-generating prior, $\mathcal{G}_{\text{SCM}}$, which was originally introduced in the pretraining of TabPFN [16], is capable of capturing causal relationships among columns observed in tabular data. Moreover, SCMs model both feature dependencies and the conditional distribution $p(y|\mathbf{x})$. To sample a dataset $\mathcal{D}$ from $\mathcal{G}_{\text{SCM}}$, a directed acyclic graph (DAG) is first randomly constructed, after which the features $\mathbf{x}$ and target $y$ are generated in a sequential manner, following the conditional dependencies defined by the DAG structure.

**TBP.** The data-generating tree-based priors, $\mathcal{G}_{\text{TBP}}$, consist of trees and ensembles of trees, which are known for their strong predictive performance on tabular tasks and which are commonly used to model decision boundaries in tabular data. TBPs primarily focus on modeling $p(y|\mathbf{x})$ using complex threshold-based splits, with ensemble trees helping to smooth the resulting axis-aligned decision boundaries. In this work, we consider $\mathcal{G}_{\text{DT}}$, $\mathcal{G}_{\text{ET}}$, $\mathcal{G}_{\text{GB}}$, and $\mathcal{G}_{\text{RF}}$, where DT, ET, GB, RF refer to decision tree, extra tree, gradient boosting, and random forest, respectively. To use these models as data generators, they are first fit on a synthetically generated training dataset, after which features $\mathbf{x}$ are sampled from a simple distribution (e.g., multivariate standard normal) and targets $y$ are drawn according to the learned conditional distribution $\hat{p}(y \mid \mathbf{x})$. (See Appendix A.2.1 for additional details on the data generation process.) We group these priors together, and we refer to them as "indirectly sampled" priors. In addition, we introduce a custom "directly sampled" RF prior, $\mathcal{G}_{\text{DSRF}}$, that does not require model fitting. Instead, it directly constructs random conditional distributions $p(y \mid \mathbf{x})$ by sampling random split indices and thresholds. (See Appendix A.2.2 for details on each of the priors.)

### 3.2 Prior Mixture Promoting Diversity and Distinctiveness

Here, we provide an in-depth study on characteristics of effective mixture $\mathcal{G}' = \{\mathcal{G}'_i\}_{i=1}^{M'}$ of $M'$ data-generating priors from $\mathcal{G} = \{\mathcal{G}_i\}_{i=1}^{M}$, where $M' \leq M$. We first pretrain $M$ models on data from each candidate prior, and we evaluate them across priors to construct a generalizability matrix $\mathbf{G} \in \mathbb{R}^{M \times M}$, where the rows of $\mathbf{G}$ denote the model pretrained on draws from $\mathcal{G}_i$, and the columns denote the test data generated on draws from $\mathcal{G}_j$, with $\mathbf{G}_{ij}$ denoting the metric value. We also evaluate these $M$ models on real-world datasets to form a performance vector $\mathbf{P} \in \mathbb{R}^{M}$, where $\mathbf{P}_i$ denotes the performance of a model pretrained on data from $\mathcal{G}_i$ and evaluated on these real datasets. We find three key factors when characterizing good priors: (1) **Performance**, quantified by a higher value of $\mathbf{P}_i$; (2) **Diversity**, quantified by a lower diagonal value $\mathbf{G}_{ii}$, which indicates greater difficulty in overfitting to the same prior; (3) **Distinctiveness**, quantified by a lower off-diagonal value $\mathbf{G}_{ij}$, which shows how well a model pretrained on $\mathcal{G}_i$ performs on data from $\mathcal{G}_j$. More specifically, given a current mixture $\mathcal{G}'$, the maximum of the off-diagonal $\mathbf{G}_{ij}$ in the $j^{\text{th}}$ column for $i$ such that $\mathcal{G}_i \in \mathcal{G}'$ is a measure of the distinctiveness of $\mathcal{G}_j$. Then, adding the prior $\mathcal{G}_j$ with the smallest maximum, i.e., $\min_{1 \leq j \leq M} \max_{1 \leq i \leq M, \mathcal{G}_i \in \mathcal{G}'} \mathbf{G}_{ij}$, increases the coverage of the mixture. Ultimately, prior importance balances performance and diversity.

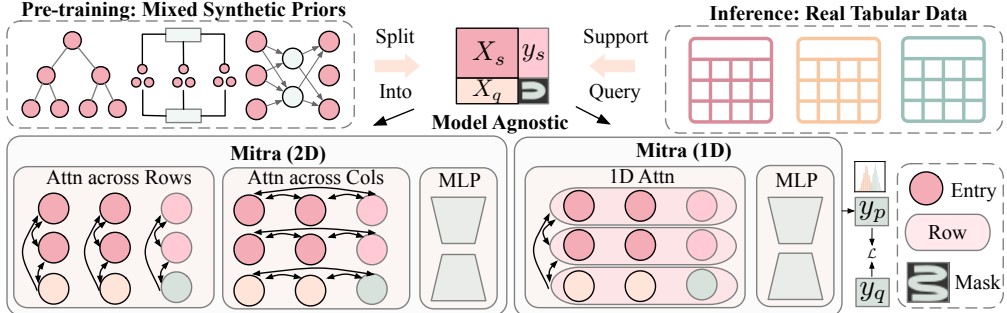

Figure 2: MITRA Pipeline: Model agnostic visualized with either 1D or 2D attention.

Table 1 provides an illustrative example using the AUC metric of how these three factors help explain the prior importance findings in our ablation study in Table 6 (below; see Appendix C.1 for similar findings in other aggregated metrics). We see that $\mathcal{G}_{\text{SCM}}$ is of high quality due to both diversity (low diagonal value $\mathbf{G}_{ii}$) and strong performance on real datasets $\mathbf{P}$. Interestingly, while $\mathcal{G}_{\text{DSRF}}$ ranks second in performance on $\mathbf{P}$, the third best performing prior $\mathcal{G}_{\text{ET}}$ shows higher quality based on its ability to increase the diversity and distinctiveness, as validated in Table 6. For diversity, the diagonal of $\mathcal{G}_{\text{DSRF}}$ is significantly higher than the diagonal of $\mathcal{G}_{\text{ET}}$ (0.984 vs. 0.871). For distinctiveness, the off-diagonals in the SCM row of $\mathbf{G}$ measure how much unique information is added from the next prior. We see that the model pretrained with $\mathcal{G}_{\text{SCM}}$ predicts on test data drawn from $\mathcal{G}_{\text{DSRF}}$ significantly better than that from $\mathcal{G}_{\text{ET}}$ (0.960 vs. 0.751), showing that $\mathcal{G}_{\text{ET}}$ has higher distinctiveness.

### 3.3 Pretraining TFM on Prior Mixture

We use the insights from the prior subsection to assign weights $w_i$ to each prior in the mixture $\mathcal{G}' = \{\mathcal{G}'_i\}_{i=1}^{M'}$, subject to $\sum_{i=1}^{M'} w_i = 1$. During pretraining, we sample generators $\mathcal{G}'_i$ proportional to $w_i$, and we generate synthetic datasets for pretraining. Given a sampled table $D^{(i)}$ from a generator $\mathcal{G}'_i$ in the mixture $\mathcal{G}'$, we randomly sub-sample $s$ entries as the support set or in-context examples and $q$ entries as the query samples. We then optimize the likelihood over the masked query labels: $\mathcal{L} = \mathbb{E}_D\big[\log p_\theta(y_{\text{qry}_1:\text{qry}_q} | \mathbf{x}_{\text{sup}_1:\text{sup}_s}, y_{\text{sup}_1:\text{sup}_s}, \mathbf{x}_{\text{qry}_1:\text{qry}_q})\big]$. Algorithm 1 in Appendix A.3 summarizes our synthetic prior generation procedure. We present more pretraining details in Appendix B.3.2. Figure 2 illustrates the overall MITRA model pipeline with two popular architectures: one-dimensional row-wise attention [16, 2], and two-dimensional element-wise attention [5, 17]. Our priors are model-agnostic, and we demonstrate their effectiveness across both architectures in the next section.

## 4 Empirical Results

In this section, we show that MITRA achieves SOTA performance on both classification and regression tasks (Section 4.2). We demonstrate that MITRA is model agnostic and consistently improves the performance with both 1D attention (MITRA 1D) and 2D attention architectures (Section 4.3). Furthermore, we highlight MITRA's better sample efficiency (Section 4.4), strong performance when combined with advanced ensembling techniques (Section 4.5), and strong fine-tuning performance (Section 4.6). We also conduct an ablation study to quantify the importance of each prior (Section 4.7), which supports our findings in Section 3.2. Finally, we analyze the scaling law with respect to both model size and synthetic dataset size (Section 4.8). See Appendix B for additional experimental setting details and Appendix C for additional experimental results.

### 4.1 Experimental Settings

**Datasets.** For the classification task, we compare MITRA on three established 10-fold benchmarks: TabRepo [30]; Tabzilla [25]; and AutoML benchmarks [11]. We additionally evaluate on a concurrent benchmark TabArena [8] in Appendix C.4. For the regression task, we compare on the 10-fold TabRepo [30] benchmark. We evaluate both MITRA and its variant MITRA 1D that is trained on a 1D attention model with the same mixture of priors. To compare with 1D models, e.g., TabPFN that support features up to 100, and 2D models, e.g., TabPFNv2 that support features up to 500, we evaluate on both small-feature and large-feature benchmarks. For the small-feature benchmark, we use 66 TabRepo classification datasets, 75 TabZilla classification datasets, and 10 TabRepo regression datasets, that have up to 3,000 rows and 100 features following TabPFN [16]. To evaluate

Table 2: **MITRA wins on all three** classification benchmarks (We show the overall merged results and detail the separate benchmark results in Appendix C.4, i.e., Table 13 (TabRepo), Table 14 (TabZilla), Table 15 (AMLB) due to space limits). Winner/runner-up in ▨ / ▨. +e means ensembling in ICL, and +f means fine-tuning. The $95\%$ confidence interval is shown in parentheses for the Elo. Aggregated metrics show mean and std (shown in parentheses) of the corresponding metric.

| Model | Ranking Metrics | | | | | Aggregated Metrics | | |
|---|---|---|---|---|---|---|---|---|
| | Avg. Rank ↓ | Elo ↑ | Winrate ↑ | RAcc ↑ | C Δ ↓ | AUC ↑ | ACC ↑ | CE ↓ |
| MITRA (+ef) | **7.2** | **1136** $_{(+4/-4)}$ | **0.69** | **0.82** | **20.1** | **0.905** $_{(0.124)}$ | **0.858** $_{(0.143)}$ | **0.328** $_{(0.317)}$ |
| Attic (+ef) | 7.4 | 1128 $_{(+4/-4)}$ | 0.68 | 0.81 | 21.7 | 0.903 $_{(0.125)}$ | 0.857 $_{(0.143)}$ | 0.332 $_{(0.317)}$ |
| TabPFNv2 (+e) | 8.0 | 1107 $_{(+4/-4)}$ | 0.65 | 0.8 | 23.3 | 0.901 $_{(0.13)}$ | 0.856 $_{(0.144)}$ | 0.338 $_{(0.318)}$ |
| TabPFNv2 (+ef) | 8.6 | 1085 $_{(+4/-4)}$ | 0.62 | 0.76 | 25.3 | 0.897 $_{(0.129)}$ | 0.846 $_{(0.15)}$ | 0.363 $_{(0.341)}$ |
| TabICL (+e) | 9.5 | 1053 $_{(+4/-4)}$ | 0.58 | 0.75 | 30.9 | 0.889 $_{(0.14)}$ | 0.836 $_{(0.15)}$ | 0.367 $_{(0.323)}$ |
| MITRA (+e) | 9.7 | 1046 $_{(+3/-3)}$ | 0.57 | 0.73 | 31.2 | 0.896 $_{(0.131)}$ | 0.847 $_{(0.148)}$ | 0.36 $_{(0.328)}$ |
| TabPFNv2 | 9.8 | 1043 $_{(+4/-3)}$ | 0.56 | 0.73 | 29.2 | 0.891 $_{(0.139)}$ | 0.846 $_{(0.147)}$ | 0.352 $_{(0.324)}$ |
| Attic (+e) | 9.9 | 1037 $_{(+4/-3)}$ | 0.55 | 0.73 | 31.8 | 0.896 $_{(0.13)}$ | 0.848 $_{(0.148)}$ | 0.364 $_{(0.328)}$ |
| TabICL | 10.6 | 1015 $_{(+4/-4)}$ | 0.52 | 0.7 | 33.5 | 0.884 $_{(0.141)}$ | 0.832 $_{(0.152)}$ | 0.374 $_{(0.323)}$ |
| MITRA | 10.6 | 1015 $_{(+4/-4)}$ | 0.52 | 0.69 | 32.9 | 0.891 $_{(0.134)}$ | 0.841 $_{(0.15)}$ | 0.368 $_{(0.33)}$ |
| MITRA 1D (+f) | 11.2 | 992 $_{(+4/-4)}$ | 0.49 | 0.68 | 34.5 | 0.893 $_{(0.13)}$ | 0.842 $_{(0.15)}$ | 0.367 $_{(0.331)}$ |
| Attic | 11.2 | 992 $_{(+3/-4)}$ | 0.49 | 0.68 | 35.2 | 0.884 $_{(0.139)}$ | 0.834 $_{(0.156)}$ | 0.376 $_{(0.332)}$ |
| CatBoost | 11.3 | 988 $_{(+4/-4)}$ | 0.48 | 0.67 | 35.7 | 0.888 $_{(0.133)}$ | 0.837 $_{(0.15)}$ | 0.375 $_{(0.324)}$ |
| TabForestPFN (+f) | 11.6 | 980 $_{(+4/-4)}$ | 0.47 | 0.65 | 35.3 | 0.886 $_{(0.136)}$ | 0.84 $_{(0.148)}$ | 0.377 $_{(0.331)}$ |
| RealMLP | 12.2 | 958 $_{(+4/-4)}$ | 0.44 | 0.62 | 35.4 | 0.878 $_{(0.142)}$ | 0.827 $_{(0.164)}$ | 0.412 $_{(0.394)}$ |
| XGBoost | 13.1 | 926 $_{(+4/-4)}$ | 0.4 | 0.58 | 39.5 | 0.883 $_{(0.133)}$ | 0.833 $_{(0.149)}$ | 0.388 $_{(0.323)}$ |
| LightGBM | 13.4 | 917 $_{(+4/-4)}$ | 0.38 | 0.56 | 39.8 | 0.876 $_{(0.141)}$ | 0.829 $_{(0.152)}$ | 0.393 $_{(0.328)}$ |
| Random Forest | 13.7 | 903 $_{(+4/-4)}$ | 0.36 | 0.54 | 44.1 | 0.874 $_{(0.14)}$ | 0.822 $_{(0.152)}$ | 0.471 $_{(0.423)}$ |
| MLP | 13.7 | 903 $_{(+4/-4)}$ | 0.36 | 0.51 | 40.1 | 0.869 $_{(0.145)}$ | 0.82 $_{(0.161)}$ | 0.414 $_{(0.346)}$ |
| MITRA 1D | 14.0 | 894 $_{(+4/-4)}$ | 0.35 | 0.55 | 42.5 | 0.869 $_{(0.143)}$ | 0.815 $_{(0.163)}$ | 0.414 $_{(0.351)}$ |
| TabForestPFN | 14.3 | 883 $_{(+4/-4)}$ | 0.34 | 0.52 | 42.6 | 0.864 $_{(0.153)}$ | 0.814 $_{(0.167)}$ | 0.414 $_{(0.353)}$ |

on large-feature benchmarks, we use 29 classification datasets from AMLB benchmark with up to 10,000 rows and 500 features. This is consistent with the evaluation protocol of TabPFNv2 [17]. We provide the dataset IDs in Appendix B.1.

**Baselines.** We compare MITRA with both SOTA TFMs (TabPFNv2 [17], TabICL [29], Attic [5], TabPFN [16], TabForestPFN [2]), and competitive classical and neural baselines requiring dataset-specific tuning (RealMLP [18], AutoGluon [7], LightGBM [20], XGBoost [3], CatBoost [28], MLP [7]). For the latter, we use their bagged version implemented in AutoGluon 1.3.

**Metrics.** For classification tasks, we report aggregated metrics including AUC-ROC (AUC), accuracy (ACC) and cross-entropy (CE). For regression tasks, we report $R^2$, root mean squared error (RMSE), and mean absolute error (MAE). Aggregated metrics can be disproportionately influenced by a small number of datasets with extreme performance, as noted in prior work [30]. To address this, we complement these metrics with more robust and comprehensive rank-based metrics that better capture relative performance across datasets: average rank, Elo [6], winrate, rescaled accuracy (RAcc), and champion delta (C∆). We provide the definitions of these metrics in Appendix B.2.

**Evaluation Protocols.** For TFMs including MITRA, we evaluate the following three settings: (1) *in-context learning* (ICL) performance; (2) *ICL with ensembling* techniques of feature shuffling, class order shuffling and random feature transformations [17, 29], denoted as "+e" in the following sections; and (3) *fine-tuning* that continues training the model on the training set of the target data, denoted as "+f" in the following sections. We use "bagging" to describe fine-tuning with bagging ensemble, and we show its advanced ensemble performance in Section 4.5.

## 4.2 MITRA achieves SOTA classification and regression performance

**Classification.** We merge the three classification benchmarks and keep a unique set of 137 datasets to report an overall ranking and aggregated performance in Table 2. Detailed method configurations and individual benchmark results on TabRepo (Table 13), TabZilla (Table 14), and AMLB (Table 15) are provided in Appendix C.4. MITRA wins in the overall results and across all three benchmarks with varying feature dimensionality or sample size, and it consistently achieves the best performance with fine-tuning and ensembling, across both ranking-based and aggregated metrics. Notably, the ICL performance of MITRA closely matches that of TabPFNv2, despite being pretrained on a maximum of 16 features (one-tenth of the maximum pretraining features in TabPFNv2), which showcases the strong generalizability of our priors.

Table 3: MITRA also demonstrates better performance with 1D attention model, showing that our priors are *model-agnostic*. Winner/runner-up in shades of green ▨ / ▨. The 95% confidence interval is shown in parentheses for the Elo. The columns in the aggregated metrics are mean and std (shown in parentheses) of the corresponding metric.

| Model | Ranking Metrics | | | | | Aggregated Metrics | | |
|---|---|---|---|---|---|---|---|---|
| | Avg. Rank ↓ | Elo ↑ | Winrate ↑ | RAcc ↑ | C Δ ↓ | AUC ↑ | ACC ↑ | CE ↓ |
| MITRA 1D (+f) | **3.0** | **1057** $_{(+7/-7)}$ | **0.6** | **0.68** | **16.5** | **0.886** $_{(0.135)}$ | **0.835** $_{(0.155)}$ | **0.38** $_{(0.349)}$ |
| TabForestPFN (+f) | 3.2 | 1038 $_{(+7/-6)}$ | 0.56 | 0.64 | 18.8 | 0.878 $_{(0.142)}$ | 0.832 $_{(0.154)}$ | 0.391 $_{(0.337)}$ |
| TabPFN (+e) | 3.4 | 1012 $_{(+6/-7)}$ | 0.52 | 0.6 | 24.4 | 0.862 $_{(0.155)}$ | 0.809 $_{(0.17)}$ | 0.418 $_{(0.346)}$ |
| MITRA 1D | 3.7 | 972 $_{(+6/-6)}$ | 0.45 | 0.53 | 27.2 | 0.865 $_{(0.147)}$ | 0.812 $_{(0.164)}$ | 0.417 $_{(0.343)}$ |
| TabPFN | 3.8 | 970 $_{(+7/-6)}$ | 0.45 | 0.53 | 26.3 | 0.86 $_{(0.156)}$ | 0.808 $_{(0.17)}$ | 0.426 $_{(0.349)}$ |
| TabForestPFN | 3.9 | 951 $_{(+7/-6)}$ | 0.42 | 0.49 | 27.7 | 0.859 $_{(0.159)}$ | 0.81 $_{(0.169)}$ | 0.419 $_{(0.349)}$ |

Table 4: **MITRA wins on TabRepo** 10-fold regression benchmark. Winner/runner-up in shades of green ▨ / ▨. +e means adding ensemble in ICL, and +f means adding fine-tuning.

| Model | Ranking Metrics | | | | | Aggregated Metrics | | |
|---|---|---|---|---|---|---|---|---|
| | Avg. Rank ↓ | Elo ↑ | Winrate ↑ | RAcc ↑ | C Δ ↓ | $R^2$ ↑ | RMSE ↓ | MAE ↓ |
| **MITRA (+ef)** | **4.3** | **1140** $_{(+20/-20)}$ | **0.7** | **0.82** | **10.9** | **0.636** $_{(0.306)}$ | 2401.274 $_{(7700.93)}$ | 1351.15 $_{(4100.89)}$ |
| TabPFNv2 (+e) | 5.1 | 1090 $_{(+22/-18)}$ | 0.63 | 0.72 | 12.7 | 0.615 $_{(0.332)}$ | 2374.55 $_{(7495.47)}$ | **1304.54** $_{(3960.35)}$ |
| RealMLP | 5.8 | 1044 $_{(+19/-20)}$ | 0.56 | 0.7 | 16 | 0.627 $_{(0.304)}$ | 2424.34 $_{(7574.57)}$ | 1385.57 $_{(4209.89)}$ |
| CatBoost | 5.8 | 1044 $_{(+19/-21)}$ | 0.56 | 0.69 | 15.7 | 0.629 $_{(0.301)}$ | 2465.09 $_{(7711.582)}$ | 1444.57 $_{(4383.87)}$ |
| TabPFNv2 (+ef) | 6.1 | 1023 $_{(+18/-18)}$ | 0.53 | 0.62 | 15.5 | 0.600 $_{(0.335)}$ | **2372.76** $_{(7513.58)}$ | 1295.36 $_{(3936.25)}$ |
| MITRA (+e) | 6.4 | 1008 $_{(+19/-21)}$ | 0.51 | 0.63 | 19.5 | 0.604 $_{(0.311)}$ | 2469.12 $_{(7922.80)}$ | 1372.43 $_{(4166.35)}$ |
| TabPFNv2 | 6.4 | 1008 $_{(+19/-21)}$ | 0.51 | 0.59 | 15.9 | 0.601 $_{(0.347)}$ | 2436.86 $_{(7790.76)}$ | 1337.64 $_{(4063.73)}$ |
| XGBoost | 6.7 | 989 $_{(+19/-18)}$ | 0.48 | 0.62 | 18.2 | 0.625 $_{(0.298)}$ | 2572.80 $_{(7975.02)}$ | 1573.52 $_{(4767.09)}$ |
| LightGBM | 6.8 | 984 $_{(+19/-19)}$ | 0.47 | 0.61 | 20.3 | 0.629 $_{(0.289)}$ | 2665.90 $_{(8218.40)}$ | 1571.68 $_{(4762.81)}$ |
| MITRA | 7.1 | 963 $_{(+19/-20)}$ | 0.44 | 0.59 | 20.6 | 0.599 $_{(0.317)}$ | 2465.02 $_{(7858.76)}$ | 1387.39 $_{(4214.92)}$ |
| MLP | 8.1 | 904 $_{(+20/-20)}$ | 0.36 | 0.5 | 22.3 | 0.595 $_{(0.328)}$ | 2778.66 $_{(8748.78)}$ | 1557.71 $_{(4729.35)}$ |
| Random Forest | 9.4 | 804 $_{(+22/-22)}$ | 0.23 | 0.32 | 27.9 | 0.585 $_{(0.319)}$ | 2797.35 $_{(8626.88)}$ | 1705.43 $_{(5161.03)}$ |

**Regression**. We evaluate on TabRepo regression datasets (Table 4)). MITRA again demonstrates the best performance across the various metrics, showing that our mixture of prior is task-agnostic.

### 4.3 MITRA priors are model agnostic

As shown in Table 3, when pretrained with the same mixture of priors, MITRA 1D also outperforms other 1D attention-based counterparts, e.g., TabPFN and TabForestPFN, which rely on less diverse priors. This highlights that our mixture of priors is model-agnostic and can consistently enhance performance across different architectures. Note that TabForestPFN does not offer native ensemble logic, and TabPFN does not offer native fine-tuning logic. We report results under the capabilities available in the respective baselines to ensure a fair comparison.

### 4.4 MITRA is more sample efficient

We compare the sample efficiency of MITRA against leading TFMs, i.e., TabPFNv2 and TabICL, in Table 5. We down-sample the number of ICL examples of TabRepo classification benchmark to 10%, 25%, 50%, and 75% of the original size, and MITRA consistently achieves better performance with ensemble and fine-tuning. We demonstrate in Appendix C.2 that such improvement can be attributed to the increased diversity of priors in the mixture, which enhances the model's ability to generalize from limited data.

### 4.5 MITRA shows the best performance with advanced ensembling techniques

To further boost performance, we implement a bagging-based ensemble for MITRA, denoted as MITRA (bagging). Specifically, we fine-tune a separate MITRA instance for each fold of an 8-fold (stratified) cross-validation ensemble [22], and we aggregate their predictions via uniform averaging at test time. This allows MITRA to benefit from both data-level diversity and model-level robustness. While cross-validation ensembles are widely used for achieving top performance with classical tabular models [7], to the best of our knowledge, our work is the first to demonstrate cross-validation ensembles for fine-tuned TFMs. We compare MITRA (bagging) against the strongest ensemble methods reported in previous works on TabRepo: the Post-Hoc Ensemble (PHE) of TabPFNv2, and the AutoGluon 1.3 best quality preset, which combines a diverse set of classical and neural tabular

Table 5: MITRA shows better sample efficiency. Winner in shades of green ▪.

| Model | Ranking Metrics | | | | | Aggregated Metrics | | |
|---|---|---|---|---|---|---|---|---|
| | Avg. Rank ↓ | Elo ↑ | Winrate ↑ | RAcc ↑ | C Δ ↓ | AUC ↑ | ACC ↑ | CE ↓ |
| MITRA (ds=1) | **4.2** | **1234** $_{(+8/-8)}$ | **0.77** | **0.88** | **13.7** | **0.882** $_{(0.125)}$ | **0.84** $_{(0.162)}$ | **0.36** $_{(0.361)}$ |
| TabPFNv2 (ds=1) | 4.5 | 1217 $_{(+7/-8)}$ | 0.75 | 0.87 | 16.3 | 0.879 $_{(0.127)}$ | 0.838 $_{(0.162)}$ | 0.372 $_{(0.361)}$ |
| TabICL (ds=1) | 5.6 | 1144 $_{(+7/-7)}$ | 0.67 | 0.82 | 25.4 | 0.858 $_{(0.145)}$ | 0.816 $_{(0.171)}$ | 0.406 $_{(0.369)}$ |
| MITRA (ds=0.75) | **5.3** | **1163** $_{(+8/-7)}$ | **0.69** | **0.84** | **20.3** | **0.876** $_{(0.129)}$ | **0.835** $_{(0.165)}$ | **0.371** $_{(0.366)}$ |
| TabPFNv2 (ds=0.75) | 5.5 | 1155 $_{(+8/-7)}$ | 0.68 | 0.83 | 21.9 | 0.872 $_{(0.134)}$ | 0.832 $_{(0.165)}$ | 0.384 $_{(0.365)}$ |
| TabICL (ds=0.75) | 6.7 | 1081 $_{(+7/-7)}$ | 0.59 | 0.77 | 29.6 | 0.852 $_{(0.146)}$ | 0.81 $_{(0.172)}$ | 0.419 $_{(0.372)}$ |
| MITRA (ds=0.5) | **6.7** | **1083** $_{(+7/-7)}$ | **0.59** | **0.78** | **28.3** | **0.868** $_{(0.135)}$ | **0.827** $_{(0.168)}$ | **0.388** $_{(0.372)}$ |
| TabPFNv2 (ds=0.5) | 6.9 | 1072 $_{(+7/-8)}$ | 0.58 | 0.77 | 29.4 | 0.864 $_{(0.138)}$ | 0.824 $_{(0.168)}$ | 0.401 $_{(0.369)}$ |
| TabICL (ds=0.5) | 7.9 | 1012 $_{(+7/-7)}$ | 0.51 | 0.72 | 34.9 | 0.844 $_{(0.146)}$ | 0.799 $_{(0.176)}$ | 0.439 $_{(0.375)}$ |
| MITRA (ds=0.25) | **9.4** | **923** $_{(+7/-7)}$ | **0.4** | **0.64** | **41.1** | **0.849** $_{(0.143)}$ | **0.808** $_{(0.174)}$ | **0.43** $_{(0.378)}$ |
| TabPFNv2 (ds=0.25) | 9.6 | 912 $_{(+7/-7)}$ | 0.39 | 0.62 | 42.8 | 0.843 $_{(0.147)}$ | 0.802 $_{(0.178)}$ | 0.441 $_{(0.372)}$ |
| TabICL (ds=0.25) | 10.5 | 855 $_{(+8/-8)}$ | 0.32 | 0.56 | 46.3 | 0.819 $_{(0.152)}$ | 0.776 $_{(0.181)}$ | 0.487 $_{(0.38)}$ |
| MITRA (ds=0.1) | **12.1** | **740** $_{(+9/-9)}$ | **0.21** | **0.36** | **54.1** | **0.808** $_{(0.151)}$ | **0.771** $_{(0.18)}$ | **0.515** $_{(0.377)}$ |
| TabPFNv2 (ds=0.1) | 12.4 | 719 $_{(+9/-10)}$ | 0.19 | 0.33 | 55.1 | 0.801 $_{(0.156)}$ | 0.764 $_{(0.182)}$ | 0.519 $_{(0.376)}$ |
| TabICL (ds=0.1) | 12.7 | 689 $_{(+10/-10)}$ | 0.16 | 0.24 | 56.8 | 0.777 $_{(0.153)}$ | 0.742 $_{(0.182)}$ | 0.561 $_{(0.384)}$ |

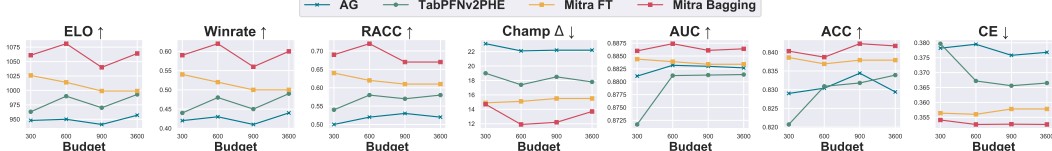

Figure 3: Mitra (bagging) shows the best performance compared with TabPFNv2 PHE, AutoGluon best-quality and Mitra (+ef) across 300- to 3600-second budgets on 1-fold TabRepo benchmark.

models. As shown in Figure 3, across different training budgets from 300, 600, 900 and 3600 seconds per dataset, MITRA (bagging) consistently outperforms both TabPFNv2 PHE and the AutoGluon ensemble, demonstrating the best performance with advanced ensembling techniques in the most competitive settings. Table 16 reports more details on ranking and aggregated performance over the unified classification benchmark.

## 4.6  MITRA shows the best fine-tuning performance

We compare the combined fine-tuning and ensemble performance of MITRA with TabPFNv2, TabICL, and Attic on TabRepo as a function of the number of estimators in the ensemble. As shown in Figure 4, MITRA consistently shows better fine-tuning performance across various ensemble sizes. Fine-tuning and ensembling of TabPFNv2 barely improves their ensemble-alone performance.[2] A likely reason for the strong gains from fine-tuning in MITRA is that it is pretrained with a maximum of 16 input features, so that adapting to downstream datasets with larger feature spaces provides substantial benefits. We also hypothesize that more diverse priors in MITRA contribute to its fine-tuning effectiveness, as they enable the model to generalize from a broader set of inductive biases, making the model more generalizable and adaptable to task-specific fine-tuning.

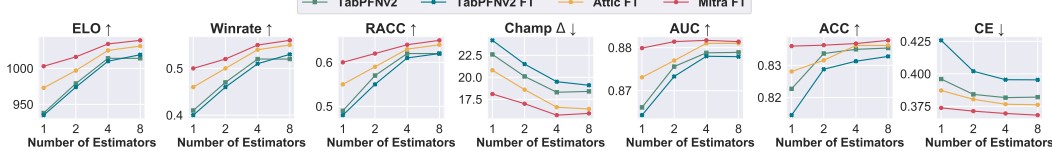

Figure 4: MITRA shows significantly better fine-tuning performance across ensemble sizes.

## 4.7  Prior Importance Ablation Study

We systematically study the importance of each prior in the MITRA prior mixture by iteratively adding the best-performing prior at each step. (See Appendix C.3 for details.) Table 6 presents several key findings that support and extend the analysis in Table 1 from Section 3.2: (1) **Both diversity and performance on real datasets are important.** Among all priors, data generated from

---

[2]We verified the correctness of the TabPFNv2 fine-tuning procedure via communication with its authors.

Table 6: TabRepo 10-fold classification ablation study on the effect of each prior in the mixture consisting of 50% SCM and 50% TBPs with equal weighting. The 95% confidence interval is shown in parentheses for the Elo. The columns in the aggregated metrics are mean and std (shown in parentheses) of the corresponding metric. Note that SCM + DT is a variant of Attic [5].

| Model | Ranking Metrics | | | | | Aggregated Metrics | | |
|---|---|---|---|---|---|---|---|---|
| | Avg. Rank ↓ | Elo ↑ | Winrate ↑ | RAcc ↑ | CΔ ↓ | AUC ↑ | ACC ↑ | CE ↓ |
| RF | 15.9 | $815_{(+7/-6)}$ | 0.25 | 0.38 | 42.8 | $0.831_{(0.147)}$ | $0.782_{(0.177)}$ | $0.521_{(0.468)}$ |
| DT | 15.4 | $840_{(+6/-7)}$ | 0.28 | 0.43 | 42.0 | $0.839_{(0.148)}$ | $0.791_{(0.177)}$ | $0.523_{(0.532)}$ |
| GB | 15.0 | $855_{(+6/-6)}$ | 0.3 | 0.52 | 38.3 | $0.843_{(0.147)}$ | $0.796_{(0.177)}$ | $0.501_{(0.464)}$ |
| ET | 14.3 | $883_{(+6/-6)}$ | 0.36 | 0.58 | 36.2 | $0.847_{(0.140)}$ | $0.803_{(0.169)}$ | $0.503_{(0.579)}$ |
| DSRF | 13.9 | $899_{(+6/-6)}$ | 0.36 | 0.58 | 34.5 | $0.849_{(0.144)}$ | $0.799_{(0.175)}$ | $0.473_{(0.410)}$ |
| **SCM** | 11.1 | $1000_{(+5/-6)}$ | 0.5 | 0.68 | 25.5 | $0.857_{(0.142)}$ | $0.812_{(0.176)}$ | $0.416_{(0.378)}$ |
| SCM + RF | 10.9 | $1005_{(+5/-6)}$ | 0.5 | 0.71 | 25.2 | $0.858_{(0.143)}$ | $0.816_{(0.173)}$ | $0.413_{(0.374)}$ |
| SCM + DT | 10.3 | $1025_{(+5/-5)}$ | 0.53 | 0.73 | 24.5 | $0.861_{(0.135)}$ | $0.816_{(0.169)}$ | $0.412_{(0.369)}$ |
| SCM + GB | 10.2 | $1028_{(+6/-5)}$ | 0.54 | 0.73 | 22.3 | $0.859_{(0.142)}$ | $0.817_{(0.172)}$ | $0.411_{(0.374)}$ |
| SCM + DSRF | 9.4 | $1058_{(+6/-5)}$ | 0.58 | 0.75 | 21.3 | $0.862_{(0.140)}$ | $0.819_{(0.171)}$ | $0.407_{(0.371)}$ |
| **SCM + ET** | 9.3 | $1063_{(+5/-5)}$ | 0.59 | 0.75 | **20.7** | $0.866_{(0.133)}$ | $0.823_{(0.169)}$ | $0.398_{(0.371)}$ |
| SCM + ET + DT | 10.2 | $1028_{(+5/-5)}$ | 0.54 | 0.74 | 22.4 | $0.869_{(0.129)}$ | $0.824_{(0.168)}$ | $0.403_{(0.371)}$ |
| SCM + ET + RF | 9.9 | $1042_{(+6/-5)}$ | 0.56 | 0.75 | 22.4 | **0.870**$_{(0.129)}$ | $0.825_{(0.166)}$ | $0.401_{(0.369)}$ |
| SCM + ET + DSRF | 9.6 | $1050_{(+5/-5)}$ | 0.57 | 0.74 | 22.1 | $0.860_{(0.139)}$ | $0.815_{(0.172)}$ | $0.410_{(0.373)}$ |
| **SCM + ET + GB** | **8.9** | **1076**$_{(+5/-6)}$ | **0.6** | **0.77** | **20.9** | **0.870**$_{(0.129)}$ | **0.827**$_{(0.164)}$ | **0.396**$_{(0.368)}$ |
| SCM + ET + GB + DSRF | 9.7 | $1046_{(+6/-6)}$ | 0.56 | 0.73 | 22.2 | $0.860_{(0.142)}$ | $0.817_{(0.171)}$ | $0.408_{(0.366)}$ |
| SCM + ET + GB + RF | 9.6 | $1050_{(+6/-6)}$ | 0.57 | 0.75 | 22.2 | $0.868_{(0.131)}$ | $0.823_{(0.167)}$ | $0.403_{(0.367)}$ |
| **SCM + ET + GB + DT** | 9.4 | $1059_{(+6/-5)}$ | 0.58 | 0.76 | 21.1 | **0.870**$_{(0.130)}$ | $0.826_{(0.165)}$ | $0.397_{0.369}$ |
| SCM + ET + GB + DT + DSRF | 9.5 | $1053_{(+5/-5)}$ | 0.57 | 0.75 | 22.9 | $0.859_{(0.140)}$ | $0.813_{(0.172)}$ | $0.411_{(0.368)}$ |
| **SCM + ET + GB + DT + RF** | 9.3 | $1062_{(+6/-5)}$ | 0.59 | 0.76 | 21.5 | $0.866_{(0.134)}$ | $0.823_{(0.167)}$ | $0.402_{(0.369)}$ |
| **SCM + ET + GB + DT + RF + DSRF** (MITRA) | 9.3 | $1062_{(+6/-6)}$ | 0.59 | **0.77** | 21.4 | $0.868_{(0.133)}$ | $0.822_{(0.168)}$ | $0.403_{(0.369)}$ |

$\mathcal{G}_{\text{SCM}}$ shows the highest importance, aligning with its high entry in the performance vector and low diagonal in Table 1. In contrast, despite being the second-best stand-alone prior, $\mathcal{G}_{\text{DSRF}}$ contributes the least when added to the mixture, due to its high diagonal (0.984) and strong overlap with other priors (off-diagonals > 0.96). On the other hand, while $\mathcal{G}_{\text{RF}}$ exhibits low overlap (lowest diagonal of 0.761 and off-diagonals in [0.708, 0.768]), it also shows the worst performance on real datasets, as indicated by the performance vector, thus explaining why it is not as beneficial in the mixture. (2) **Our mixture of priors promotes complementary strengths and improves generalization.** We observe that combining $\mathcal{G}_{\text{SCM}}$ with every tree-based prior improves over either alone, which emphasizes the importance of a prior mixture. In particular, we see that combining $\mathcal{G}_{\text{ET}}$ with $\mathcal{G}_{\text{SCM}}$ significantly boosts performance, yielding an Elo improvement of 63. This supports the findings in Table 1 that adding $\mathcal{G}_{\text{ET}}$ into the mixture is effective since it is both diverse, as measured by its lower diagonal (0.871), and distinctive, as measured by its low off-diagonal in the SCM row (0.751). Similarly, we see that adding $\mathcal{G}_{\text{GB}}$ to the mixture further improves the performance. Adding the remaining priors $\mathcal{G}_{\text{DT}}$, $\mathcal{G}_{\text{DSRF}}$, $\mathcal{G}_{\text{RF}}$ leads to a few configurations with similarly good performance on real datasets. This aligns with our findings in Table 1 that they are less important due to either lower performance on real datasets or higher diagonal or off-diagonal values. We choose to include these priors in the final mixture to represent a more complete family of tree priors. Moreover, including them in the full mixture improves sample efficiency (See Appendix C.2 that shows MITRA is more sample efficient than $\{\mathcal{G}_{\text{SCM}}, \mathcal{G}_{\text{ET}}, \mathcal{G}_{\text{GB}}\}$ (MITRA-Mix2) and Attic [5].)

## 4.8 Scaling Behavior for TFMs

Beyond prior construction, two critical factors influencing pretraining are the model size and the amount of training data. We investigate their respective scaling behaviors in Figure 5, by evaluating the performance on TabRepo given different model sizes and varying amount of pretraining data. Specifically, we vary the model depth across 6 configurations (4, 8, 12, 16, 20, 24 layers), all pretrained with the same on-the-fly generated mixture priors. For the *model size* scaling law, we observe that larger models achieve better performance in early training stages and also converge to higher final accuracy. However, performance gains begin to saturate beyond 12 layers, indicating a trade-off between model capacity and computational efficiency when selecting the appropriate model size. Regarding the *sample size* scaling law, we find that performance improves rapidly in the early stages and then gradually plateaus, with diminishing returns after approximately 18K steps. In our setting each step involves 2,048 new synthetic datasets, so this suggests the model saturates after encountering around 37 million unique datasets.

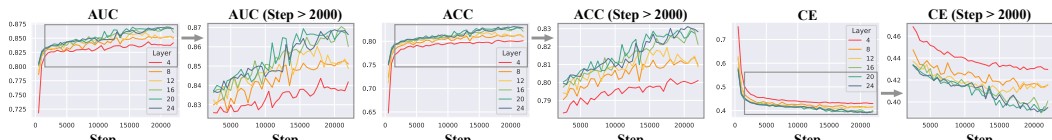

Figure 5: Model size scaling law. We vary the model size across six configurations (4, 8, 12, 16, 20, and 24 layers), each pretrained on 45 million unique samples. The plot zooms in on pretraining steps beyond 2,000 to provide a clearer view of performance trends. A larger version focusing only on this zoomed-in region is provided in Figure 25 in Appendix.

## 5    Conclusion and Discussion

**Conclusion.** We provide the first systematic investigation into the role of synthetic priors in pre-training TFMs, demonstrating how prior effectiveness depends on both its standalone performance on real tabular datasets as well as its diversity and distinctiveness within a mixture. Based on our analysis, we construct a diverse, high-performing, and model-agnostic mixture of synthetic priors. Leveraging this mixture, we develop MITRA, a SOTA TFM that consistently outperforms existing TFMs and other strong tabular baselines, across both classification and regression tasks. Limitations are discussed in Appendix D. **Broader Impact.** Our work advances the understanding and design of synthetic data for pretraining foundation models in structured domains, reducing the need for costly real labeled data and reducing privacy risks associated with training on sensitive real-world records.

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

# Contents

# List of Tables

## List of Figures

# A MITRA

In this section, we provide details on the in-context learning (ICL) preliminaries, our data generation methods and priors, and the overall MITRA algorithm.

## A.1 TFM Preliminaries

To pretrain a TFM on purely synthetic data, each dataset $\mathcal{D} = \{(\mathbf{x}_n, y_n)\}_{n=1}^N$ is sampled from a prior distribution $\mathcal{G}$ that generates datasets with varying numbers of features, samples, classes (for classification tasks), and categorical attributes. Once a TFM $f_\theta$ has been pretrained, it can be used to perform ICL as follows. The model is given a support set $\mathcal{D}_{\text{sup}}$ consisting of $s = N_{\text{sup}}$ labeled rows $\{(\mathbf{x}_{\text{sup}_n}, y_{\text{sup}_n})\}_{n=1}^s$, along with $q = N_{\text{qry}}$ unlabeled query rows $\{\mathbf{x}_{\text{qry}_n}\}_{n=1}^q$, where $s + q = N$. It then predicts the corresponding query labels $\{y_{\text{qry}_n}\}_{n=1}^q$ in a single forward pass:

$$\hat{y}_{\text{qry}_1}, \cdots, \hat{y}_{\text{qry}_q} = f_\theta([(\mathbf{x}_{\text{sup}_1}, y_{\text{sup}_1}), \cdots, (\mathbf{x}_{\text{sup}_s}, y_{\text{sup}_s}), \mathbf{x}_{\text{qry}_1}, \cdots, \mathbf{x}_{\text{qry}_q}]),$$

without the need to update its parameter $\theta$.

## A.2 Data Generation

Here, we discuss the specific parameters and modeling choices in the data generation for feature-target pairs $(\mathbf{x}, y) \in \mathcal{D}$ and the details on the priors that we use in our data-generating mixture of priors in MITRA. For simplicity, Figure 6 shows a visualization of a 2D dataset generated from our mixture of priors in MITRA. In addition, Figure 7 shows a t-SNE visualization of high-dimensional data samples from our mixture used during pretraining. We see both continuous and categorical features represented. The classification labels $y$ are depicted in color.

### A.2.1 Feature and Target Generation

**Feature $\mathbf{x}$.** We design $\mathbf{x}$ to include $N_{\text{cont}}$ continuous and $N_{\text{cat}}$ categorical components, such that $N_{\text{cont}} + N_{\text{cat}} = d$. The number of categorical components is determined by $N_{\text{cat}} = \lfloor p_{\text{cat}}(d+1) \rfloor$, where $p_{\text{cat}}$ is a categorical percentage uniformly sampled. We then uniformly sample the $N_{\text{cat}}$ categorical feature indices $\mathcal{I}_{\text{cat}}$ in $\mathcal{I} = \{1, \ldots, d\}$ without replacement. For each continuous feature index $j \in \mathcal{I}_{\text{cont}} = \mathcal{I} \setminus \mathcal{I}_{\text{cat}}$, we take $\mathbf{x}_j$ to be i.i.d. Gaussian noise. For each categorical feature $\mathbf{x}_k$, we generate its number of classes from a Geometric distribution. Lastly, we model each $\mathbf{x}_k$ via a multinomial distribution over its number of classes $N_c^k$.

**Target $y$.** For each target $y \in \mathcal{Y}$, we handle its generation differently depending on whether the generating prior uses "direct" or "indirect" sampling. For direct sampling, we directly simulate $y$ from a random conditional distribution $p(y \mid \mathbf{x})$. For indirect sampling, we first require fitting a classifier or regressor on a synthetically generated training dataset $(\mathbf{x}, y)$ for the corresponding task. In classification tasks, the label $y \in \mathcal{Y} \subset \mathbb{Z}$ is generated similarly to the aforementioned label generating process for the categorical features. In regression tasks, we take $y$ to be normal. The final label $y_2$ is generated as the output from the fitted estimator on a new feature vector $\mathbf{x}_2$.

### A.2.2 Synthetic Data-Generating Priors

We include a mixture of SCMs with TBPs, with both indirectly (ET, GB, DT, RF) and directly sampled priors (SCM, DSRF).

**Indirectly Sampled Priors.** Algorithm 2 shows the data-generating procedure for "indirectly" sampled priors. We refer to these priors as indirectly sampled methods since they first require training data $D = [\mathbf{X}, \mathbf{Y}] \in \mathbb{R}^{B \times (d+1)}$ to fit the estimator, i.e., classifier or regressor for the corresponding task, where $B$ denotes the base size. Then, we generate features $\mathbf{X}_2 \in \mathbb{R}^{N \times d}$ according to the feature generation procedure described in Subsection A.2.1. The final $\mathbf{Y}_2 \in \mathbb{R}^N$ is output from the fitted estimator by predicting on this $\mathbf{X}_2$ input. We note that TabForestPFN [2] and Attic [5] use an indirectly sampled DT prior, and TabICL [29] uses an indirectly sampled XGBoost prior. Our data generation is similar to that in TabForestPFN [2] and Attic [5] with the differences that we directly fit a Classifier for classification tasks rather than using a Regressor, and use our direct multinomial label

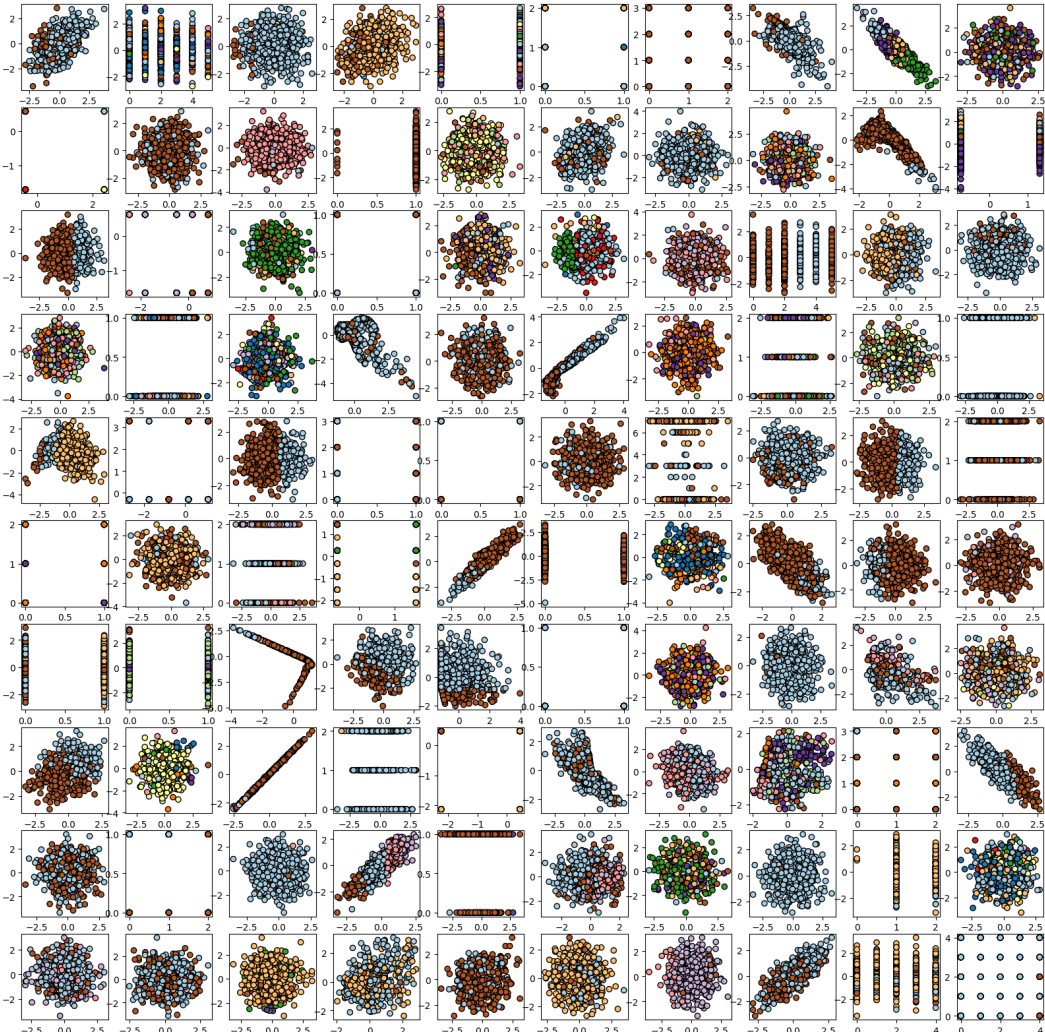

Figure 6: **Randomly generated 2D Dataset from mixture of priors in MITRA**. The features $\mathbf{x} \in \mathbb{R}^2$ drawn from our mixture of SCM and TBP data-generating priors in MITRA on classification tasks. The classification labels $y$ are shown in color.

generation procedure (see subsection A.2.1) for both the target labels and categorical feature labels. Hence, we eliminate the need for using the quantile transform to bucketize the continuous values to labels. For these indirectly sampled TBPs, we use the classifiers and regressors from scikit-learn [27].

**Directly Sampled Priors.** Algorithm 5 shows the data-generating procedure for our directly sampled random forest (DRSF) TBP prior. We refer to these priors as directly sampled because they first sample a function $f \in \mathcal{F}$ from function space $\mathcal{F}$ and then generate the targets $y_n = f(\mathbf{x}_n)$ for data $\mathbf{x}_n$. Hence, directly sampled methods only need to form $\mathbf{X}_2 \in \mathbb{R}^{N \times d}$ once and then the directly-sampled data-generating priors directly output $\mathbf{Y}_2 \in \mathbb{R}$.

For DSRF, we generate the features $\mathbf{X}_2$ using the same feature generation process from Subsection A.2.1. For DSRF, we must first construct random trees from $\mathcal{F}$ (see Algorithm 4). To do so, we sample the following: (1) random split indices in $\{0, \dots d-1\}$, where $d$ denotes the feature dimension; (2) random split thresholds in a specified range thres-int. Each node in the tree stores its corresponding split index and split threshold. We store the split intervals in a dictionary to track the sub-intervals corresponding to the feature split index to sample from on future splits as the algorithm progresses. We follow the convention from DTs, where the tree is split on a feature index $i$ and value $v$, and all datapoints such that $\mathbf{x}[i] \leq v$, are split to the left side of the tree and those such that

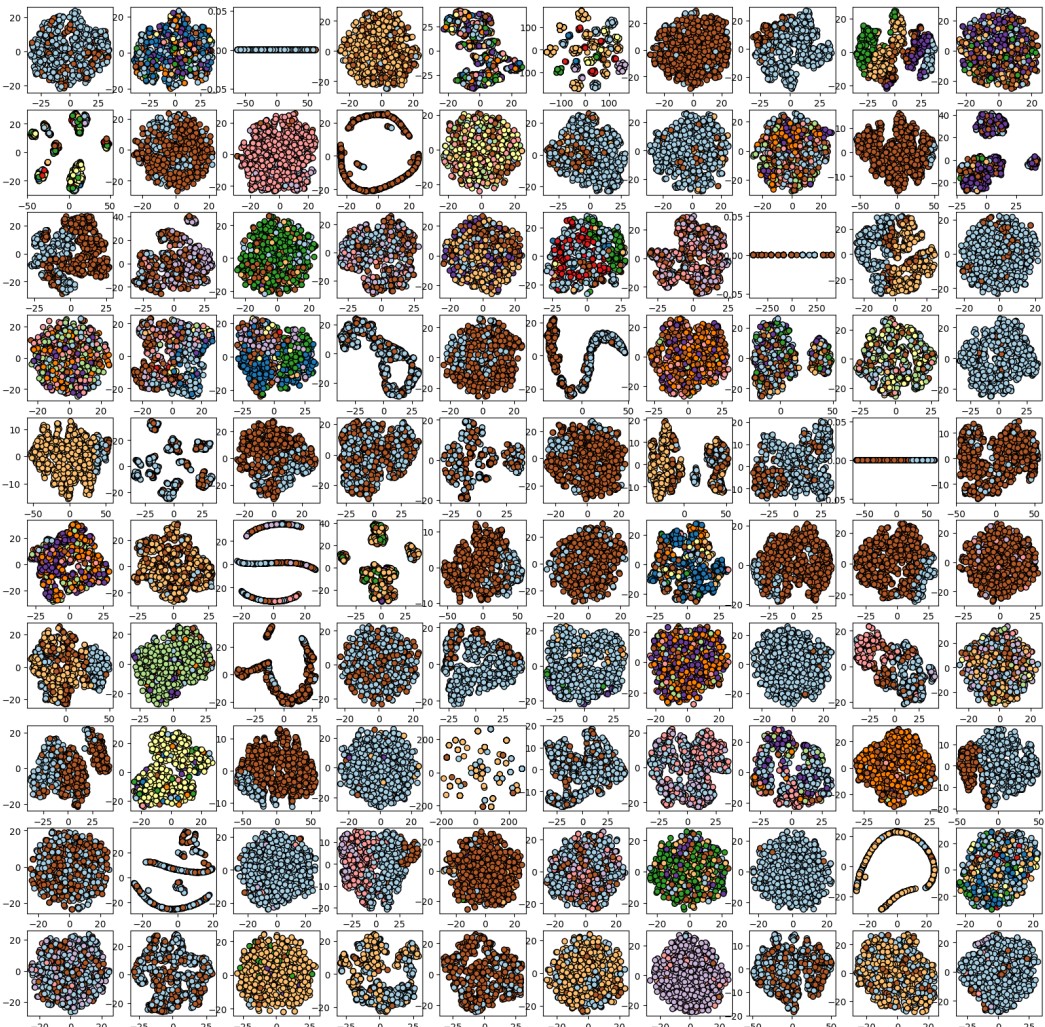

Figure 7: **t-SNE visualization of random high-dimensional dataset from mixture of priors in** **MITRA.** The features $\mathbf{x} \in \mathbb{R}^d$ drawn from our mixture of SCM and TBP data-generating priors in MITRA. The classification labels $y$ are shown in color.

$\mathbf{x}[i] > v$ are on the right side of the tree. We also control the number of nodes with no children. We construct the leaf node labels differently depending on the task type. For classification, we uniformly randomly sample a starting index in $\{0, \ldots, N_c - 1\}$, where $N_c$ denotes the number of classes, and we increment it for each subsequent leaf node added modulo the number of classes. For regression, we sample the leaf nodes from a Gaussian distribution. We sample number of estimators (random trees) $N_e$, and we traverse each tree until a leaf node is reached to get a target value for that tree (see Algorithm 3). Lastly, we compute the final label $y$ using majority voting over these $N_e$ values for classification tasks and using the mean for regression tasks.

For details on the SCM prior, see Appendix C.1 of the TabPFN paper [16].

## A.3 Algorithm

Algorithm 1 provides an overview of our MITRA method.

The population version of the likelihood in Section 3.3 takes a form of an expectation over the table $D$, with each table sampled from the prior mixture. Given the sampled table, the model assumes $q$ query rows are conditionally independent given the in-context examples, so that the log-likelihood within the expectation is decomposed into a sum of $q$ individual terms, one per query row. When

the query label corresponds to a classification task, its (conditional) distribution is assumed to be categorical, making the training objective equivalent to minimizing the cross-entropy loss. When the query label corresponds to a regression task, its (conditional) distribution is assumed to be Gaussian and the training objective corresponds to minimizing the Mean Squared Error (MSE) loss.

---

**Algorithm 1** MITRA: Mixture of Synthetic Priors for pretraining TFMs

---

**Require:** Set of data generators $\{\mathcal{G}_i\}_{i=1}^M$, with mixture weights $\{w_i\}_{i=1}^M$ such that $\sum_{i=1}^M w_i = 1$,
**Require:** Support set size $s$, query set size $q$, number of pretraining steps $T$.
 1: **for** $t = 1$ to $T$ **do**
 2:      Sample generator index $i \sim w_i$, for $i = 1, \ldots, M$.
 3:      Sample synthetic dataset $\mathcal{D}^{(i)} = \{(\mathbf{x}_n^{(i)}, y_n^{(i)})\}_{n=1}^{N_i} \leftarrow \mathcal{G}_i$.
 4:      Randomly partition $\mathcal{D}^{(i)}$ into support set $\mathcal{D}_{\text{sup}}^{(i)}$ and query set $\mathcal{D}_{\text{qry}}$ with $|\mathcal{D}_{\text{sup}}^{(i)}| = s$ and $|\mathcal{D}_{\text{qry}}^{(i)}| = q$.
 5:      Construct input sequence: $[(\mathbf{x}_{\text{sup}_1}^{(i)}, y_{\text{sup}_1}^{(i)}), \ldots, (\mathbf{x}_{\text{sup}_s}^{(i)}, y_{\text{sup}_s}^{(i)}), \mathbf{x}_{\text{qry}_1}^{(i)}, \ldots, \mathbf{x}_{\text{qry}_q}^{(i)}]$.
 6:      Predict query labels via TFM: $\hat{y}_{\text{qry}_{1:q}}^{(i)} = f_\theta(\cdot)$.
 7:      Compute loss: $\mathcal{L} = \log p_\theta(y_{\text{qry}_{1:q}}^{(i)} | \mathcal{D}_{\text{sup}}^{(i)}, \mathbf{x}_{\text{qry}_{1:q}}^{(i)})$.
 8:      Update model parameters $\theta$ using gradient of $\mathcal{L}$.
 9: **end for**

---

**Algorithm 2** Indirectly Sampled TBPs.

---

**Require:** Task generator TBP, base size $B$, feature dimension $d$, number of samples $N$, number of classes $N_c$, task type $\mathcal{T} \in \{\text{CLASSIFICATION}, \text{REGRESSION}\}$.
 1: Generate training data $\mathbf{D} = [\mathbf{X}, \mathbf{Y}] \in \mathbb{R}^{B \times (d+1)}$, where $\mathbf{X} = [\mathbf{x}_1; \ldots; \mathbf{x}_B] \in \mathbb{R}^{B \times d}$ and $\mathbf{Y} = [y_1, \ldots, y_B]^\top \in \mathbb{R}^B$, according to the procedure in Subsection A.2.1.
 2: Instantiate model $z \leftarrow \begin{cases} \text{TBPClassifier}, & \text{if } \mathcal{T} = \text{CLASSIFICATION}; \\ \text{TBPRegressor}, & \text{otherwise.} \end{cases}$
 3: Fit model: $z.\text{fit}(\mathbf{X}, \mathbf{Y})$.
 4: Sample testing features $\mathbf{X}_2 \in \mathbb{R}^{N \times d}$.
 5: Generate predictions: $\mathbf{Y}_2 \leftarrow z.\text{predict}(\mathbf{X}_2) \in \mathbb{R}^N$.
 6: **return** $[\mathbf{X}_2, \mathbf{Y}_2] \in \mathbb{R}^{N \times (d+1)}$.

---

**Algorithm 3** Target Generation From DT Traversal

---

 1: **function** DT-TRAVERSAL($\mathbf{x}$, tree)
 2:      $(\text{ind}, \text{thres}) \leftarrow \text{tree.value}$
 3:      **if** tree.target is not None **then**                    ▷ Leaf node reached
 4:          $y \leftarrow \text{tree.target}$
 5:      **else if** $\mathbf{x}[\text{ind}] \leq \text{thres}$ **then**
 6:          $y \leftarrow$ DT-TRAVERSAL($\mathbf{x}$, tree.left)           ▷ Traverse left subtree
 7:      **else**
 8:          $y \leftarrow$ DT-TRAVERSAL($\mathbf{x}$, tree.right)        ▷ Traverse right subtree
 9:      **end if**
10:      **return** $y$
11: **end function**

---

---

**Algorithm 4** Construct Random Decision Tree (DT)

---

**Require:** Feature dimension $d$, number of classes $N_c$, task type $\mathcal{T} \in$ {CLASSIFICATION, REGRESSION}, minimum and maximum tree depths $d_{\min} > 0$, $d_{\max}$, split threshold interval `thres-int`, probability of no children $p_{\text{nc}}$, Gaussian parameters for regression leaf nodes $(\mu, \sigma)$, global leaf counter $N_{\text{leaf}} = 0$.

1: **class** NODE(depth, $d_{\min}$, $d_{\max}$, thres-int)
2:     value $\leftarrow$ None                                                    ▷ Split rule: (feature index, threshold)
3:     left, right $\leftarrow$ None                                                         ▷ Child nodes
4:     d $\leftarrow$ depth                                                              ▷ Node depth
5:     target $\leftarrow$ None                                                      ▷ Leaf prediction value
6:     rs $\leftarrow$ {}                                               ▷ Dict. of (feature index, split intervals)
7:     Store $d_{\min}, d_{\max}$, and thres-int                   ▷ Min and max depths, threshold interval
8: **end class**

9: **function** RANDDT([lb, ub]=thres-int, tree=None, depth=0, rs = {}, ind = None)
10:     tree $\leftarrow$ NODE(depth, $d_{\min}$, $d_{\max}$, thres-int)
11:     **if** depth $> 0$ **then**
12:         tree.rs $\leftarrow$ rs;   tree.rs[ind] $\leftarrow$ [lb, ub]
13:     **else if** depth $= 0$ and $\mathcal{T} =$ CLASSIFICATION **then**
14:         target $\sim \mathcal{U}(\{0, \ldots, N_c - 1\})$
15:     **end if**
16:     ind $\sim \mathcal{U}(\{0, \ldots, d-1\})$
17:     [lb, ub] $\leftarrow$ tree.rs[ind] **if** ind in tree.rs, **else** tree.thres-int
18:     thres $\sim \mathcal{U}([\text{lb}, \text{ub}])$
19:     tree.value $\leftarrow$ (ind, thres)
20:     **if** (tree.d $< d_{\max}$ and $p \sim \mathcal{U}([0,1]) \geq p_{\text{nc}}$) or tree.d $< d_{\min}$ **then**          ▷ Add children
21:         tree.left $\leftarrow$ RANDDT([lb, thres], tree.left, tree.d $+ 1$, tree.rs, ind)
22:         tree.right $\leftarrow$ RANDDT([thres, ub], tree.right, tree.d $+ 1$, tree.rs, ind)
23:     **else**                                                        ▷ Assign target value to leaf node
24:         **if** $\mathcal{T} =$ CLASSIFICATION **then**
25:             tree.target $\leftarrow N_{\text{leaf}} \bmod N_c$
26:             $N_{\text{leaf}} \leftarrow N_{\text{leaf}} + 1$
27:         **else**
28:             tree.target $\sim \mathcal{N}(\mu, \sigma)$
29:         **end if**
30:     **end if**
31:     **return** tree
32: **end function**

---

**Algorithm 5** Directly Sampled Random Forest (DSRF)

---

**Require:** Feature dimension $d$, number of samples $N$, number of classes $N_c$, task type $\mathcal{T} \in$ {CLASSIFICATION, REGRESSION}, number of estimators $N_e$, minimum depth $d_{\min} > 0$, maximum depth $d_{\max}$, split threshold interval `thres-int`, probability of terminal node $p_{nc}$, Gaussian parameters for regression leaves $(\mu, \sigma)$.

1: Generate feature matrix $\mathbf{X}_2 \in \mathbb{R}^{N \times d}$ as described in Subsection A.2.1.
2: **for** $n = 1$ to $N$ **do**
3:     **for** $i = 1$ to $N_e$ **do**
4:         tree $\leftarrow$ RANDDT()                ▷ Sample a random DT from function space $\mathcal{F}$ (Alg. 4)
5:         target$[i] \leftarrow$ DT-TRAVERSAL($\mathbf{X}_2[n, :]$, tree)     ▷ Evaluate tree on input (Alg. 3)
6:     **end for**
7:     **if** $\mathcal{T} =$ CLASSIFICATION **then**
8:         $\mathbf{Y}_2[n] \leftarrow$ MAJORITYVOTE(target)
9:     **else**
10:        $\mathbf{Y}_2[n] \leftarrow$ MEAN(target)
11:     **end if**
12: **end for**
13: **return** $[\mathbf{X}_2, \mathbf{Y}_2] \in \mathbb{R}^{N \times (d+1)}$

---

# B Experimental Settings

In this section, we discuss the details on the benchmarking datasets, metrics, and implementation.

## B.1 Benchmarking Datasets

Table 7 provides the description and statistics of the benchmarking datasets with various number of rows and features. As discussed in Section 4.1, to compare with 1D models (e.g., TabPFN that support features up to 100) and 2D models (e.g., TabPFNv2) that support features up to 500, we evaluate on both small-feature and large-feature benchmarks. For the small-feature benchmark, we use 66 TabRepo classification datasets, 75 TabZilla classification datasets, and 10 TabRepo regression datasets, that have up to 3,000 rows and 100 features following TabPFN [16]. To evaluate on large-feature benchmarks, we use 29 classification datasets from AMLB benchmark with up to 10,000 rows and 500 features. We additionally evaluate on a large-feature regression benchmark of 28 datasets from AMLB and OpenML-CTR23 benchmarks, with up to 10,000 rows and 500 features. This is consistent with the evaluation protocol of TabPFNv2 [17].

The dataset task IDs are provided as follows:

**TabRepo**: 2, 11, 37, 2073, 2077, 3512, 3549, 3560, 3581, 3583, 3606, 3608, 3616, 3623, 3664, 3667, 3690, 3702, 3704, 3747, 3749, 3766, 3783, 3793, 3799, 3800, 3812, 3903, 3913, 3918, 9904, 9905, 9906, 9909, 9915, 9924, 9925, 9926, 9970, 9971, 9979, 14954, 125920, 125921, 146800, 146818, 146819, 168757, 168784, 190137, 190146, 359954, 359955, 359956, 359958, 359959, 359960, 359962, 359963, 361333, 361335, 361336, 361339, 361340, 361341, 361345

**AMLB**: 2073, 146818, 146820, 168350, 168757, 168784, 168911, 190137, 190146, 190392, 190410, 190411, 359954, 359955, 359956, 359958, 359959, 359960, 359961, 359962, 359963, 359964, 359965, 359968, 359969, 359970, 359972, 359974, 359975

**TabZilla**: 4, 9, 10, 11, 14, 15, 16, 18, 22, 23, 25, 27, 29, 31, 35, 37, 39, 40, 42, 47, 48, 50, 53, 54, 59, 2079, 2867, 3512, 3540, 3543, 3549, 3560, 3561, 3602, 3620, 3647, 3731, 3739, 3748, 3779, 3797, 3902, 3903, 3913, 3917, 3918, 9946, 9957, 9971, 9978, 9979, 9984, 10089, 10093, 10101, 14954, 14967, 125920, 125921, 145793, 145799, 145847, 145977, 145984, 146024, 146063, 146065, 146192, 146210, 146800, 146817, 146818, 146819, 146821, 146822

**TabRepoReg**: 167210, 359930, 359931, 359932, 359933, 359935, 359942, 359944, 359950, 359951

**AMLB + OpenML-CTR23**: [167210, 233215, 359930, 359931, 359932, 359933, 359934, 359939, 359940, 359942, 359944, 359945, 359948, 359950, 359951, 360945, 361235, 361236, 361237, 361243, 361251, 361256, 361258, 361259, 361617, 361619, 361621, 361622

Table 7: Benchmarking Datasets Description and Statistics.

| Dataset | Task | Num. Tables | Max Num. Rows | Max Num. Features |
|---|---|---|---|---|
| TabRepo | Classification | 66 | 3,000 | 100 |
| AMLB | Classification | 29 | 10,000 | 500 |
| Tabzilla | Classification | 75 | 3,000 | 100 |
| TabRepoReg | Regression | 10 | 3,000 | 100 |
| AMLB+OpenML-CTR23 | Regression | 28 | 10,000 | 500 |

## B.2 Metrics

To ensure a comprehensive evaluation of model performance across datasets and tasks, we employ a diverse set of ranking and aggregated metrics for both classification and regression tasks. Below, we provide their formal definitions used in this work. When computing the rank-based metrics, each fold of a dataset is considered as a separate evaluation unit.

### B.2.1 Ranking-Based Metrics

To assess model performance across a suite of datasets, we adopt the following ranking-based metrics:

**Average Rank.** Let $\mathcal{M}$ denote the set of models and $\mathcal{D}$ the set of datasets. For a model $m \in \mathcal{M}$ and dataset $\delta \in \mathcal{D}$, let $\text{rank}_\delta(m)$ denote the model's rank (1 is best) on dataset $\delta$ according to a performance metric (e.g., accuracy). The average rank is defined as:

$$\text{Avg. Rank}(m) = \frac{1}{|\mathcal{D}|} \sum_{\delta \in \mathcal{D}} \text{rank}_\delta(m).$$

**Elo Rating.** Elo rating generalizes pairwise win/loss outcomes into a competitive rating system. Each model is treated as a player, and its rating is updated based on pairwise performance comparisons across datasets. The final Elo rating reflects the model's relative strength across all pairwise matchups. We implement the metric based on existing work [6].

**Winrate.** Winrate captures the proportion of datasets on which a model outperforms other models. Formally, for a model $m$:

$$\text{Winrate}(m) = \frac{1}{|\mathcal{D}||\mathcal{M}| - 1} \sum_{\delta \in \mathcal{D}} \sum_{\substack{m' \in \mathcal{M} \\ m' \neq m}} \left( \mathbb{1}[\text{E}_\delta(m) < \text{E}_\delta(m')] + \frac{1}{2} \mathbb{1}[\text{E}_\delta(m) = \text{E}_\delta(m')] \right),$$

where error $\text{E} = 1 - \text{AUC}$ for classification task, and $\text{E} = 1 - \text{R}^2$ for regression task, and $\mathbb{1}[\cdot]$ denotes the indicator function. Tie contributes half a win.

**Rescaled Accuracy.** To address the effect of dataset difficulty on raw scores, we scale by the best-performing model within each dataset:

$$\text{RAcc}_\delta(m) = 1 - \text{Rescaled Error}_\delta(m), \text{Rescaled Error}_\delta(m) = \frac{\text{E}(m) - \text{E}(m^*)}{\text{E}(m') - \text{E}(m^*)},$$

where $m^* = \arg\min_m \text{E}(m)$ and $m' = \arg\max_m \text{E}(m)$ respectively denote the model with the smallest and largest errors. Error $\text{E} = 1 - \text{AUC}$ for classification task, and $\text{E} = 1 - \text{R}^2$ for regression task. The overall Rescaled Accuracy is the average over datasets:

$$\text{RAcc}(m) = \frac{1}{|\mathcal{D}|} \sum_{\delta \in \mathcal{D}} \text{RAcc}_\delta(m).$$

**Champion Delta.** Let $m^* = \arg\min_m \text{E}(m)$ denote the champion model. The Champion Delta for a model $m$ is defined as:

$$\text{C}\Delta(m) = \left( 1 - \frac{\text{E}(m^*)}{\text{E}(m)} \right) \times 100.$$

This reflects the percentage performance margin between the current model and the best-performing model. Error $\text{E} = 1 - \text{AUC}$ for classification task, and $\text{E} = 1 - \text{R}^2$ for regression task.

### B.2.2 Aggregated Classification Metrics

For aggregated classification metrics, we report standard metrics averaged across datasets.

**AUC (Area Under ROC Curve).** For a binary classifier, AUC measures the area under the Receiver Operating Characteristic curve, which plots the true positive rate (TPR) against the false positive rate (FPR). Formally:

$$\text{AUC} = \int_0^1 \text{TPR}(t) \, d\,\text{FPR}(t),$$

where $t$ is a threshold on predicted probability scores. For multiclass classification, we adopt the one-vs-one (OvO) strategy and compute the average AUC over all pairwise class comparisons.

**Accuracy (ACC).** Accuracy is the fraction of correctly classified instances:

$$\text{ACC} = \frac{1}{N} \sum_{i=1}^N \mathbb{1}[y_i = \hat{y}_i],$$

where $y_i$ is the ground-truth label, and $\hat{y}_i$ is the predicted label, and $\mathbb{1}[\cdot]$ denotes the indicator function.

**Cross-Entropy (CE).** Let $p_i$ be the predicted class probability vector for instance $i$, and $y_i$ the one-hot encoded ground-truth vector. The cross-entropy loss is:

$$\text{CE} = -\frac{1}{N} \sum_{i=1}^N \sum_{c=1}^C y_{i,c} \log p_{i,c},$$

where $C$ is the number of classes.

### B.2.3 Aggregated Regression Metrics

For aggregated regression metrics, we report standard metrics averaged across regression datasets.

**$R^2$ (Coefficient of Determination).** The $R^2$ score quantifies the proportion of variance explained by the model:

$$R^2 = 1 - \frac{\sum_{i=1}^N (y_i - \hat{y}_i)^2}{\sum_{i=1}^N (y_i - \bar{y})^2},$$

where $\bar{y}$ is the mean of the ground-truth values.

**Root Mean Squared Error (RMSE).** RMSE penalizes large prediction errors more heavily:

$$\text{RMSE} = \sqrt{\frac{1}{N} \sum_{i=1}^N (y_i - \hat{y}_i)^2},$$

**Mean Absolute Error (MAE).** MAE measures the average magnitude of prediction errors:

$$\text{MAE} = \frac{1}{N} \sum_{i=1}^N |y_i - \hat{y}_i|.$$

### B.3 Training and Inference

Our implementation is based on PYTORCH. We discuss the specific pretraining details and model hyperparameters below.

### B.3.1 Transformer Architecture

MITRA is built on Transformer architecture [33] with 12 layers, 512 embedding size and 4 attention heads. Each Transformer layer includes both row-wise attention and column-wise attention implemented using FlashAttention [4]. The resulting model contains 72M parameters. MITRA 1D is built on Transformer architecture, and each layer contains row-wise attention. The resulting model contains 37M parameters.

### B.3.2 MITRA Pretraining

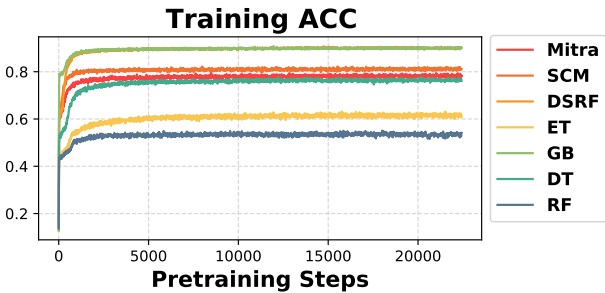

Figure 8: Comparison of training accuracy with respect to pretraining steps, with data generated from the mixture of priors in MITRA, vs. data from its prior individually.

For pretraining MITRA, we use eight 40GB A100 GPUs. MITRA is trained on 45 million synthetically generated datasets. This training takes approximately 60 hours on 8 GPUs (Nvidia A100s). To normalize features, we apply uniform quantile transform based on the support set, followed by standard normalization based on the mean and standard deviation from the support set. For regression tasks, we additionally apply min-max normalization for the target column using the minimum and maximum values of the support set for each table. Figure 8 illustrates that training curves tend to converge quickly around 2000 steps and are oscillatory. Interestingly, Figure 8 highlights the diversity findings in the diagonal of the ACC generalizability matrix in Table 9. We see that the training accuracy converges to a lower value for models generated with data from more diverse priors. The mixture priors of MITRA lie in between the less diverse TBPs (DSRF, GB) and the more diverse TBPs (RF, ET, DT), and slightly below SCM, which indicates that the TBPs add diversity to the mixture. In our experiments, we observe that the validation performance on real-world datasets improves as the pre-training continues.

### B.3.3 Ensembling and Finetuning Parameters for TFMs

For models incorporating ensembling, we use the default number of estimators for each model, i.e., 4 for TabPFNv2 on classification tasks, 8 for TabPFNv2 on regression tasks, 32 for TabICL, 3 for TabPFN. We find that on TabZilla and AMLB classification benchmarks, TabPFNv2 with 8 estimators performs better than using 4 estimators and the performance saturates after that. Accordingly, we increase the number of estimators to 8 for TabPFNv2 on these two benchmarks to ensure competitive performance. All models with +f are fine-tuned for 50 epochs, which is a setting that typically triggers early stopping on most datasets. Note that TabPFN only supports up to 100 features so it is not evaluated on the AMLB classification dataset. In addition, TabPFN and TabICL do not support regression. Attic has been observed to have training stability issues on regression tasks, and its reported performance is inferior to that of XGBoost, which we include as a baseline for regression. Table 8 summarizes the number of estimators for each dataset.

Table 8: Number of Estimators for the Ensemble TFMs.

| Dataset | MITRA (+ef) | Attic (+ef) | TabPFNv2 (+ef) | TabPFNv2 (+e) | MITRA (+e) | Attic (+e) | TabICL (+e) | TabPFN (+e) |
|---|---|---|---|---|---|---|---|---|
| TabRepo | 4 | 4 | 4 | 4 | 32 | 32 | 32 | 3 |
| AMLB | 8 | 8 | 8 | 8 | 32 | 32 | 32 | – |
| Tabzilla | 8 | 8 | 8 | 8 | 32 | 32 | 32 | 3 |
| Reg. | 8 | – | 8 | 8 | 32 | – | – | – |

### B.3.4 Statistical Model Hyperparameters

For RealMLP, LightGBM, XGBoost, CatBoost, MLP, we use their default hyperparameters in AutoGluon [7].

## C Additional Empirical Results

In this section, we report additional empirical results including the generalizability matrix **G** and performance vector **P** computed in other aggregated metrics, additional sample efficiency results and 2D decision boundary visualizations, ablations, classification and regression per dataset results, and the further improved performance of MITRA on the aggregated metrics using the advanced ensembling bagging method.

### C.1 Generalizability Matrix and Performance Vector Metrics

Here, we show the generalizability matrix **G** and performance vector **P** on TabRepo on metrics, i.e., accuracy (ACC) (Table 9) and cross-entropy (CE) (Table 10) in addition to the AUC table reported in Table 1. The rows of **G** denote the model pretrained with data generated from each prior. We then test those pretrained models on synthetic data generated from each prior distribution. We generate 100 tables, each with $N = 1000$ samples (rows), where the number of samples in the support $s = 800$ and the number of samples in the query $q = 200$.

We see that similar findings hold across the various aggregated metrics in these tables. In particular, Table 9 further emphasizes the distinctiveness of the TBP, ET, where the off-diagonal $\mathbf{G}_{ij}$ corresponding to the model pretrained with data drawn from $\mathcal{G}_{\text{ET}}$ is only 0.577. For the ranking-based metrics of each individual prior, see Table 6.

Table 9: Diversity (diagonal) and distinctiveness (off-diagonals) of each prior distribution. Each entry $\mathbf{G}_{ij}$ in the Generalizability Matrix represents the **ACC** of a TFM pretained on data generated from $\mathcal{G}_i$ and evaluated on data from $\mathcal{G}_j$. Each element $\mathbf{P}_i$ in the Performance Vector corresponds to the **ACC** of a TFM pretained on data from $\mathcal{G}_i$ and evaluated on real-world datasets.

| Train | Test | | | | | | |
| | SCM | ET | GB | DT | RF | DSRF | TabRepo |
|---|---|---|---|---|---|---|---|
| SCM | $0.841_{(0.155)}$ | $0.577_{(0.249)}$ | $0.902_{(0.093)}$ | $0.747_{(0.198)}$ | $0.562_{(0.241)}$ | $0.909_{(0.113)}$ | $0.812_{(0.176)}$ |
| ET | $0.808_{(0.159)}$ | $0.715_{(0.210)}$ | $0.928_{(0.077)}$ | $0.854_{(0.141)}$ | $0.613_{(0.239)}$ | $0.941_{(0.096)}$ | $0.803_{(0.169)}$ |
| GB | $0.800_{(0.174)}$ | $0.625_{(0.258)}$ | $0.942_{(0.060)}$ | $0.814_{(0.177)}$ | $0.588_{(0.252)}$ | $0.944_{(0.092)}$ | $0.796_{(0.177)}$ |
| DT | $0.799_{(0.168)}$ | $0.712_{(0.213)}$ | $0.929_{(0.077)}$ | $0.861_{(0.134)}$ | $0.611_{(0.238)}$ | $0.939_{(0.098)}$ | $0.791_{(0.177)}$ |
| RF | $0.807_{(0.160)}$ | $0.629_{(0.235)}$ | $0.904_{(0.096)}$ | $0.785_{(0.177)}$ | $0.602_{(0.230)}$ | $0.907_{(0.112)}$ | $0.782_{(0.177)}$ |
| DSRF | $0.806_{(0.211)}$ | $0.672_{(0.235)}$ | $0.927_{(0.094)}$ | $0.840_{(0.154)}$ | $0.607_{(0.240)}$ | $0.952_{(0.081)}$ | $0.799_{(0.175)}$ |

Table 10: Diversity (diagonal) and distinctiveness (off-diagonals) of each prior distribution. Each entry $\mathbf{G}_{ij}$ in the Generalizability Matrix represents the **CE** of a TFM pretained on data generated from $\mathcal{G}_i$ and evaluated on data from $\mathcal{G}_j$. Each element $\mathbf{P}_i$ in the Performance Vector corresponds to the **CE** of a TFM pretained on data from $\mathcal{G}_i$ and evaluated on real-world datasets.

| Train | Test | | | | | | |
| | SCM | ET | GB | DT | RF | DSRF | TabRepo |
|---|---|---|---|---|---|---|---|
| SCM | $0.366_{(0.359)}$ | $1.054_{(0.609)}$ | $0.309_{(0.275)}$ | $0.690_{(0.521)}$ | $1.128_{(0.627)}$ | $0.273_{(0.303)}$ | $0.416_{(0.378)}$ |
| ET | $0.465_{(0.387)}$ | $0.764_{(0.565)}$ | $0.220_{(0.233)}$ | $0.399_{(0.381)}$ | $1.001_{(0.642)}$ | $0.173_{(0.270)}$ | $0.503_{(0.579)}$ |
| GB | $0.469_{(0.403)}$ | $1.022_{(0.720)}$ | $0.162_{(0.165)}$ | $0.535_{(0.503)}$ | $1.107_{(0.714)}$ | $0.163_{(0.254)}$ | $0.501_{(0.464)}$ |
| DT | $0.475_{(0.400)}$ | $0.777_{(0.574)}$ | $0.216_{(0.231)}$ | $0.383_{(0.368)}$ | $1.007_{(0.648)}$ | $0.177_{(0.271)}$ | $0.523_{(0.532)}$ |
| RF | $0.457_{(0.370)}$ | $0.967_{(0.623)}$ | $0.273_{(0.264)}$ | $0.579_{(0.474)}$ | $1.028_{(0.628)}$ | $0.248_{(0.296)}$ | $0.521_{(0.468)}$ |
| DSRF | $0.449_{(0.492)}$ | $0.868_{(0.632)}$ | $0.219_{(0.296)}$ | $0.445_{(0.420)}$ | $1.021_{(0.655)}$ | $0.134_{(0.222)}$ | $0.473_{(0.410)}$ |

### C.2 Sample Efficiency

Table 11 illustrates the improved sample efficiency of MITRA compared to its ablations: MITRA-Mix2 (SCM + ET + GB) and Attic (a variant of SCM + DT). In particular, the differences in the Elo become larger as the downsampling ratio, ds, is further decreased with the largest gains occuring when ds = 0.1. This result emphasizes the generalizability of our mixture priors in comparison to mixtures of less priors in data-scarce scenarios.

Table 11: More diverse priors (MITRA) show better sample efficiency. Winner under each down-sampling ratio ds $\in \{1.0, 0.75, 0.5, 0.25, 0.1\}$ in **bold**.

| Model | Ranking Metrics | | | | | Aggregated Metrics | | |
|---|---|---|---|---|---|---|---|---|
| | Avg. Rank ↓ | Elo ↑ | Winrate ↑ | RAcc ↑ | C Δ ↓ | AUC ↑ | ACC ↑ | CE ↓ |
| **MITRA-Mix2** (ds=1.0) | **4.4** | **1234** $_{(+8/-8)}$ | **0.76** | **0.88** | 12.1 | 0.8817$_{(0.1256)}$ | **0.8407** $_{(0.1607)}$ | 0.3612$_{(0.3624)}$ |
| MITRA (ds=1.0) | 4.5 | 1227$_{(+8/-8)}$ | 0.75 | **0.88** | **11.7** | **0.8818** $_{(0.125)}$ | 0.8396$_{(0.1625)}$ | **0.3609** $_{(0.3614)}$ |
| Attic (ds=1.0) | 4.6 | 1221$_{(+8/-8)}$ | 0.75 | 0.87 | 12.8 | 0.8812$_{(0.125)}$ | 0.839$_{(0.1628)}$ | 0.3682$_{(0.3607)}$ |
| **MITRA** (ds=0.75) | **5.6** | **1151** $_{(+7/-8)}$ | **0.67** | **0.83** | **18.5** | **0.8758** $_{(0.1289)}$ | **0.8344** $_{(0.1647)}$ | **0.3716** $_{(0.3668)}$ |
| Attic (ds=0.75) | **5.6** | 1150$_{(+7/-7)}$ | **0.67** | **0.83** | 19.0 | 0.8743$_{(0.13)}$ | 0.8324$_{(0.165)}$ | 0.3795$_{(0.3651)}$ |
| MITRA-Mix2 (ds=0.75) | 5.7 | 1146$_{(+7/-7)}$ | 0.66 | 0.82 | 18.6 | 0.8755$_{(0.1292)}$ | 0.8338$_{(0.1641)}$ | 0.3727$_{(0.3672)}$ |
| **MITRA** (ds=0.5) | **7.1** | **1060** $_{(+7/-7)}$ | **0.56** | **0.76** | 26.6 | **0.8684** $_{(0.135)}$ | **0.8267** $_{(0.1679)}$ | **0.3888** $_{(0.3723)}$ |
| MITRA-Mix2 (ds=0.5) | 7.2 | 1056$_{(+7/-7)}$ | **0.56** | **0.76** | **26.3** | 0.8674$_{(0.135)}$ | 0.8257$_{(0.1695)}$ | 0.3899$_{(0.3726)}$ |
| Attic (ds=0.5) | 7.3 | 1047$_{(+7/-7)}$ | 0.55 | 0.75 | 27.7 | 0.8665$_{(0.1351)}$ | 0.824$_{(0.1683)}$ | 0.3965$_{(0.3696)}$ |
| **MITRA** (ds=0.25) | **9.9** | **888** $_{(+7/-8)}$ | **0.36** | **0.59** | **39.7** | **0.8492** $_{(0.1427)}$ | **0.8076** $_{(0.1744)}$ | **0.4317** $_{(0.3781)}$ |
| MITRA-Mix2 (ds=0.25) | 10.0 | 884$_{(+7/-7)}$ | **0.36** | **0.59** | 40.1 | 0.8488$_{(0.1412)}$ | 0.807$_{(0.1746)}$ | 0.4327$_{(0.3776)}$ |
| Attic (ds=0.25) | 10.2 | 874$_{(+7/-8)}$ | 0.35 | 0.58 | 41.5 | 0.8457$_{(0.1439)}$ | 0.8043$_{(0.1748)}$ | 0.4416$_{(0.3727)}$ |
| **MITRA** (ds=0.1) | **12.5** | **700** $_{(+9/-9)}$ | **0.18** | **0.25** | **53.3** | **0.8078** $_{(0.1505)}$ | **0.7702** $_{(0.1799)}$ | **0.5158** $_{(0.3772)}$ |
| MITRA-Mix2 (ds=0.1) | 12.7 | 681$_{(+11/-10)}$ | 0.17 | 0.23 | 53.8 | 0.8065$_{(0.1499)}$ | 0.7662$_{(0.1796)}$ | 0.5182$_{(0.3766)}$ |
| Attic (ds=0.1) | 12.7 | 680$_{(+8/-11)}$ | 0.16 | 0.22 | 53.7 | 0.7995$_{(0.1532)}$ | 0.7648$_{(0.1794)}$ | 0.5237$_{(0.377)}$ |

## C.3  Ablations

In Table 6, we perform an ablation study to analyze the importance of each prior in our mixture. We begin with ranking the performance of the model pretrained on data drawn from each prior individually. For the next step, we select the prior with the best performance, which in this case is SCM. We then add each remaining prior one-by-one with that selected prior to determine the next best pairing. We continue this procedure until we have added all the priors in MITRA in a forward process. We see that the ranking of the priors in decreasing order is $\{SCM, ET, GB, DT, RF, DSRF\}$, which aligns with the performance, diversity and distinctiveness findings from the generalizability matrices $\mathbf{G}$ and performance vectors $\mathbf{P}$ in the various metrics in Tables 1 and Tables 9 - 10.

In Table 12, we study the effect of the percentage $p$ between SCM and the tree-based priors (TBP) in our mixture of priors in MITRA on the TabRepo dataset. In our experimental results, we set $p = 0.5$ for an equal weighting of SCM and TBP. We see that on TabRepo there are several values of $p$ that show improved performance of the mixture over SCM alone ($p = 1.0$) and TBP alone ($p = 0$). The average rankings for $p = 0.5, 0.4, 0.6$ are tied at 3.2 with slightly better Elo for $p = 0.5$ at 1040 vs. 1030 for the other variants. In addition, SCM alone has better performance over the TBPs alone with an average ranking of 3.7 vs. 4.4. The mixture of TBPs without SCM has a lack of distinctiveness since the models trained on data drawn from a TBP can predict on data drawn from these other TBPs better than models trained on SCM can, as measured by the off-diagonals of the generalizability matrix $\mathbf{G}$. Removing SCM from the mixture also removes the top performing prior as measured by the performance vector $\mathbf{P}$. Note that priors works have combined SCM with 1 TBP but not multiple TBPs. In addition, these works have not studied the effect of the percentage of SCM and the corresponding TBP, e.g., TabICL [29] combines $p = 0.7$ for SCM with $p = 0.3$ XGBoost-based SCM, and TabForestPFN [2] and Attic [5] combine $p = 0.5$ SCM with $p = 0.5$ DT.

Table 12: TabRepo 10-fold classification ablation study on the effect of the percentage $p$ of SCM in the mixture of priors and the total amount of tree-based priors (TBP) is $1 - p$.

| $p$ | Ranking Metrics | | | | | Aggregated Metrics | | |
|---|---|---|---|---|---|---|---|---|
| | Avg. Rank ↓ | Elo ↑ | Winrate ↑ | RAcc ↑ | C Δ ↓ | AUC ↑ | ACC ↑ | CE ↓ |
| **0.5** (**MITRA**) | **3.2** | **1040** $_{(+9/-10)}$ | **0.57** | **0.69** | **14.3** | **0.868** $_{(0.133)}$ | **0.822** $_{(0.168)}$ | **0.403** $_{(0.369)}$ |
| 0.4 | **3.2** | 1030 $_{(+9/-9)}$ | 0.55 | 0.68 | 15.1 | 0.864$_{(0.133)}$ | 0.817$_{(0.171)}$ | 0.406$_{(0.369)}$ |
| 0.6 | **3.2** | 1030 $_{(+9/-9)}$ | 0.55 | 0.67 | 14.6 | 0.865 $_{(0.135)}$ | 0.822 $_{(0.169)}$ | 0.404 $_{(0.368)}$ |
| 0.7 | 3.3 | 1029$_{(+10/-10)}$ | 0.55 | 0.66 | 15.3 | 0.863$_{(0.139)}$ | 0.818 $_{(0.171)}$ | 0.406$_{(0.368)}$ |
| 1.0 (SCM) | 3.7 | 983$_{(+10/-10)}$ | 0.47 | 0.57 | 19.1 | 0.857$_{(0.142)}$ | 0.812$_{(0.176)}$ | 0.416$_{(0.378)}$ |
| 0.0 (TBP) | 4.4 | 888$_{(+10/-11)}$ | 0.32 | 0.35 | 29.5 | 0.856$_{(0.137)}$ | 0.808$_{(0.168)}$ | 0.451$_{(0.392)}$ |

Table 13: **MITRA wins on TabRepo** 10-fold classification benchmark. Winner/runner-up in ▨ / ▨. +e means ensembling in ICL, and +f means fine-tuning. The 95% confidence interval is shown in parentheses for the Elo. The columns in the aggregated metrics are mean and std (shown in parentheses) of the corresponding metric.

| Model | Ranking Metrics | | | | | Aggregated Metrics | | |
|---|---|---|---|---|---|---|---|---|
| | Avg. Rank ↓ | Elo ↑ | Winrate ↑ | RAcc ↑ | C Δ ↓ | AUC ↑ | ACC ↑ | CE ↓ |
| **MITRA** (+ef) | **7.8** | **1141** $_{(+5/-5)}$ | **0.69** | **0.8** | **17.9** | **0.882** $_{(0.124)}$ | **0.837** $_{(0.164)}$ | **0.370** $_{(0.367)}$ |
| Attic (+ef) | 7.9 | 1135 $_{(+5/-6)}$ | 0.69 | 0.79 | 19.2 | 0.880 $_{(0.125)}$ | 0.835 $_{(0.163)}$ | 0.377 $_{(0.366)}$ |
| TabPFNv2 (+e) | 8.4 | 1120 $_{(+5/-6)}$ | 0.67 | 0.78 | 20.4 | 0.879 $_{(0.126)}$ | 0.835 $_{(0.164)}$ | 0.382 $_{(0.367)}$ |
| TabPFNv2 (+ef) | 8.9 | 1102 $_{(+6/-6)}$ | 0.64 | 0.76 | 21.6 | 0.878 $_{(0.125)}$ | 0.831 $_{(0.162)}$ | 0.396 $_{(0.367)}$ |
| MITRA (+e) | 9.6 | 1076 $_{(+5/-6)}$ | 0.61 | 0.74 | 25.8 | 0.877 $_{(0.126)}$ | 0.831 $_{(0.164)}$ | 0.394 $_{(0.367)}$ |
| Attic (+e) | 10.0 | 1063 $_{(+5/-5)}$ | 0.59 | 0.74 | 26.7 | 0.877 $_{(0.126)}$ | 0.830 $_{(0.164)}$ | 0.401 $_{(0.367)}$ |
| TabICL (+e) | 10.2 | 1057 $_{(+5/-6)}$ | 0.58 | 0.73 | 28.6 | 0.859 $_{(0.144)}$ | 0.813 $_{(0.172)}$ | 0.415 $_{(0.374)}$ |
| TabPFNv2 | 10.3 | 1055 $_{(+5/-5)}$ | 0.58 | 0.72 | 24.8 | 0.866 $_{(0.136)}$ | 0.823 $_{(0.167)}$ | 0.396 $_{(0.371)}$ |
| MITRA | 10.7 | 1043 $_{(+5/-5)}$ | 0.56 | 0.71 | 28.1 | 0.868 $_{(0.133)}$ | 0.821 $_{(0.168)}$ | 0.403 $_{(0.369)}$ |
| TabICL | 11.4 | 1020 $_{(+6/-5)}$ | 0.53 | 0.69 | 30.7 | 0.853 $_{(0.144)}$ | 0.807 $_{(0.176)}$ | 0.422 $_{(0.375)}$ |
| Attic | 11.7 | 1009 $_{(+5/-5)}$ | 0.51 | 0.67 | 30.7 | 0.858 $_{(0.141)}$ | 0.812 $_{(0.174)}$ | 0.413 $_{(0.370)}$ |
| CatBoost | 11.9 | 1003 $_{(+5/-5)}$ | 0.50 | 0.68 | 32.3 | 0.865 $_{(0.130)}$ | 0.816 $_{(0.168)}$ | 0.416 $_{(0.374)}$ |
| MITRA 1D (+f) | 12.4 | 987 $_{(+5/-6)}$ | 0.48 | 0.65 | 31.7 | 0.868 $_{(0.133)}$ | 0.822 $_{(0.171)}$ | 0.405 $_{(0.377)}$ |
| TabForestPFN (+f) | 12.6 | 980 $_{(+5/-5)}$ | 0.47 | 0.64 | 33.2 | 0.861 $_{(0.135)}$ | 0.817 $_{(0.167)}$ | 0.417 $_{(0.373)}$ |
| RealMLP | 13.4 | 955 $_{(+6/-5)}$ | 0.44 | 0.6 | 33.3 | 0.851 $_{(0.140)}$ | 0.802 $_{(0.183)}$ | 0.453 $_{(0.402)}$ |
| TabPFNv1 (+e) | 14.2 | 929 $_{(+5/-5)}$ | 0.40 | 0.55 | 36.8 | 0.832 $_{(0.151)}$ | 0.787 $_{(0.183)}$ | 0.463 $_{(0.385)}$ |
| LightGBM | 14.3 | 926 $_{(+5/-5)}$ | 0.40 | 0.58 | 36.1 | 0.858 $_{(0.133)}$ | 0.812 $_{(0.169)}$ | 0.427 $_{(0.373)}$ |
| XGBoost | 14.3 | 924 $_{(+5/-6)}$ | 0.39 | 0.57 | 37.5 | 0.859 $_{(0.131)}$ | 0.813 $_{(0.166)}$ | 0.429 $_{(0.369)}$ |
| MITRA 1D | 15.0 | 902 $_{(+5/-6)}$ | 0.36 | 0.54 | 38.3 | 0.842 $_{(0.140)}$ | 0.794 $_{(0.179)}$ | 0.448 $_{(0.381)}$ |
| Random Forest | 15.0 | 901 $_{(+6/-5)}$ | 0.36 | 0.52 | 40.9 | 0.844 $_{(0.136)}$ | 0.797 $_{(0.170)}$ | 0.540 $_{(0.513)}$ |
| TabPFNv1 | 15.0 | 901 $_{(+5/-5)}$ | 0.36 | 0.51 | 38.4 | 0.829 $_{(0.151)}$ | 0.785 $_{(0.184)}$ | 0.469 $_{(0.386)}$ |
| MLP | 15.3 | 890 $_{(+5/-5)}$ | 0.35 | 0.47 | 37.7 | 0.839 $_{(0.141)}$ | 0.794 $_{(0.180)}$ | 0.464 $_{(0.387)}$ |
| TabForestPFN | 15.6 | 880 $_{(+6/-6)}$ | 0.34 | 0.5 | 40.2 | 0.834 $_{(0.156)}$ | 0.791 $_{(0.185)}$ | 0.447 $_{(0.380)}$ |

Table 14: **MITRA wins on TabZilla** 10-fold classification benchmark. Winner/runner-up in ▨ / ▨. +e means adding ensemble in ICL, and +f means adding fine-tuning.

| Model | Ranking Metrics | | | | | Aggregated Metrics | | |
|---|---|---|---|---|---|---|---|---|
| | Avg. Rank ↓ | Elo ↑ | Winrate ↑ | RAcc ↑ | C Δ ↓ | AUC ↑ | ACC ↑ | CE ↓ |
| **MITRA** (+ef) | **8.6** | **1110** $_{(+5/-4)}$ | **0.66** | **0.82** | **20.7** | **0.913** $_{(0.132)}$ | 0.867 $_{(0.143)}$ | **0.304** $_{(0.302)}$ |
| Attic (+ef) | 8.8 | 1102 $_{(+5/-5)}$ | 0.65 | 0.82 | 21.8 | 0.912 $_{(0.133)}$ | **0.867** $_{(0.142)}$ | 0.305 $_{(0.302)}$ |
| TabPFNv2 (+e) | 9.5 | 1079 $_{(+5/-5)}$ | 0.61 | 0.8 | 25.1 | 0.909 $_{(0.142)}$ | 0.863 $_{(0.144)}$ | 0.315 $_{(0.301)}$ |
| TabPFNv2 (+ef) | 9.9 | 1067 $_{(+5/-4)}$ | 0.6 | 0.77 | 25.7 | 0.903 $_{(0.143)}$ | 0.851 $_{(0.156)}$ | 0.345 $_{(0.342)}$ |
| TabICL (+e) | 10.3 | 1055 $_{(+4/-4)}$ | 0.58 | 0.77 | 30.0 | 0.907 $_{(0.141)}$ | 0.850 $_{(0.145)}$ | 0.333 $_{(0.305)}$ |
| MITRA (+e) | 10.3 | 1055 $_{(+4/-4)}$ | 0.58 | 0.77 | 30.2 | 0.907 $_{(0.143)}$ | 0.860 $_{(0.142)}$ | 0.325 $_{(0.295)}$ |
| Attic (+e) | 10.6 | 1043 $_{(+5/-5)}$ | 0.56 | 0.77 | 31.4 | 0.906 $_{(0.142)}$ | 0.862 $_{(0.141)}$ | 0.328 $_{(0.295)}$ |
| MITRA | 11.1 | 1028 $_{(+4/-5)}$ | 0.54 | 0.74 | 30.8 | 0.905 $_{(0.141)}$ | 0.858 $_{(0.140)}$ | 0.329 $_{(0.295)}$ |
| TabPFNv2 | 11.2 | 1026 $_{(+5/-4)}$ | 0.54 | 0.74 | 30.9 | 0.901 $_{(0.149)}$ | 0.856 $_{(0.144)}$ | 0.327 $_{(0.305)}$ |
| TabICL | 11.2 | 1025 $_{(+4/-4)}$ | 0.54 | 0.74 | 32.3 | 0.903 $_{(0.142)}$ | 0.851 $_{(0.143)}$ | 0.338 $_{(0.303)}$ |
| Attic | 11.6 | 1014 $_{(+5/-4)}$ | 0.52 | 0.73 | 33.8 | 0.901 $_{(0.141)}$ | 0.851 $_{(0.145)}$ | 0.340 $_{(0.297)}$ |
| MITRA 1D (+f) | 12.3 | 990 $_{(+5/-5)}$ | 0.48 | 0.69 | 34.5 | 0.901 $_{(0.139)}$ | 0.850 $_{(0.146)}$ | 0.348 $_{(0.319)}$ |
| CatBoost | 12.7 | 979 $_{(+4/-5)}$ | 0.47 | 0.68 | 35.8 | 0.898 $_{(0.142)}$ | 0.848 $_{(0.145)}$ | 0.352 $_{(0.299)}$ |
| RealMLP | 12.7 | 979 $_{(+5/-5)}$ | 0.47 | 0.66 | 33.2 | 0.895 $_{(0.143)}$ | 0.844 $_{(0.157)}$ | 0.380 $_{(0.402)}$ |
| TabPFNv1 (+e) | 12.7 | 979 $_{(+4/-4)}$ | 0.47 | 0.67 | 35.1 | 0.892 $_{(0.152)}$ | 0.837 $_{(0.158)}$ | 0.367 $_{(0.331)}$ |
| TabForestPFN (+f) | 12.8 | 976 $_{(+5/-5)}$ | 0.47 | 0.66 | 35.4 | 0.895 $_{(0.144)}$ | 0.849 $_{(0.146)}$ | 0.359 $_{(0.327)}$ |
| TabPFNv1 | 13.3 | 960 $_{(+5/-5)}$ | 0.44 | 0.63 | 36.4 | 0.891 $_{(0.153)}$ | 0.836 $_{(0.157)}$ | 0.374 $_{(0.333)}$ |
| MLP | 13.8 | 942 $_{(+5/-5)}$ | 0.42 | 0.59 | 37.7 | 0.892 $_{(0.150)}$ | 0.843 $_{(0.150)}$ | 0.365 $_{(0.314)}$ |
| XGBoost | 14.3 | 929 $_{(+5/-5)}$ | 0.40 | 0.59 | 39.4 | 0.892 $_{(0.143)}$ | 0.841 $_{(0.148)}$ | 0.367 $_{(0.303)}$ |
| MITRA 1D | 14.4 | 924 $_{(+5/-5)}$ | 0.39 | 0.61 | 39.8 | 0.886 $_{(0.151)}$ | 0.834 $_{(0.155)}$ | 0.380 $_{(0.333)}$ |
| Random Forest | 14.7 | 915 $_{(+5/-5)}$ | 0.38 | 0.57 | 43.4 | 0.888 $_{(0.150)}$ | 0.836 $_{(0.151)}$ | 0.458 $_{(0.482)}$ |
| TabForestPFN | 14.7 | 913 $_{(+5/-4)}$ | 0.38 | 0.59 | 39.8 | 0.884 $_{(0.156)}$ | 0.834 $_{(0.159)}$ | 0.384 $_{(0.348)}$ |
| LightGBM | 14.8 | 911 $_{(+5/-5)}$ | 0.37 | 0.55 | 40.4 | 0.879 $_{(0.157)}$ | 0.835 $_{(0.152)}$ | 0.375 $_{(0.308)}$ |

## C.4 Classification

We report detailed results for individual benchmarks in Table 13 (TabRepo), Table 14 (TabZilla), and Table 15 (AMLB), with aggregated results across all three benchmarks in Table 2 of the main text. These tables show that MITRA (+ef) has the best performance across all 3 of these benchmark datasets, which is consistent with the aggregated results.

To further enhance performance, we apply the advanced bagging strategy introduced in Section 4.5 of the main text to both MITRA and the top-performing baseline Attic. Table 16 shows the results

Table 15: **MITRA wins on AMLB** 10-fold classification benchmark. Winner/runner-up in ▨ / ▨. +e means ensembling in ICL, and +f means fine-tuning. The 95% confidence interval is shown in parentheses for the Elo. The columns in the aggregated metrics are mean and std (shown in parentheses) of the corresponding metric.

| Model | Ranking Metrics | | | | | Aggregated Metrics | | |
|---|---|---|---|---|---|---|---|---|
| | Avg. Rank ↓ | Elo ↑ | Winrate ↑ | RAcc ↑ | C Δ ↓ | AUC ↑ | ACC ↑ | CE ↓ |
| MITRA (+ef) | **5.8** | **1202**$_{(+11/-11)}$ | **0.76** | **0.84** | **17.6** | 0.926$_{(0.076)}$ | 0.858$_{(0.124)}$ | 0.341$_{0.292}$ |
| Attic (+ef) | 6.2 | 1186$_{(+10/-10)}$ | 0.74 | 0.83 | 19.4 | 0.926$_{(0.076)}$ | 0.857$_{(0.124)}$ | 0.344$_{(0.293)}$ |
| TabPFNv2 (+e) | 6.9 | 1156$_{(+10/-10)}$ | 0.71 | 0.81 | 19.7 | **0.927**$_{0.0752}$ | **0.858**$_{(0.124)}$ | 0.345$_{(0.298)}$ |
| TabICL (+e) | 7.5 | 1129$_{(+9/-9)}$ | 0.67 | 0.78 | 23.7 | 0.919$_{(0.080)}$ | 0.847$_{(0.125)}$ | 0.360$_{(0.290)}$ |
| TabPFNv2 (+ef) | 7.8 | 1117$_{(+9/-9)}$ | 0.66 | 0.76 | 24.2 | 0.921$_{(0.077)}$ | 0.848$_{(0.133)}$ | 0.368$_{(0.320)}$ |
| TabPFNv2 | 9.4 | 1059$_{(+9/-8)}$ | 0.58 | 0.74 | 27.1 | 0.923$_{(0.077)}$ | 0.852$_{(0.126)}$ | 0.359$_{(0.305)}$ |
| TabICL | 9.4 | 1058$_{(+9/-9)}$ | 0.58 | 0.7 | 29.1 | 0.914$_{(0.087)}$ | 0.841$_{(0.126)}$ | 0.372$_{(0.293)}$ |
| MITRA 1D (+f) | 10.6 | 1014$_{(+8/-9)}$ | 0.52 | 0.66 | 30.9 | 0.921$_{(0.077)}$ | 0.850$_{(0.124)}$ | 0.360$_{(0.294)}$ |
| CatBoost | 10.8 | 1006$_{(+8/-9)}$ | 0.51 | 0.64 | 33.1 | 0.916$_{(0.078)}$ | 0.844$_{(0.126)}$ | 0.376$_{(0.304)}$ |
| TabForestPFN (+f) | 10.8 | 1005$_{(+8/-9)}$ | 0.51 | 0.65 | 30.1 | 0.920$_{(0.078)}$ | 0.848$_{(0.126)}$ | 0.364$_{(0.298)}$ |
| MITRA (+e) | 11.2 | 993$_{(+8/-9)}$ | 0.49 | 0.62 | 36.1 | 0.911$_{(0.085)}$ | 0.831$_{(0.147)}$ | 0.406$_{(0.350)}$ |
| Attic (+e) | 11.3 | 986$_{(+9/-9)}$ | 0.48 | 0.63 | 34.9 | 0.910$_{(0.085)}$ | 0.832$_{(0.147)}$ | 0.401$_{(0.348)}$ |
| MITRA | 12.6 | 942$_{(+9/-9)}$ | 0.42 | 0.54 | 39.0 | 0.909$_{(0.087)}$ | 0.826$_{(0.147)}$ | 0.415$_{(0.354)}$ |
| RealMLP | 12.6 | 940$_{(+9/-10)}$ | 0.42 | 0.58 | 35.0 | 0.911$_{(0.082)}$ | 0.834$_{(0.138)}$ | 0.406$_{(0.330)}$ |
| Attic | 12.7 | 935$_{(+9/-9)}$ | 0.41 | 0.58 | 38.1 | 0.908$_{(0.086)}$ | 0.829$_{(0.148)}$ | 0.405$_{(0.353)}$ |
| XGBoost | 13.1 | 924$_{(+9/-10)}$ | 0.4 | 0.54 | 38.2 | 0.912$_{(0.080)}$ | 0.839$_{(0.127)}$ | 0.388$_{(0.308)}$ |
| LightGBM | 13.4 | 912$_{(+9/-9)}$ | 0.38 | 0.51 | 38.0 | 0.910$_{(0.082)}$ | 0.838$_{(0.132)}$ | 0.389$_{(0.314)}$ |
| Random Forest | 13.6 | 905$_{(+9/-9)}$ | 0.37 | 0.5 | 42.3 | 0.908$_{(0.087)}$ | 0.833$_{(0.131)}$ | 0.446$_{(0.330)}$ |
| MLP | 14.5 | 869$_{(+10/-10)}$ | 0.33 | 0.46 | 39.9 | 0.897$_{(0.097)}$ | 0.820$_{(0.146)}$ | 0.424$_{(0.340)}$ |
| MITRA 1D | 15.3 | 834$_{(+10/-10)}$ | 0.28 | 0.44 | 45.1 | 0.899$_{(0.090)}$ | 0.818$_{(0.147)}$ | 0.423$_{(0.347)}$ |
| TabForestPFN | 15.5 | 828$_{(+9/-9)}$ | 0.28 | 0.43 | 44.4 | 0.899$_{(0.089)}$ | 0.817$_{(0.146)}$ | 0.423$_{(0.344)}$ |

of MITRA (bagging) and Attic (bagging) evaluated on the unified set of the three classification benchmark datasets. Notably, MITRA (bagging) further improves MITRA (+ef) and shows even larger gains compared to other baselines.

We additionally evaluate MITRA and baselines on the TabArena benchmark [8] in Table 17. MITRA remains the state-of-the-art tabular models on TabArena. Specifically, MITRA achieves Pareto efficiency both in training time and inference time, with TabPFNv2 (HPO + ensemble) being significantly slower. MITRA is the strongest single model, outperforming all methods even when they perform hyperparameter tuning for 200 iterations. MITRA is only outperformed once the hyperparameter configurations of TabPFNv2 are ensembled together. We leave constructing a search space for MITRA for HPO and HPO + ensemble results as future work.

## C.5 Regression

In Table 18, we report additional regression results on a larger benchmark that combines AMLB and OpenML-CTR23 with features up to 500 and rows up to 10k. This benchmark contains more datasets with a larger number of rows than the other benchmark datasets. Results show that MITRA (+ef) and TabPFNV2 (+ef) have similar performances, with TabPFNv2 (+e) having the top performance. We choose to separate the two regression benchmarks (Table 4 and Table 18) to show differences in performance on these small-scale and large-scale regression datasets. These results show that MITRA performs better on small-scale datasets. Its performance limitation on this larger-scale dataset can be explained by the fact that in pretraining it only sees up to 16 features and up to 640 rows. Notably, it can outperform TabPFNv2 on benchmarks with up to 100 features and 3k rows, and on some benchmarks with up to 500 features and 10k rows, despite being pretrained on one-tenth of the maximum pretraining features and one-third of the maximum pretraining rows in TabPFNv2. These findings suggest that MITRA generalizes well beyond its pretraining regime. Future work includes increasing the maximum number of rows and features during the pretraining process.

## C.6 Critical Differences

We visualize the critical differences between MITRA and baselines across three classification benchmarks (TabRepo in Figure 9, TabZilla in Figure 10 and AMLB in Figure 11) and two regression benchmarks (TabRepo in Figure 12 and AMLB + OpenML-CTR23 in Figure 13).

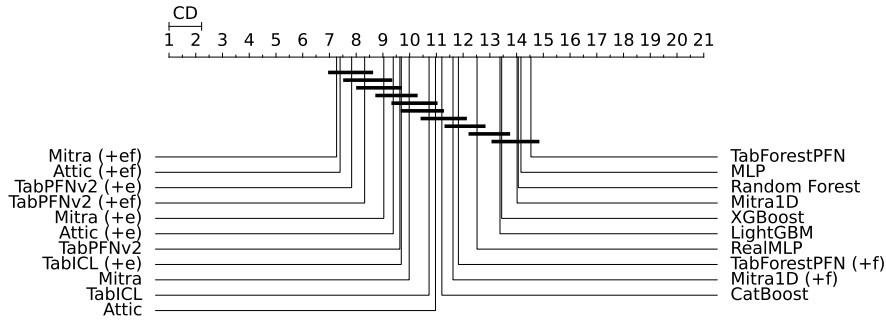

Figure 9: Critical difference plot on TabRepo classification benchmark.

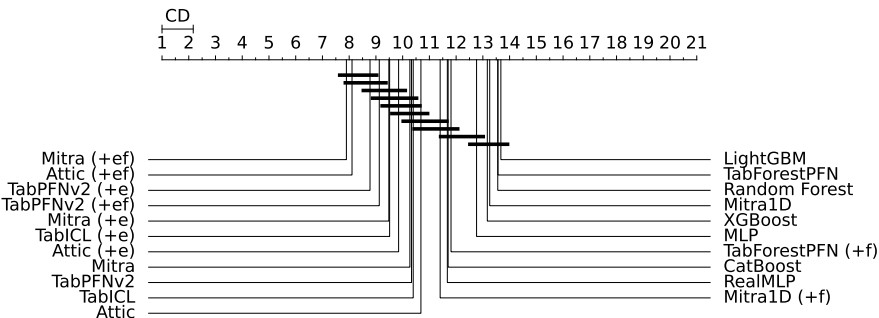

Figure 10: Critical difference plot on TabZilla classification benchmark.

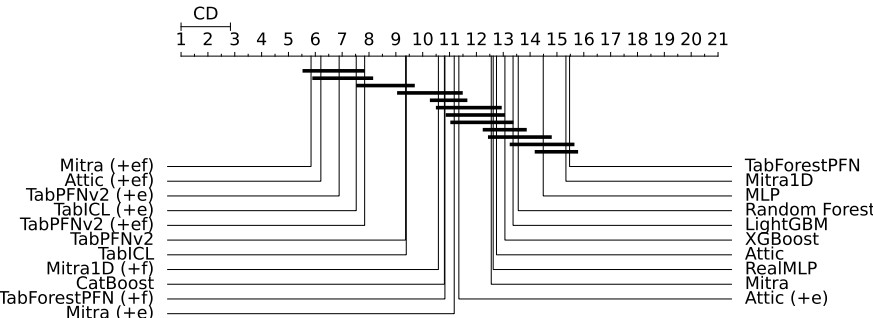

Figure 11: Critical difference plot on AMLB classification benchmark.

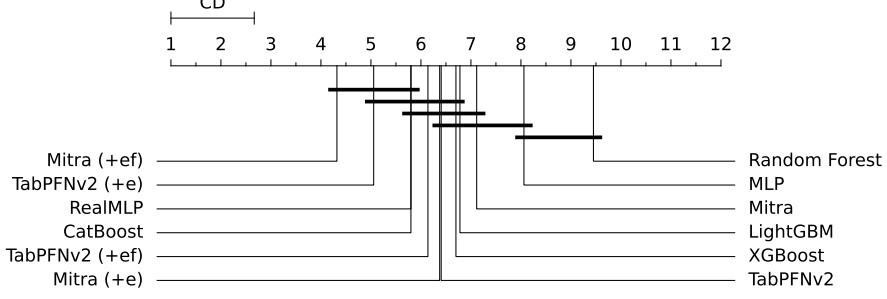

Figure 12: Critical difference plot on TabRepo regression benchmark.

Table 16: Adding MITRA (bagging) and Attic (bagging) in the aggregated classification benchmark results in Table 2 of the main text. +e means adding ensemble in ICL, and +f means adding fine-tuning. The 95% confidence interval is shown in parentheses for the Elo. The columns in the aggregated metrics are mean and std (shown in parentheses) of the corresponding metric.

| Model | Ranking Metrics | | | | | Aggregated Metrics | | |
|---|---|---|---|---|---|---|---|---|
| | Avg. Rank ↓ | Elo ↑ | Winrate ↑ | RAcc ↑ | C Δ ↓ | AUC ↑ | ACC ↑ | CE ↓ |
| **MITRA** (bagging) | **7.9** | **1135** $_{(+3/-4)}$ | **0.68** | **0.82** | 20.8 | **0.905** $_{(0.125)}$ | **0.86** $_{(0.144)}$ | **0.325** $_{(0.318)}$ |
| MITRA (+ef) | 8.2 | 1124 $_{(+4/-4)}$ | 0.67 | 0.82 | 20.7 | 0.904 $_{(0.124)}$ | 0.858 $_{(0.144)}$ | 0.327 $_{(0.318)}$ |
| Attic (bagging) | 8.4 | 1119 $_{(+4/-4)}$ | 0.66 | 0.81 | 22.9 | 0.9 $_{(0.131)}$ | 0.855 $_{(0.144)}$ | 0.333 $_{(0.319)}$ |
| Attic (+ef) | 8.5 | 1115 $_{(+4/-4)}$ | 0.66 | 0.81 | 22.3 | 0.903 $_{(0.125)}$ | 0.858 $_{(0.143)}$ | 0.331 $_{(0.318)}$ |
| TabPFNv2 (+e) | 9.1 | 1094 $_{(+4/-3)}$ | 0.63 | 0.79 | 23.8 | 0.901 $_{(0.13)}$ | 0.856 $_{(0.144)}$ | 0.337 $_{(0.319)}$ |
| TabPFNv2 (+ef) | 9.7 | 1073 $_{(+4/-3)}$ | 0.6 | 0.76 | 25.8 | 0.897 $_{(0.129)}$ | 0.846 $_{(0.151)}$ | 0.362 $_{(0.342)}$ |
| TabICL (+e) | 10.7 | 1041 $_{(+4/-4)}$ | 0.56 | 0.74 | 31.2 | 0.888 $_{(0.14)}$ | 0.836 $_{(0.15)}$ | 0.366 $_{(0.324)}$ |
| MITRA (+e) | 11.0 | 1032 $_{(+4/-4)}$ | 0.55 | 0.73 | 31.7 | 0.896 $_{(0.132)}$ | 0.847 $_{(0.148)}$ | 0.359 $_{(0.328)}$ |
| TabPFNv2 | 11.1 | 1030 $_{(+4/-3)}$ | 0.54 | 0.73 | 29.6 | 0.891 $_{(0.139)}$ | 0.846 $_{(0.147)}$ | 0.352 $_{(0.324)}$ |
| Attic (+e) | 11.3 | 1023 $_{(+3/-3)}$ | 0.53 | 0.73 | 32.2 | 0.896 $_{(0.131)}$ | 0.848 $_{(0.148)}$ | 0.363 $_{(0.328)}$ |
| TabICL | 11.9 | 1004 $_{(+3/-3)}$ | 0.51 | 0.7 | 33.9 | 0.884 $_{(0.141)}$ | 0.833 $_{(0.152)}$ | 0.373 $_{(0.324)}$ |
| MITRA | 12.0 | 1001 $_{(+4/-3)}$ | 0.5 | 0.68 | 33.5 | 0.891 $_{(0.134)}$ | 0.841 $_{(0.151)}$ | 0.368 $_{(0.331)}$ |
| MITRA 1D (+f) | 12.6 | 979 $_{(+3/-4)}$ | 0.47 | 0.67 | 34.9 | 0.892 $_{(0.13)}$ | 0.843 $_{(0.15)}$ | 0.365 $_{(0.331)}$ |
| Attic | 12.7 | 979 $_{(+4/-4)}$ | 0.47 | 0.67 | 35.7 | 0.884 $_{(0.139)}$ | 0.834 $_{(0.156)}$ | 0.375 $_{(0.332)}$ |
| CatBoost | 12.8 | 976 $_{(+4/-4)}$ | 0.47 | 0.67 | 36.0 | 0.888 $_{(0.133)}$ | 0.837 $_{(0.15)}$ | 0.374 $_{(0.324)}$ |
| TabForestPFN (+f) | 13.0 | 969 $_{(+4/-4)}$ | 0.46 | 0.65 | 35.6 | 0.886 $_{(0.136)}$ | 0.84 $_{(0.148)}$ | 0.375 $_{(0.33)}$ |
| RealMLP | 13.6 | 948 $_{(+3/-3)}$ | 0.43 | 0.62 | 35.7 | 0.878 $_{(0.142)}$ | 0.827 $_{(0.164)}$ | 0.411 $_{(0.394)}$ |
| XGBoost | 14.6 | 914 $_{(+3/-3)}$ | 0.38 | 0.58 | 39.8 | 0.883 $_{(0.133)}$ | 0.833 $_{(0.149)}$ | 0.388 $_{(0.323)}$ |
| LightGBM | 14.9 | 905 $_{(+4/-4)}$ | 0.37 | 0.56 | 40.1 | 0.876 $_{(0.141)}$ | 0.829 $_{(0.153)}$ | 0.392 $_{(0.328)}$ |
| MLP | 15.3 | 893 $_{(+4/-4)}$ | 0.35 | 0.51 | 40.4 | 0.869 $_{(0.145)}$ | 0.82 $_{(0.161)}$ | 0.413 $_{(0.346)}$ |
| Random Forest | 15.3 | 892 $_{(+4/-4)}$ | 0.35 | 0.54 | 44.5 | 0.874 $_{(0.14)}$ | 0.822 $_{(0.153)}$ | 0.47 $_{(0.424)}$ |
| MITRA 1D | 15.6 | 882 $_{(+4/-4)}$ | 0.34 | 0.54 | 42.8 | 0.869 $_{(0.143)}$ | 0.815 $_{(0.163)}$ | 0.414 $_{(0.351)}$ |
| TabForestPFN | 15.8 | 873 $_{(+4/-4)}$ | 0.33 | 0.52 | 42.8 | 0.864 $_{(0.154)}$ | 0.814 $_{(0.167)}$ | 0.414 $_{(0.353)}$ |

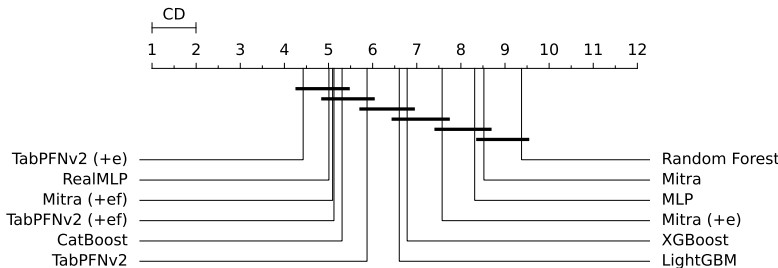

Figure 13: Critical difference plot on AMLB + OpenML-CTR23 regression benchmark.

## C.7 Timing Efficiency

We compute the running time of MITRA and baselines on eight 40GB A100 machines. In Figure 14, we present the performance metrics (Elo, winrate, and AUC) alongside the average running time on the TabRepo benchmark. MITRA (+ef) achieves a gain of 138 Elo over CatBoost while being approximately 3.5× faster. We have additionally measured single-GPU fine-tuning performance. MITRA (+ef) on a single GPU takes similar time (89 seconds) as CatBoost (83 seconds) while achieving a gain of 138 Elo. We present more details on training and inference times on the TabArena benchmark in Table 17.

## C.8 Decision Boundary Visualizations

We visualize the decision boundaries of MITRA and baseline methods on a set of representative 2D simulated datasets. Each dataset consists of 1,000 samples drawn from a known ground-truth distribution, with 10% used as support samples and the remaining 90% as query samples. For MITRA and TabPFNv2, we adopt the ICL setting without ensembling. For classical models, we use their default hyperparameters. Overall, MITRA demonstrates effective few-shot generalization

Table 17: Additional results on TabArena. "Single" represents single-model results without HPO. "HPO" represents results after 200-iteration hyperparameter optimization. "HPO + ensemble" represents the ensemble results of hyperparameter optimization. MITRA is the strongest single model, outperforming all methods even when they perform hyperparameter tuning. "Train Time" and "Infer Time" represent the median training and inference time in seconds per 1K rows. MITRA achieves Pareto efficiency both in training time and inference time.

| Model | Avg. Rank ↓ | Elo ↑ | Winrate ↑ | C Δ ↓ | Train Time | Infer Time |
|---|---|---|---|---|---|---|
| TabPFNv2 (HPO + ensemble) | **6.2** | **1745** | **0.89** | **0.05** | 3445.6 | 48.2 |
| **Mitra (single)** | 7.4 | 1699 | 0.86 | 0.07 | 457.2 | 34.6 |
| TabM (HPO + ensemble) | 9.9 | 1620 | 0.8 | 0.1 | 2828.4 | 1.6 |
| TabICL (single) | 10.1 | 1617 | 0.8 | 0.07 | 8.9 | 1.7 |
| RealMLP (HPO + ensemble) | 10.9 | 1597 | 0.78 | 0.09 | 6796.3 | 12.4 |
| TabPFNv2 (HPO) | 10.9 | 1596 | 0.78 | 0.08 | 3445.6 | 1 |
| AutoGluon1.3 (4h) | 12.6 | 1551 | 0.74 | 0.1 | 2309.2 | 2.6 |
| TabPFNv2 (single) | 12.7 | 1548 | 0.74 | 0.1 | 4.1 | 0.4 |
| LightGBM (HPO + ensemble) | 13.7 | 1525 | 0.72 | 0.12 | 647.6 | 1.7 |
| TabM (HPO) | 14.2 | 1512 | 0.71 | 0.11 | 2828.4 | 0.2 |
| LightGBM (HPO) | 16.2 | 1470 | 0.66 | 0.12 | 647.6 | 0.3 |
| CatBoost (HPO + ensemble) | 16.5 | 1465 | 0.66 | 0.12 | 1465.9 | 0.7 |
| CatBoost (HPO) | 17.3 | 1444 | 0.64 | 0.12 | 1465.9 | 0.1 |
| TabM (single) | 17.8 | 1437 | 0.63 | 0.14 | 10.4 | 0.2 |
| CatBoost (single) | 17.8 | 1434 | 0.63 | 0.14 | 5.7 | 0.1 |
| ModernNCA (HPO) | 18 | 1428 | 0.62 | 0.12 | 5944.9 | 0.5 |
| XGBoost (HPO + ensemble) | 18.5 | 1420 | 0.61 | 0.13 | 766.1 | 1.9 |
| EBM (HPO + ensemble) | 20 | 1390 | 0.58 | 0.16 | 1109.1 | 0.2 |
| XGBoost (HPO) | 20.1 | 1386 | 0.58 | 0.14 | 766.1 | 0.3 |
| RealMLP (HPO) | 20.2 | 1383 | 0.57 | 0.13 | 6796.3 | 0.7 |
| ModernNCA (HPO + ensemble) | 20.4 | 1381 | 0.57 | 0.13 | 5944.9 | 8.4 |
| ModernNCA (single) | 20.8 | 1368 | 0.56 | 0.15 | 14.8 | 0.3 |
| TorchMLP (HPO + ensemble) | 20.9 | 1370 | 0.56 | 0.14 | 2862.1 | 2.2 |
| FastaiMLP (HPO + ensemble) | 21.1 | 1366 | 0.55 | 0.16 | 1358.6 | 8.1 |
| TabDPT (single) | 22.7 | 1329 | 0.52 | 0.15 | 27.5 | 8.9 |
| EBM (HPO) | 23.1 | 1323 | 0.51 | 0.17 | 1109.1 | 0 |
| EBM (single) | 23.8 | 1307 | 0.49 | 0.18 | 5.3 | 0.1 |
| RealMLP (single) | 25.6 | 1270 | 0.45 | 0.16 | 22.5 | 1.6 |
| FastaiMLP (HPO) | 25.6 | 1273 | 0.45 | 0.17 | 1358.6 | 0.9 |
| ExtraTrees (HPO + ensemble) | 26.1 | 1261 | 0.44 | 0.18 | 370.9 | 1.5 |
| TorchMLP (HPO) | 27 | 1241 | 0.42 | 0.16 | 2862.1 | 0.2 |
| XGBoost (single) | 28.2 | 1214 | 0.39 | 0.17 | 2.4 | 0.2 |
| ExtraTrees (HPO) | 28.6 | 1205 | 0.39 | 0.2 | 370.9 | 0.2 |
| RandomForest (HPO + ensemble) | 30.5 | 1159 | 0.34 | 0.2 | 527.4 | 1.4 |
| LightGBM (single) | 30.7 | 1159 | 0.34 | 0.18 | 2.9 | 0.1 |
| RandomForest (HPO) | 33.1 | 1094 | 0.29 | 0.21 | 527.4 | 0.1 |
| TorchMLP (single) | 33.4 | 1087 | 0.28 | 0.22 | 10.4 | 0.2 |
| FastaiMLP (single) | 34.6 | 1056 | 0.25 | 0.23 | 4.7 | 0.6 |
| Linear (HPO + ensemble) | 35.2 | 1032 | 0.24 | 0.29 | 88.6 | 0.3 |
| Linear (HPO) | 36.2 | 1005 | 0.22 | 0.29 | 88.6 | 0.1 |
| RandomForest (single) | 36.3 | 1000 | 0.21 | 0.26 | 0.4 | 0.1 |
| Linear (single) | 36.8 | 985 | 0.21 | 0.31 | 2.3 | 0.1 |
| ExtraTrees (single) | 37.7 | 952 | 0.19 | 0.28 | 0.4 | 0.1 |
| KNN (HPO + ensemble) | 42.6 | 716 | 0.08 | 0.48 | 3 | 0.2 |
| KNN (HPO) | 43.7 | 623 | 0.05 | 0.5 | 3 | 0 |
| KNN (single) | 45.2 | 415 | 0.02 | 0.59 | 0.1 | 0 |

capabilities.[3] As illustrated in Figure 15 and Figure 21, when the data distribution is axis-aligned, MITRA produces more regular and less fragmented decision boundaries than TabPFNv2. This result suggests that a lower functional complexity that appears to support better generalization. On other representative 2D datasets—GP data (Figure 16), linearly separable data (Figure 18), Gaussian mixtures (Figure 19), sine waves (Figure 20), and star-shaped distributions (Figure 23)—MITRA has

---

[3]MITRA achieves comparable or superior generalization performance to TabPFNv2 on most datasets. One notable exception arises on the spiral dataset (Figure 22). However, we observe MITRA demonstrates greater robustness to increasing noise levels in this simulated data. In contrast, TabPFNv2 exhibits a pronounced performance drop, resembling a phase transition, as noise level increases.

Table 18: AMLB 10 fold regression benchmark. +e means adding ensemble in ICL, and +f means adding fine-tuning. The 95% confidence interval is shown in parentheses for the Elo. The columns in the aggregated metrics are mean and std (shown in parentheses) of the corresponding metric.

| Model | Ranking Metrics | | | | | Aggregated Metrics | | |
|---|---|---|---|---|---|---|---|---|
| | Avg. Rank ↓ | Elo ↑ | Winrate ↑ | RAcc ↑ | C Δ ↓ | $R^2$ ↑ | RMSE ↓ | MAE ↓ |
| **TabPFNv2 (+e)** | **4.4** | **1137**$_{(+13/-12)}$ | **0.69** | **0.82** | **14.8** | 0.683 $_{(0.324)}$ | **1720.58** $_{(5869.98)}$ | 1002.44 $_{(3340.38)}$ |
| RealMLP | 5 | 1097$_{(+12/-13)}$ | 0.64 | 0.77 | 16.4 | **0.685**$_{(0.317)}$ | 1743.95 $_{(5919.86)}$ | **991.60**$_{(3310.13)}$ |
| MITRA (+ef) | 5.1 | 1091$_{(+12/-11)}$ | 0.63 | 0.79 | 18.3 | 0.678$_{(0.322)}$ | 1773.96$_{(6086.74)}$ | 1064.63$_{(3520.63)}$ |
| TabPFNv2 (+ef) | 5.1 | 1089$_{(+12/-11)}$ | 0.63 | 0.77 | 16.7 | 0.672$_{(0.328)}$ | 1749.62$_{(5972.03)}$ | 1017.13$_{(3337.77)}$ |
| CatBoost | 5.3 | 1077$_{(+11/-12)}$ | 0.61 | 0.75 | 19.1 | 0.678$_{(0.315)}$ | 1773.29$_{(6051.57)}$ | 1083.56$_{(3611.41)}$ |
| TabPFNv2 | 5.9 | 1041$_{(+12/-12)}$ | 0.56 | 0.69 | 21.2 | 0.669$_{(0.333)}$ | 1889.36$_{(6334.77)}$ | 1140.71$_{(3827.81)}$ |
| LightGBM | 6.6 | 995$_{(+12/-12)}$ | 0.49 | 0.65 | 22.3 | 0.676$_{(0.311)}$ | 1827.53$_{(6250.13)}$ | 1140.90$_{(3827.07)}$ |
| XGBoost | 6.8 | 984$_{(+11/-12)}$ | 0.47 | 0.65 | 21.7 | 0.670$_{(0.314)}$ | 1804.76$_{(6167.97)}$ | 1121.92$_{(3770.51)}$ |
| MITRA (+e) | 7.6 | 933$_{(+12/-12)}$ | 0.4 | 0.59 | 28.8 | 0.650$_{(0.332)}$ | 1964.38$_{(6586.97)}$ | 1263.88$_{(4148.94)}$ |
| MLP | 8.3 | 883$_{(+13/-13)}$ | 0.34 | 0.47 | 28.1 | 0.638$_{(0.362)}$ | 2153.05$_{(7161.28)}$ | 1105.91$_{(3649.86)}$ |
| MITRA | 8.5 | 869$_{(+12/-12)}$ | 0.32 | 0.51 | 31.6 | 0.641$_{(0.337)}$ | 1992.53$_{(6648.48)}$ | 1294.52$_{(4233.03)}$ |
| Random Forest | 9.4 | 805$_{(+13/-14)}$ | 0.24 | 0.37 | 34.9 | 0.639$_{(0.325)}$ | 1960.06$_{(6652.34)}$ | 1233.41$_{(4145.83)}$ |

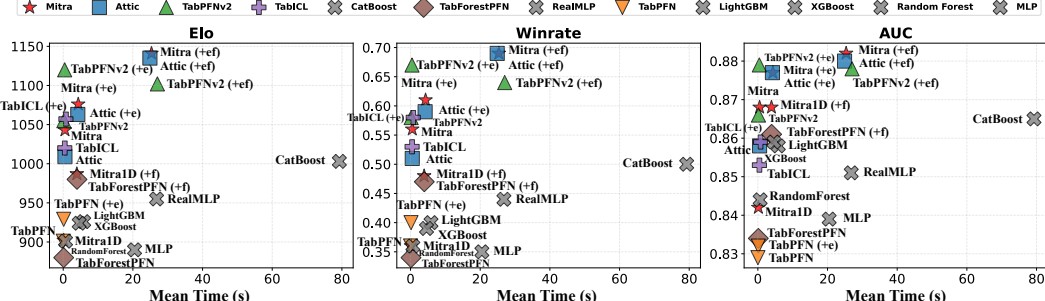

Figure 14: Average running time vs. performance metrics (Elo, winrate, and AUC) on TabRepo.

decision boundaries that fall between those of tree-based classifiers and the TabPFNv2 model, which highlights the effect of pre-training on a mixture of synthetic priors. In the spiral (Figure 22) and Swiss roll (Figure 24) examples, the Gaussian Process (GP) classifier shows strong performance, which motivates our future work to incorporate GP-based priors into the pretraining mixture.

# D  Limitations and Future Work

While our current mixture of priors demonstrates strong performance, it can be further improved by employing hyperparameter optimization (HPO) to adapt the mixture weights for specific downstream tasks or domains. In addition, we plan to incorporate other continuous priors, e.g., Gaussian Processes, which model smooth boundaries directly into the mixture, to better generalize to tasks outside of tabular domain, e.g., time series forecasting. Lastly, although MITRA achieves competitive results overall, it does not consistently outperform TabPFNv2 on large-feature regression tasks, and we plan to scale pretraining to datasets with larger numbers of rows and features to yield further gains in generalization to real-world, high-dimensional settings.

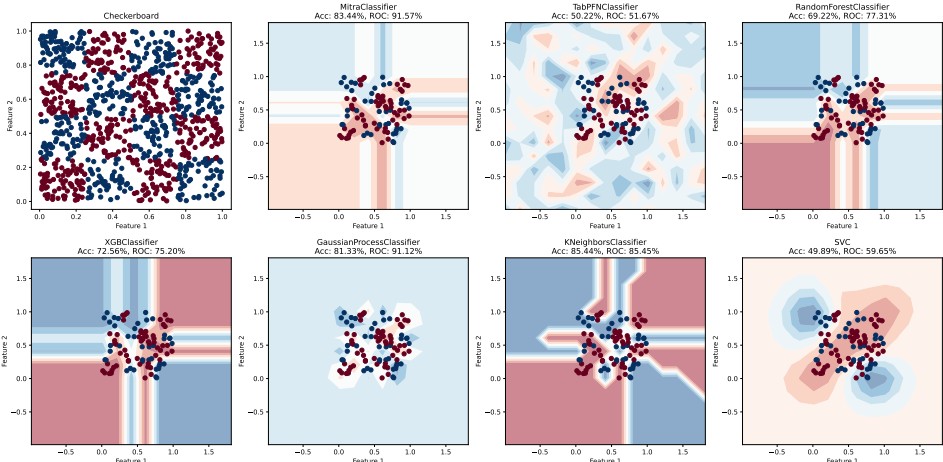

Figure 15: Decision boundaries of MITRA and baselines on 2D checkerboard data. MITRA shows more regular and less fragmented decision boundaries than TabPFNv2.

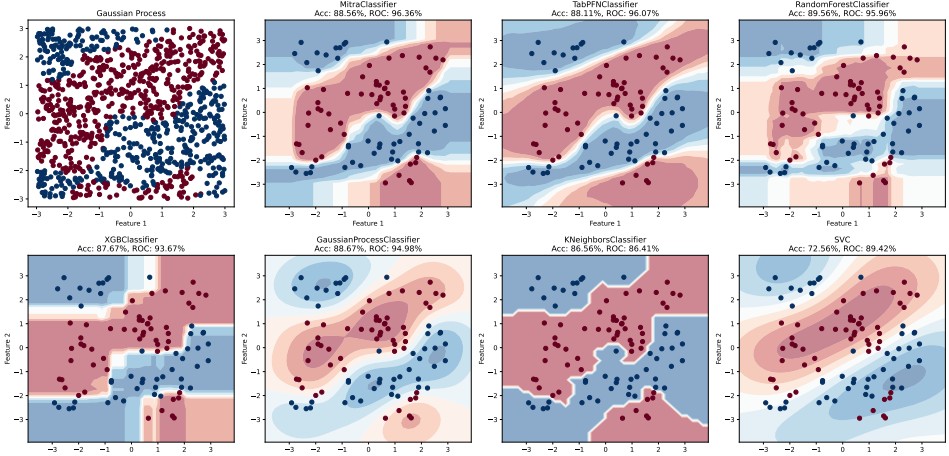

Figure 16: Decision boundaries of MITRA and baselines on 2D Gaussian Process data.

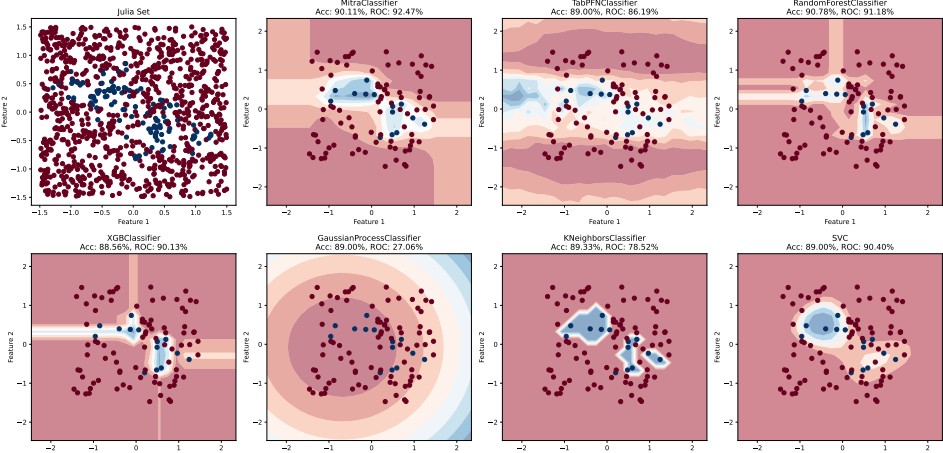

Figure 17: Decision boundaries of MITRA and baselines on 2D Julia set data.

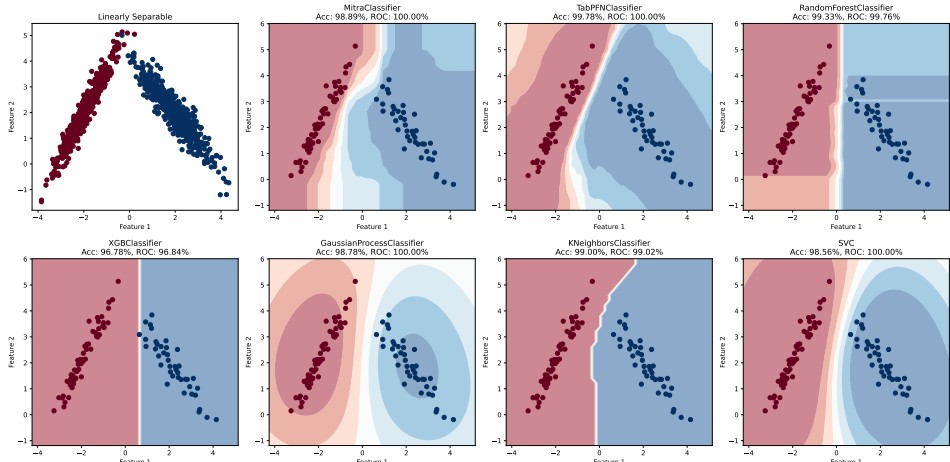

Figure 18: Decision boundaries of MITRA and baselines on 2D linearly separable data.

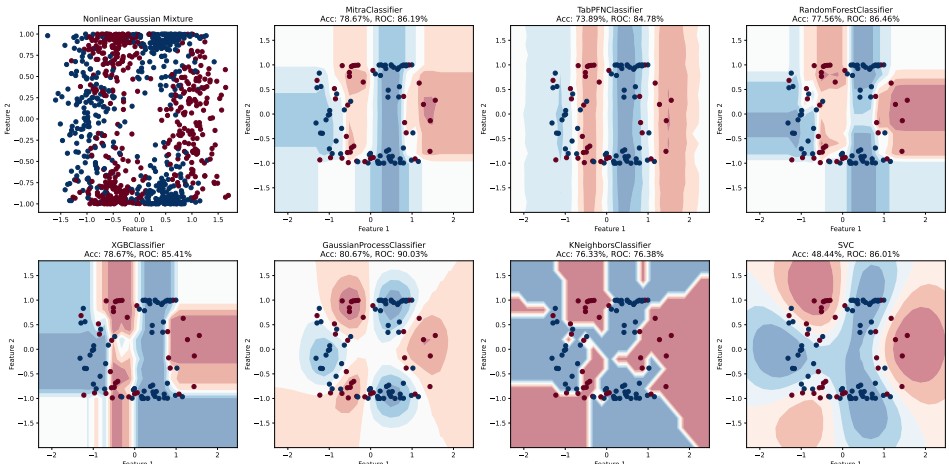

Figure 19: Decision boundaries of MITRA and baselines on 2D nonlinear Gaussian mixture data.

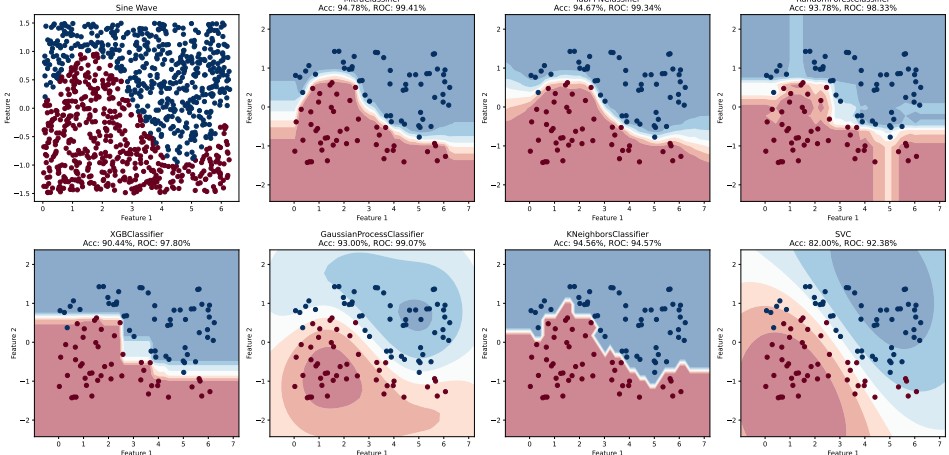

Figure 20: Decision boundaries of MITRA and baselines on 2D sine wave data.

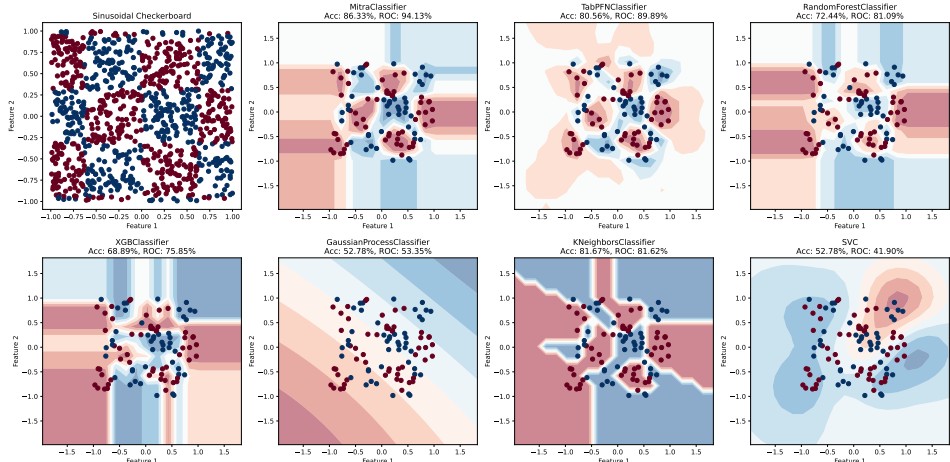

Figure 21: Decision boundaries of MITRA and baselines on 2D sinusoidal checkerboard data. MITRA shows more regular and less fragmented decision boundaries than TabPFNv2.

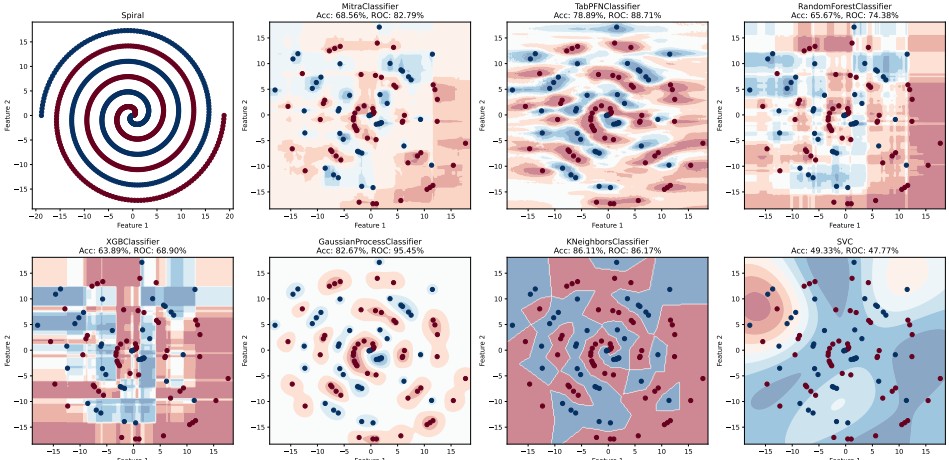

Figure 22: Decision boundaries of MITRA and baselines on 2D spiral data.

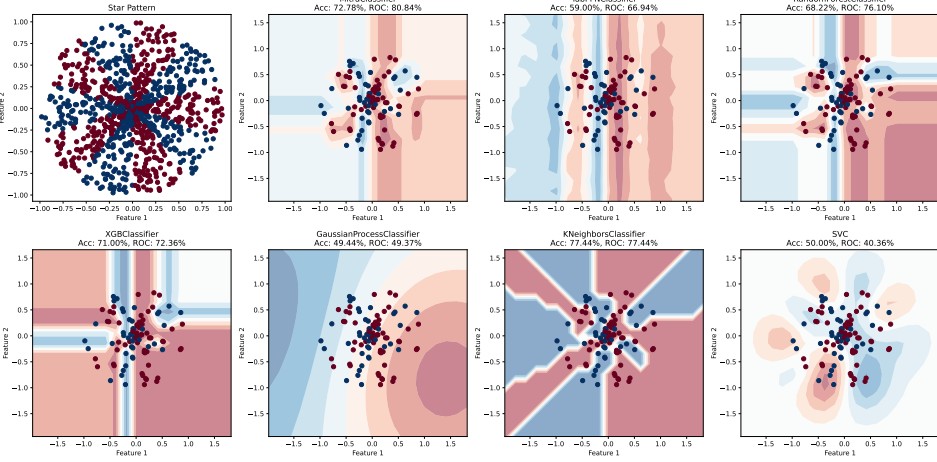

Figure 23: Decision boundaries of MITRA and baselines on 2D star data.

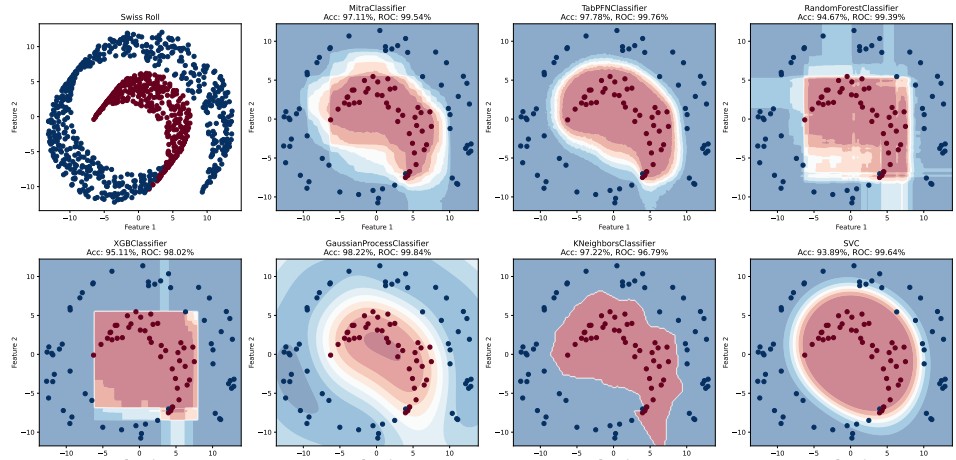

Figure 24: Decision boundaries of MITRA and baselines on 2D Swiss roll data (from [27]).

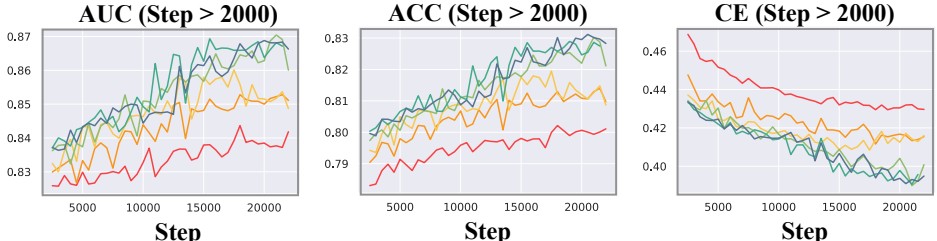

Figure 25: Large view of Figure 5, showing only the zoomed-in region for greater clarity.

