# OpenReview forum: "Mitra: Mixed Synthetic Priors for Enhancing Tabular Foundation Models"
_NeurIPS.cc/2025/Conference — NeurIPS 2025 poster_

### Official Review · Reviewer_EUuF · 2025-06-30

**Clarity:** 3
**Significance:** 3
**Originality:** 2
**Rating:** 5
**Confidence:** 5

**Summary:**

The authors propose Mitra, a mixture of priors to train tabular foundational models. The authors provide detailed analysis/ablations on the different components of the mixture, how to pick them and their impact. The proposed mixture of priors yields state-of-the-art results across three diverse benchmarks when used to train tabular foundation models.

**Questions:**

- Could the authors provide the code (I could not find it)
- The authors mention that they sample 50% from the SCM prior and 50% from the TBP in Line 192. This is a bit vague, how is the 50% spread out for the TBP components?
- The authors mention that a reason why Mitra benefits more from fine-tuning, is because it was trained with synthetic datasets with up to 16 features compared to TabPFNv2 which was trained with 100. I am curious on the reasoning behind the choice of training with 16 features and not 100?
- Was HPO performed for the baselines? It would be great if TabM was included in the experimental protocol, it is a recent work which claims SOTA results in the tabular domain.

**Ethical Concerns:**

["NO or VERY MINOR ethics concerns only"]

**Final Justification:**

The authors addressed the majority of my concerns during the rebuttal.

- The authors proposed to amend the incorrect information regarding the runtime of the method compared to the baselines. Additionally, the authors provided training and inference times for the methods considered.
- Lastly, the authors provided information regarding fine-tuning on a larger number of features proving that it does not yield higher performance compared to prior works.

**Limitations:**

The limitations are not part of the main paper.

**Paper Formatting Concerns:**

I did not notice any formatting issues.

**Quality:**

3

**Strengths And Weaknesses:**

**Strengths**:
- The paper is well written.
- The authors consider diverse well-established benchmarks.
- The proposed work achieves state-of-the-art results.

**Weaknesses**:
- The related work section can be improved, in particular the deep learning aspect of tabular methods. [1][2][3][4]
- The code is not provided (with what I observed).
- The novelty is limited given that prior works do use tree-based priors (TabForest, TabICL)
- The authors provide a list of criteria on the different component selection of the mixture of priors, however, the results paint a slightly different picture (the RF prior fits the Diversity and Distinctive attributes before the ET prior, yet the performance does not improve substantially.)
- The proposed work does not outperform the current state-of-the-art without fine-tuning. The main idea/motivation of TabFM is not training models on downstream tasks, while the authors do go back to fine-tuning. It would be great to point out the time it takes to fine-tune, although the comparison with AutoGluon and TabPFNv2 in Figure 3, does give an idea.

**Typos:**

- Line 227: all three datasets -> all three benchmarks

**References**:

[1] Somepalli, G., Goldblum, M., Schwarzschild, A., Bruss, C. B., & Goldstein, T. (2021). Saint: Improved neural networks for tabular data via row attention and contrastive pre-training. arXiv preprint arXiv:2106.01342.

[2]  Kadra, A., Lindauer, M., Hutter, F., & Grabocka, J. (2021). Well-tuned simple nets excel on tabular datasets. Advances in neural information processing systems, 34, 23928-23941.

[3] Gorishniy, Y., Rubachev, I., Khrulkov, V., & Babenko, A. (2021). Revisiting deep learning models for tabular data. Advances in neural information processing systems, 34, 18932-18943.

[4] Gorishniy, Y., Kotelnikov, A., & Babenko, A. TabM: Advancing tabular deep learning with parameter-efficient ensembling. In The Thirteenth International Conference on Learning Representations.

---

> ### Author Rebuttal · Authors · 2025-07-31
>
> We sincerely thank the reviewer for the thoughtful and constructive feedback. We appreciate that the reviewer finds (1) our benchmarks **diverse**, (2) our results achieve **state-of-the-art** performance, and (3) our paper **well written**. We address the reviewer’s concerns as follows.
>
> > Novelty.
>
> We acknowledge that prior works such as TabForest and TabICL have used tree-based priors. However, these works adopt tree-based priors in a heuristic manner, without explaining why incorporating these priors are beneficial. We are the first to systematically study what makes a mixture of priors effective and identify three key factors (performance, diversity, and distinctiveness) for better generalization. In Table 1 and Table 6, we show that following our principled analysis, our proposed mixture of priors performs better than the priors used in TabForestPFN (SCM + DT) and TabICL (SCM + GB), which only contain 1 tree-based method.
>
> > On fine-tuning.
>
> We thank the reviewer for the thoughtful feedback. We acknowledge that one motivation for TabFMs is to introduce a powerful ICL model, but it is not the only motivation or benefit. While ICL performance is crucial, a truly foundation model for practical use must also be an excellent starting point for efficient adaption. Our central goal is to build a foundation optimized for maximum adaptability, providing the most practical path to state-of-the-art performance with efficient adaptation. Mitra’s ICL performance is comparable to TabICL (concurrent work) as shown in Table 2, and by fine-tuning from a strong and efficiently adaptable foundation, Mitra ultimately achieves the best performance.
>
> 1. **Performance gap re: ICL vs ensemble + fine-tuning**: We acknowledge the performance gap between ICL and ensemble. We attribute this to (a) Mitra’s pretraining uses fewer features, and (b) TabPFNv2’s use of comprehensive feature preprocessing such as quantile transform and SVD that diversifies its ensemble members. Preliminary attempts to naively increase Mitra’s pretraining features or adopt TabPFNv2’s preprocessing did not yield gains out-of-the-box, highlighting a complex interplay between prior mixture, architecture, and training dynamics. We leave a systematic exploration of this as future work. Note that our primary focus is getting the best performance with minimal adaptation, to that end, Mitra sets a new state-of-the-art with fast fine-tuning, significantly outperforming TabPFNv2 (+e/+ef) by **~30-50 Elo** in Table 2 with std only +4/-4 and **~100** over the top ICL performance.
> 2. **Mitra challenges SOTA in ICL mode with fewer pretrained features**:
>     1. Mitra was pre-trained with <= 16 features for pretraining computational efficiency, whereas TabPFNv2 was pre-trained with <= 160 features. Despite this discrepancy, Mitra achieves ICL performance comparable to TabPFNv2 on downstream datasets with <= 100 features. This demonstrates the strong **generalizability** and sample efficiency of our prior mixture, allowing the model to learn more robust patterns that generalize better to unseen, larger feature spaces.
>     2. To further validate that the performance gap is not an effect of our prior mixture, we evaluate Mitra and TabPFNv2 ICL on small datasets with <= 16 features and <= 2k rows. **On these datasets, Mitra shows better ICL performance than TabPFNv2**.
>     3. Mitra also has better **sample efficiency** as demonstrated in Section 4.4 and Table 5, making it highly effective in real-world scenarios with limited labeled data and an overall best choice across both low- and high-label regimes.
>
> |Model|AUC|ACC|CE|
> |-|-|-|-|
> |Mitra(+ef)|0.851±0.137|0.805±0.184|0.433±0.387|
> |TabPFNv2(+e)|0.849±0.139|0.802±0.183|0.447±0.384|
> |TabPFNv2(+ef)|0.851±0.137|0.804±0.181|0.450±0.381|
> |Mitra(+e)|0.847±0.138|0.800±0.185|0.453±0.388|
> |**Mitra**|0.837±0.145|0.791±0.187|0.460±0.390|
> |TabICL(+e)|0.828±0.163|0.784±0.188|0.470±0.385|
> |**TabPFNv2**|0.831±0.153|0.787±0.188|0.463±0.390|
> |TabICL|0.826±0.159|0.778±0.192|0.477±0.384|
>
> |Model|Avg. Rank|Elo|Winrate|RAcc|Champ Delta|
> |:-|-:|:-|-:|-:|-:|
> |Mitra(+ef)|3.9|1057(+13/-12)|0.59|0.67|13.5|
> |TabPFNv2(+e)|4.0|1047(+12/-12)|0.58|0.66|15.1|
> |TabPFNv2(+ef)|4.0|1041(+11/-11)|0.57|0.65|15.3|
> |Mitra(+e)|4.6|990(+12/-11)|0.48|0.55|20.1|
> |**Mitra**|4.7|982(+12/-12)|0.47|0.54|22.1|
> |TabICL(+e)|4.8|976(+12/-12)|0.46|0.53|23.1|
> |**TabPFNv2**|4.9|968(+13/-12)|0.45|0.52|19.8|
> |TabICL|5.2|938(+12/-12)|0.40|0.43|25.8|
>
> 3. **Diverse priors create a better model foundation for efficient fine-tuning**: The synthetic priors, particularly tree-based ones, are designed to capture decision boundaries. As suggested by prior works (e.g., TabForestPFN, Attic), these are not perfect replicas of real datasets but are ideal for creating a foundation that can be effectively adapted. The diversity of our mixture gives rise to a better model foundation than TabPFNv2, and the gap could not be closed by TabPFNv2 even after extensive fine-tuning, as noted by its fine-tuning performance in Table 2 and confirmed with its authors. More importantly, this aligns with what users care about: achieving the best predictive performance on their datasets efficiently. Mitra fine-tuning is not only highly effective but also efficient, running **~3.5× faster than CatBoost and faster than RealMLP as shown in Figure 14**, delivering state-of-the-art performance with minimal adaptation cost.
> 4. **Pretraining on our priors enables effective fine-tuning**: Our mixture of priors provides a critical foundation that enables effective fine-tuning. To assess the impact of pretraining, we compare Mitra (+ef) with an identical model trained directly on the downstream dataset without any pretraining (denoted as Mitra-no-pretrain (+ef)). For Mitra-no-pretrain (+ef), we conducted a hyper-parameter search over training epochs {50, 100, 200} and maximum learning rates {0.01, 0.001, 0.0001}, reporting its best performance as follows. Training from scratch performs significantly worse, highlighting the importance of pretraining on our mixture of priors.
>
> |Model|AUC|ACC|CE|
> |-|-|-|-|
> |Mitra|0.868±0.133|0.822±0.168|0.403±0.369|
> |Mitra(+e)|0.877±0.126|0.831±0.164|0.394±0.367|
> |**Mitra(+ef)**|0.882±0.124|0.837±0.164|0.370±0.367|
> |**Mitra-no-pretrain(+ef)**|0.672±0.127|0.626±0.194|0.810±0.462|
>
> > HPO for baselines and add TabM.
>
> We have added results on a recent benchmark TabArena [5], where all baselines are performed with HPO, including TabM. We show that Mitra remains the state-of-the-art tabular models on TabArena. **Due to character limit, we only show the top methods. The full table can be referred to in Response to Reviewer Uza4**. Specifically,
>
> 1. **Mitra achieves Pareto efficiency** both in training time and inference time, with TabPFNv2 (HPO + ensemble) being significantly slower.
> 2. **Mitra is the strongest single model**, outperforming all methods even when they perform hyperparameter tuning for 200 iterations. Mitra is only outperformed once the hyperparameter configurations of TabPFNv2 are ensembled together. We are still working on constructing a search space for Mitra, and thus do not have a HPO and HPO + ensemble result at this time.
> 3. **Mitra performs favorably to all non-foundational methods**, including TabM and RealMLP.
>
> |Method|Avg. Rank|Elo|Winrate|Champ Delta|
> |:-|-:|-:|-:|-:|
> |TabPFNv2 (HPO+ensemble)|6.2|1745|0.89|0.05|
> |**Mitra (single)**|7.4|1699|0.86|0.07|
> |TabM (HPO+ensemble)|9.9|1620|0.8|0.1|
> |TabICL (single)|10.1|1617|0.8|0.07|
> |RealMLP (HPO+ensemble)|10.9|1597|0.78|0.09|
> |TabPFNv2 (HPO)|10.9|1596|0.78|0.08|
> |AutoGluon1.3 (4h)|12.6|1551|0.74|0.1|
> |TabPFNv2 (single)|12.7|1548|0.74|0.1|
> |LightGBM (HPO+ensemble)|13.7|1525|0.72|0.12|
> |TabM (HPO)|14.2|1512|0.71|0.11|
> |LightGBM (HPO)|16.2|1470|0.66|0.12|
> |CatBoost (HPO+ensemble)|16.5|1465|0.66|0.12|
> |CatBoost (HPO)|17.3|1444|0.64|0.12|
> |TabM (single)|17.8|1437|0.63|0.14|
> |CatBoost (single)|17.8|1434|0.63|0.14|
> |ModernNCA (HPO)|18|1428|0.62|0.12|
> |XGBoost (HPO+ensemble)|18.5|1420|0.61|0.13|
> ...
> |KNN (single)|45.2|415|0.02|0.59|
>
> > About RF prior.
>
> As we explained in Line 279 to Line 282, while RF shows good diversity and distinctiveness, it also shows the worst performance on real datasets, so it is not as beneficial in the mixture. We will explain this more clearly in the final version.
>
> > Code availability.
>
> The fine-tuning and inference code is already publicly available (at the time of submission the code was going through our institutional legal review as we mentioned in the checklist). However, given the rebuttal policy we are not allowed to provide an anonymous external link. We will ensure the code repository is included and accessible in the final version. Our pretraining code is currently under institutional legal review process, and we plan to release it once this review is complete.
>
> > The reason of training with 16 features instead of 100 features.
>
> Mitra was pre-trained to maximize fine-tuning performance with larger model capacity compared to TabPFNv2. Given computational constraints scaling up to larger features during pretraining requires smaller batch size and eventually leads to worse training efficiency and performance. There are concurrent papers on exploring architectural advances to scale up rows and features such as TabICL and TabFlex. Our focus in this work is on design of effective mixture of priors, and we plan to integrate these scalable architectures as future work to further improve scalability.
>
> > Related work.
>
> We will revise the related work section to better cover deep learning methods and add references [1][2][3][4] in the final version.
>
> > Prior ratio for TBP.
>
> As stated in Line 193, we evenly split the ratios among tree-based priors, each taking 10%. We will explain this more explicitly in the final version.
>
> > Typo.
>
> We will revise “all three datasets” to “all three benchmarks” in the final version.
>
> [5] TabArena: A Living Benchmark for Machine Learning on Tabular Data

---

> ### Comment · Reviewer_EUuF · 2025-08-05
>
> I would like to thank the authors for the reply. I have read the other reviews from the fellow reviewers and the authors rebuttal to all of them. I observed that additional reviewers have raised similar concerns to mine. Below are my answers to the author's rebuttal:
>
> - **Regarding the RF prior**:
>
>    The authors did not answer my concern regarding the RF prior. The point of the criteria and one of the major contributions of the work is on how to build an informative prior mixture. The authors deviate from the criteria with the argument that the prior did not improve the performance, if that is the case, then every component can be investigated in the same manner and one can skip the criteria completely. We can just observe the inference performance on the test benchmark.
>
> - **Regarding training on less than 16 features**:
>
>    I am not really fond of the reply from the authors that training on less than 16 features improved training efficiency. Since TabPFN trained on around 100 features and achieves better results compared to Mitra from a non-fine tuned perspective, this should have been investigated. Usually one starts from a prior common practice and then based on the performance modifies from that point. The authors of TabICL additionally train on more features gradually, so I am surprised on this design choice. The authors use this point as an argument for the lesser in-context learning performance, however, the authors themselves deviated from this prior established procedure.
>
> - **Regarding the fine-tuning time**:
>
>    I observed the results in the Appendix related to the running time of Mitra, however, the authors run Mitra on **8 A100 GPUs** and advocate that the proposed method is 3.5x faster than CatBoost and RealMLP. That is a significant computational resource, how is that resource being used by gradient-boosted decision tree methods and by the simpler neural networks? The comparison is not fair and missleading. Providing results on a single GPU would have been more informative.

---

> > ### Author Response · Authors · 2025-08-06
> >
> > We sincerely thank the reviewer for the follow-up comments. We note that while some initial concerns were shared among some reviewers, our rebuttal and additional experiments have resolved the concerns of the two reviewers who have responded, and they are satisfied with our clarifications. We will continue to make our best efforts to address all remaining concerns. We respond to the reviewer’s remaining questions as follows.
> >
> > > Regarding the RF prior.
> >
> > * We would like to clarify that our evaluation of priors distinguishes **individual prior performance** from **performance in the mixture**. Our criteria contain three parts as discussed on **Lines 38 - 42**: (1) individual prior performance, (2) its diversity and (3) distinctiveness within a mixture of priors. The first criterion of individual prior performance was evaluated by pretraining the TFM exclusively on each prior without considering the mixture, which led to the **Performance Vector** in **Table 1**. This step evaluates the prior *itself*, independent of any interactions in the mixture. An informative prior needs to satisfy all three criteria.
> > * In the case of RF, we observe that it shows the worst performance when the TFM is trained purely on it, as indicated by the performance vector in Table 1, thereby not satisfying criterion 1. As discussed in **Line 279 - 282**, this explains why RF is less informative in the mixture. As an additional example, we see the opposite with DSRF that while it has the second best performance as measured by the performance vector (criteria 1), its diversity and distinctiveness (criteria 2 and 3) is low. Hence, it is not as beneficial as the other priors in the mixture, which indicates that prior performance is a combination of all three of our characteristics.
> >
> > > Regarding training on less than 16 features.
> >
> > * Our decision to pretrain with up to 16 features follows prior established procedure of Attic [1], and this allows for a larger model size and adaptation capability.
> > * As we mentioned in our previous response **Performance gap re: ICL vs ensemble + fine-tuning**, we conducted preliminary experiments to naively increase Mitra’s pretraining features. We explored increasing the feature sizes up to 32, 48, and 64, using strategies such as curriculum learning in TabICL, but did not observe gains out-of-the-box. As noted by reviewer Uza4, our paper opens the door for an interesting discussion that “using lower-dimensional datasets in pre-training stage, and fine-tuning at test time is more advantageous than amortising over higher-dimensional datasets to begin with”. We leave a systematic exploration of the complex interplay between prior mixture, architecture, and training dynamics as future work. Note that our primary focus is getting the best performance with minimal adaptation, to that end, Mitra sets a new state-of-the-art with fast fine-tuning, significantly outperforming TabPFNv2 (+e/+ef) by **~30-50** Elo in Table 2 with std only +4/-4 and **~100** over the top ICL performance.
> >
> > > Regarding the fine-tuning time.
> >
> > We appreciate the reviewer’s point on runtime fairness. Our intent was to demonstrate that Mitra enables fast and scalable adaptation, not to claim that per-GPU efficiency is inherently superior to GBDTs. We have additionally measured single-GPU fine-tuning performance: Mitra (+ef) on a single GPU takes similar time (89 seconds) as CatBoost (83 seconds) while achieving a gain of **138 Elo**. We will clarify it in the final version.
> >
> > We hope the above comments address the reviewer’s concern, and we are happy to address any additional questions.
> >
> > [1] Attic: A New Architecture for Tabular In-Context Learning Transformers

---

> > > ### Comment · Reviewer_EUuF · 2025-08-06
> > >
> > > I would like to thank the authors for the reply, below are my comments to our discussion:
> > >
> > > - RF prior:
> > >
> > >    The point is that there is no clear path in my perspective on the selection of the prior mixture. This goes also in line with the fact that in the end, the authors use all the elements of the mixture. The performance and generalization matrix is not used for example to exclude a certain element from the prior or to adjust the weighting of a certain component. In that perspective, I do not find the outcome of the investigation that convincing. Additionally, as another reviewer mentioned, building the performance and generalization matrix is an expensive operation as you have to train the foundation model on each of the prior elements, so for $M$ priors you have $M$ models that you then have to perform inference on the rest of the elements and on TabRepo.
> > >
> > > - Regarding training on less than 16 features:
> > >
> > >    Did the authors provide results in the proposed work that training on a larger number of features does not provide a gain on the ICL performance and that it decreases the performance for fine-tuning?
> > >
> > > - Regarding the fine-tuning time:
> > >
> > >    I will quote the previous answer from the authors, which was in bold **"running ~3.5× faster than CatBoost and faster than RealMLP as shown in Figure 14"**. It would be great if the authors would not try and misslead the reviewers, given that this sentence was posted multiple times. The common domain practitioner does not have 8 A100 GPUs. If the authors would like to go into detail about the efficiency, the authors could post the inference time for the proposed method and the considered baselines that are not meta-learned.

---

> ### Author Response · Authors · 2025-08-06
>
> We sincerely thank the reviewer for the continued engagement and for the thoughtful follow-up comments. We address the reviewer’s questions as follows.
>
> > RF prior.
>
> Our prior analysis in Table 1 serves as an **in-depth study** to provide insights into what makes effective priors, **rather than a deterministic algorithm** for selecting the mixture. Our proposed three key characteristics—performance, diversity, and distinctiveness—are further validated through our extensive ablation study that shows that the **priors contributing most to the mixture align with this analysis**. For example as we discussed on Lines 285 - 291, combining ET with SCM significantly boosts performance, yielding an Elo improvement of **63**. Similarly, we see that adding GB to the mixture further improves the performance. Adding the remaining priors DT, DSRF, RF leads to a few configurations with similarly good performance on real datasets, aligned with our findings in Table 1 that they are less important due to either lower performance on real datasets or higher diagonal or off-diagonal values. Importantly, this prior analysis part is not a part of the actual pretraining process itself. Compared with previous works such as TabForestPFN which uses SCM + DT and TabICL which uses SCM + GB, we are the first to perform such a comprehensive study on prior importance which explains why our mixture of priors empirically achieve very strong performance. We will further clarify this in the final version. We agree that developing a deterministic algorithm is an interesting direction for future work, and we believe our study opens the door to such developments.
>
> > Regarding training on less than 16 features.
>
> We did not include these preliminary findings in the paper as they are only initial observations and not to the level of a comprehensive ablation study. We present below our preliminary investigation of scaling Mitra to a larger number of features for ICL, and we can add these results to the Appendix to help better guide future work and help motivate our design choice. We explored two approaches:
>
> 1. **Curriculum learning (TabICL-style)**: We continued pre-training from the current Mitra checkpoint while increasing the feature size up to 160 (same pre-training size as TabPFNv2). To accommodate the higher dimensionality, we reduced the batch size from 64 to 6 per device. We trained for 500 steps, saving checkpoints every 100 steps, and reported their ICL performance respectively, denoted by Curriculum (Step 100), Curriculum (Step 200), Curriculum (Step 300), Curriculum (Step 400) and Curriculum (Step 500) in the following table.
> 2. **Feature grouping (TabPFNv2-style)**: We grouped features (2, 3, or 4 features per group) and learned a feature embedding for each group, scaling the input to 32, 48, and 64 features respectively, denoted by Group2 (Features 32), Group3 (Features 48), and Group4 (Features 64) in the following table. We pre-trained for the same number of 22,000 steps as Mitra and reported their ICL performance respectively as below.
>
> In this preliminary study, we already observed a performance drop in ICL performance and therefore did not proceed to evaluate fine-tuning performance at that time. We are currently running the fine-tuning experiments and will report the results once finished (within the discussion period).
>
> Based on these findings we chose to pre-train Mitra on 16 or fewer features. While pre-training with more features could potentially yield improvements, it is not something we have investigated in detail at this time. Empirically, the current design led to strong results, and we anticipate further exploration in this area to lead to even better results.
>
> | Model|   Avg. Rank |   Elo (95% CI)  |   Winrate |   RAcc |   Champ Delta % |
> |:-|-:|-:|-:|-:|-:|
> | Mitra (+ef)              |     3.86894 | 1119 (+8/-8)    |      0.68 |           0.75 |             8.5 |
> | Mitra                    |     5.38182 | 1008 (+7/-8)    |      0.51 |           0.56 |            19.8 |
> | Curriculum (Step 300)    |     5.39621 | 1007 (+8/-8)    |      0.51 |           0.56 |            19.9 |
> | Curriculum (Step 400)    |     5.525   | 998 (+7/-7)     |      0.50 |           0.54 |            21.1 |
> | Group2 (Features 32)     |     5.66439 | 988 (+7/-7)     |      0.48 |           0.53 |            21.7 |
> | Curriculum (Step 500)    |     5.67803 | 987 (+7/-8)     |      0.48 |           0.53 |            20.7 |
> | Curriculum (Step 100)    |     5.70379 | 985 (+7/-7)     |      0.48 |           0.53 |            20.8 |
> | Curriculum (Step 200)    |     5.70379 | 985 (+8/-7)     |      0.48 |           0.53 |            21.1 |
> | Group3 (Features 48)     |     5.87576 | 973 (+7/-8)     |      0.46 |           0.49 |            22.0 |
> | Group4 (Features 64)     |     6.20227 | 950 (+7/-8)     |      0.42 |           0.45 |            23.7 |
>
> We provide response to fine-tuning time in the following comment.

---

> > ### Author Response · Authors · 2025-08-06
> >
> > > Regarding the fine-tuning time.
> >
> > We had no intention of misleading the reviewers. Our original goal in reporting "3.5× faster" was to highlight Mitra’s running time for adaptation on our multi-GPU setup, not to imply that the per-GPU efficiency is inherently superior to GBDTs. To avoid any misunderstanding, we will clarify the single-GPU running time in the final version. Mitra (+ef) on a single A100 GPU takes 89 seconds, comparable to CatBoost’s 83 seconds, while achieving 138 Elo gain.
> >
> > * We additionally provide the **training and inference time** on TabArena as follows, where Mitra (8 * bagging) runs on a **single NVIDIA L4 Tensor Core GPU** (g6.2xlarge, 24 GB vRAM, 8 vCPUs, 32 GB memory). At the time of writing, on-demand prices for g6.2xlarge instances are **$0.97/hr**, with spot pricing at **$0.35/hr**, representing a readily accessible compute instance for practical usage. On TabArena, bagged Mitra’s end-to-end runtime mean across 51 datasets was 1657s (fine-tuning) + 47s (inference) = **1704s**. Using on-demand pricing for g6.2xlarge instances, this means the average cost to fine-tune a 8-fold bagged Mitra was **$0.46**, or **$0.06** without bagging.
> > * For context, this g6.2xlarge instance is **weaker** than machine used to run the other GPU methods (TabPFNv2, TabM, TabICL, TabDPT, and ModernNCA) in the TabArena paper (NVIDIA L40S with 48 GB VRAM). This makes the runtime results not directly comparable, and we expect Mitra to run faster if using the same hardware as used by the benchmark authors. We are currently communicating with the TabArena authors to evaluate Mitra using the exact same hardware as used by the other methods for a fair comparison.
> > * We would like to clarify a previous statement: Mitra does not run faster than non-tuned CatBoost or RealMLP when equalizing compute resources. According to TabArena results, Mitra is the most expensive single model in the benchmark by a respectable margin. We do not intend to mask this fact, and apologize for any prior confusion our response may have caused. With that said, most methods require tuning to reach strong performance, and thus after running 200 iterations of HPO, other methods become far more expensive than a single bagged and fine-tuned Mitra model, as shown in the included runtime columns, while still being outperformed by a single Mitra model. Therefore, we show that Mitra is **Pareto efficient** in this scenario.
> >
> > |Method|Avg. Rank|Elo|Winrate|Champ Delta|**Median Train Time (s/1K)**|**Median Predict Time (s/1K)**|
> > |:-|-:|-:|-:|-:|-:|-:|
> > |TabPFNv2 (HPO+ensemble)|6.2|1745|0.89|0.05|3445.6|48.2|
> > |**Mitra (single)**|7.4|1699|0.86|0.07|457.2|34.6|
> > |TabM (HPO+ensemble)|9.9|1620|0.8|0.1|2828.4|1.6|
> > |TabICL (single)|10.1|1617|0.8|0.07|8.9|1.7|
> > |RealMLP (HPO+ensemble)|10.9|1597|0.78|0.09|6796.3|12.4|
> > |TabPFNv2 (HPO)|10.9|1596|0.78|0.08|3445.6|1|
> > |AutoGluon1.3 (4h)|12.6|1551|0.74|0.1|2309.2|2.6|
> > |TabPFNv2 (single)|12.7|1548|0.74|0.1|4.1|0.4|
> > |LightGBM (HPO+ensemble)|13.7|1525|0.72|0.12|647.6|1.7|
> > |TabM (HPO)|14.2|1512|0.71|0.11|2828.4|0.2|
> > |LightGBM (HPO)|16.2|1470|0.66|0.12|647.6|0.3|
> > |CatBoost (HPO+ensemble)|16.5|1465|0.66|0.12|1465.9|0.7|
> > |CatBoost (HPO)|17.3|1444|0.64|0.12|1465.9|0.1|
> > |TabM (single)|17.8|1437|0.63|0.14|10.4|0.2|
> > |CatBoost (single)|17.8|1434|0.63|0.14|5.7|0.1|
> > |ModernNCA (HPO)|18|1428|0.62|0.12|5944.9|0.5|
> > |XGBoost (HPO+ensemble)|18.5|1420|0.61|0.13|766.1|1.9|
> > |EBM (HPO+ensemble)|20|1390|0.58|0.16|1109.1|0.2|
> > |XGBoost (HPO)|20.1|1386|0.58|0.14|766.1|0.3|
> > |RealMLP (HPO)|20.2|1383|0.57|0.13|6796.3|0.7|
> > |ModernNCA (HPO+ensemble)|20.4|1381|0.57|0.13|5944.9|8.4|
> > |ModernNCA (single)|20.8|1368|0.56|0.15|14.8|0.3|
> > |TorchMLP (HPO+ensemble)|20.9|1370|0.56|0.14|2862.1|2.2|
> > |FastaiMLP (HPO+ensemble)|21.1|1366|0.55|0.16|1358.6|8.1|
> > |TabDPT (single)|22.7|1329|0.52|0.15|27.5|8.9|
> > |EBM (HPO)|23.1|1323|0.51|0.17|1109.1|0|
> > |EBM (single)|23.8|1307|0.49|0.18|5.3|0.1|
> > |RealMLP (single)|25.6|1270|0.45|0.16|22.5|1.6|
> > |FastaiMLP (HPO)|25.6|1273|0.45|0.17|1358.6|0.9|
> > |ExtraTrees (HPO+ensemble)|26.1|1261|0.44|0.18|370.9|1.5|
> > |TorchMLP (HPO)|27|1241|0.42|0.16|2862.1|0.2|
> > |XGBoost (single)|28.2|1214|0.39|0.17|2.4|0.2|
> > |ExtraTrees (HPO)|28.6|1205|0.39|0.2|370.9|0.2|
> > |RandomForest (HPO+ensemble)|30.5|1159|0.34|0.2|527.4|1.4|
> > |LightGBM (single)|30.7|1159|0.34|0.18|2.9|0.1|
> > |RandomForest (HPO)|33.1|1094|0.29|0.21|527.4|0.1|
> > |TorchMLP (single)|33.4|1087|0.28|0.22|10.4|0.2|
> > |FastaiMLP (single)|34.6|1056|0.25|0.23|4.7|0.6|
> > |Linear (HPO+ensemble)|35.2|1032|0.24|0.29|88.6|0.3|
> > |Linear (HPO)|36.2|1005|0.22|0.29|88.6|0.1|
> > |RandomForest (single)|36.3|1000|0.21|0.26|0.4|0.1|
> > |Linear (single)|36.8|985|0.21|0.31|2.3|0.1|
> > |ExtraTrees (single)|37.7|952|0.19|0.28|0.4|0.1|
> > |KNN (HPO+ensemble)|42.6|716|0.08|0.48|3|0.2|
> > |KNN (HPO)|43.7|623|0.05|0.5|3|0|
> > |KNN (single)|45.2|415|0.02|0.59|0.1|0|
> >
> > We hope the above comments address the reviewer’s concern, and we are happy to address any additional questions.

---

> ### Comment · Reviewer_EUuF · 2025-08-07
>
> I would like to thank the authors for engaging in an insightful discussion.
>
> - RF prior:
>
>    I think there was room for further investigation from the generalization and performance matrices that the authors generated. However, it is true that so far this is among the most detailed investigations so far in the domain regarding the mixture of priors.
>
> - Pre-training on more than 16 features:
>
>    I thank the authors for the additional information. I believe this information is interesting for future works and does provide validation to the argument from the authors.
>
> - Regarding the training/inference times:
>
>    I am glad the authors do not shy away from listing the cons of the proposed method. I am in favor of editing the runtime information to reflect a fair comparison between all methods. I additionally think the provided information relating to the train/inference times would be useful to include in the appendix.
>
> Based on the provided information, I believe the proposed work is of interest to the community, given that it provides the prior and a more-detailed investigation. Most of my concerns are addressed and I will recommend acceptance with the condition that the authors open-source the implementation, model weights and prior. It would be great if the authors acknowledged the last point.

---

> > ### Author Response · Authors · 2025-08-07
> >
> > We sincerely thank the reviewer for the support and acceptance recommendation of the paper!
> >
> > * **RF prior**: We agree that there is potential for further exploration of the generalization and performance matrices. We appreciate the reviewer recognizing our work among “the most detailed investigations so far in the domain regarding the mixture of priors”, and we hope our current analysis lays a strong foundation for such future work.
> > * **Pre-training on more than 16 features**: We’re glad that the additional information is helpful. As promised, we have finished the fine-tuning experiments with results shown as follows. In general, scaling up with more features shows comparable or worse performance than using 16 features. The results are inconclusive at the moment and need further investigation, and there is a possibility that fine-tuning might reduce sensitivity to pretraining given a reasonable initial pretrained model. We agree that this line of investigation is valuable and hope to explore it in greater depth in future work.
> > | Model                       |   Avg. Rank |   Elo (95% CI)           |   Winrate |   RAcc |   Champ Delta % |
> > |:----------------------------|------------:|:-----------------|----------:|---------------:|----------------:|
> > | Curriculum (Step 400) (+ef) |     7.625   |  1076 (+6/-5)    |      0.61 |           0.7  |            13.2 |
> > | Mitra (+ef)                 |     7.65606 |  1074 (+6/-5)    |      0.61 |           0.69 |            13.3 |
> > | Curriculum (Step 300) (+ef) |     7.65758 |  1074 (+6/-6)    |      0.61 |           0.7  |            12.8 |
> > | Curriculum (Step 100) (+ef) |     7.67803 |  1074 (+6/-6)    |      0.61 |           0.69 |            13.7 |
> > | Curriculum (Step 500) (+ef) |     7.68939 |  1073 (+6/-5)    |      0.61 |           0.69 |            13.9 |
> > | Curriculum (Step 200) (+ef) |     7.71515 |  1072 (+6/-6)    |      0.6  |           0.69 |            12.9 |
> > | Group2 (Features 32) (+ef)  |     8.225   |  1051 (+6/-6)    |      0.58 |           0.67 |            15.6 |
> > | Group3 (Features 48) (+ef)  |     8.77197 |  1029 (+6/-6)    |      0.54 |           0.65 |            16.4 |
> > | Group4 (Features 64) (+ef)  |     9.39318 |  1004 (+5/-6)    |      0.51 |           0.61 |            18.7 |
> > | Mitra                       |    10.4902  |   960 (+6/-6)    |      0.44 |           0.54 |            23.8 |
> > | Curriculum (Step 300)       |    10.528   |   959 (+6/-6)    |      0.44 |           0.53 |            23.8 |
> > | Curriculum (Step 400)       |    10.7205  |   951 (+6/-6)    |      0.43 |           0.52 |            25   |
> > | Curriculum (Step 500)       |    10.828   |   947 (+6/-6)    |      0.42 |           0.51 |            24.6 |
> > | Curriculum (Step 100)       |    11.0167  |   939 (+6/-5)    |      0.41 |           0.51 |            24.6 |
> > | Curriculum (Step 200)       |    11.0311  |   938 (+5/-5)    |      0.41 |           0.5  |            24.7 |
> > | Group2 (Features 32)        |    11.0326  |   938 (+6/-5)    |      0.41 |           0.51 |            25.3 |
> > | Group3 (Features 48)        |    11.1939  |   932 (+6/-6)    |      0.4  |           0.47 |            25.4 |
> > | Group4 (Features 64)        |    11.7477  |   909 (+6/-5)    |      0.37 |           0.44 |            27.2 |
> > * **Regarding the training/inference times**: We will revise the runtime section to ensure a more fair comparison across methods, and we will include the additional training and inference time in the Appendix in the final version.
> > * **Regarding code**: We believe the tabular data field's tradition of openness and reproducibility has enabled broad adoption of state-of-the-art methods, and as tabular foundation models gain momentum, it is critical that we uphold these values and we appreciate the reviewer’s emphasis on setting a strong precedent during this formative period. Our fine-tuning, inference code, and model weights are publicly available at the moment. We are fully committed and are actively working through the institutional approval process to open source the prior and training script. While we are seeing good progress on that front, we unfortunately cannot make firm guarantees and would appreciate your understanding on this matter.
> >
> > We thank the reviewer again for the support and for recognizing the value of our contributions to the community.

---

> > > ### Comment · Reviewer_EUuF · 2025-08-08
> > >
> > > Not that I think the results will have a significant change, but providing all the versions with the average rank, tends to make insights not visible, especially if the differences are minimal and one model does not dominate the rest. The authors could try additionally providing results for the Mitra + ef and Curriculum (Step 400) + ef.
> > >
> > > Regarding the possibility of not releasing the prior, I understand that the authors need institutional approval, but this would be beneficial for the community given that among the best models TabPFNv2 also does not provide the prior. Otherwise the prior descriptions should be immaculate.

---

> ### Author Response · Authors · 2025-08-09
>
> We sincerely thank the reviewer for the valuable feedback.
>
> We additionally provide the ranking and aggregated results focusing only on Mitra (+ef) and Curriculum (Step 400) (+ef) as follows. Note that ranking-based metrics may not fully reflect the differences when comparing **only** two models, particularly when they exhibit minimal performance differences. To illustrate this, we simulated a scenario where model A outperforms model B by only 0.00001 on 60% of 660 datasets and underperforms model B by 0.00001 on the remaining 40% datasets. Although the differences are negligible, model A leads model B in Elo by a significant margin of 100. This is why we used a larger set of baselines with diverse performance to help provide a more meaningful comparison for ranking-based metrics. For reference, we also provide the aggregate metrics which confirm that the overall aggregated results are very close.
>
> | Model                  |   Avg. Rank |   Elo (95% CI)            |   Winrate |   RAcc |   Champ Delta % |
> |:-----------------------|------------:|------------------|----------:|---------------:|----------------:|
> | Curriculum (Step 400) (+ef) |     1.47121 |  1010 (+12/-11)   |      0.53 |           0.63 |             3.5 |
> | Mitra (+ef)           |     1.52879 |   990 (+11/-12)   |      0.47 |           0.57 |             3.7 |
>
> Aggregated results:
>
> | Model                  | AUC          | ACC          | CE            |
> |------------------------|--------------|--------------|---------------|
> | Curriculum (Step 400) (+ef) | 0.882±0.124  | 0.837±0.163  | 0.369±0.367   |
> | Mitra (+ef)           | 0.882±0.124  | 0.837±0.164  | 0.369±0.367   |
>
> Regarding the prior release, we strongly believe in the importance of open source for the benefit of the community. We are fully committed to this principle and are actively working through the institutional approval process.
>
> We thank the reviewer again for the support and thoughtful feedback!

---

### Official Review · Reviewer_fccq · 2025-07-01

**Clarity:** 3
**Significance:** 2
**Originality:** 3
**Rating:** 5
**Confidence:** 3

**Summary:**

The paper introduces a tabular foundation model (TFM) named MITRA by attempting to carefully analyze the priors used for synthetic data generation to train the TFM. They show that carefully controlling the data distribution provides consistent gains in model performance on both tabular classification as well as regression tasks. The paper also dives into two different architectures, which attempt to highlight the model-agnostic nature of their priors.

**Questions:**

- Pretraining details in the main paper are missing, despite the main contribution being the development of a TFM. Can you please specify all the training details?
- How is the prior importance color assigned in table 1?

**Ethical Concerns:**

["NO or VERY MINOR ethics concerns only"]

**Final Justification:**

I think there is a natural dissatisfaction among reviewers due to gains only being apparent after using a bag of tricks (e.g., ensembling and finetuning). However, I still think this is a useful contribution, and therefore, after also skimming through comments from other reviewers, would recommend accepting this paper. I hope that the authors would release the model and other artifacts as the reviewer EUuF suggested, facilitating further research in this important area.

**Limitations:**

yes

**Quality:**

3

**Strengths And Weaknesses:**

# Strengths
- The paper covers an important problem
- The paper is well-written
- Comprehensive list of experiments

# Weaknesses
- **Unclear reasons for performance gains with only specific tricks (finetuning and ensembling):** the model without finetuning and ensembling is significantly inferior to other approaches in most cases, making it hard to understand the reason for this performance gain with these tricks in place.
- **Inconsistency in evaluation:** the '+e' and '+f' modes are selectively reported in many tables (e.g., Table 3), further masking the true results. I think it should be fine to highlight that this doesn't work in all cases, instead of trying to incorrectly convince the reader that it does.
- **Missing training details:** the paper didn't discuss in any way the pretraining of the model including the dataset size, model parameters, training epochs, which are mentioned in passing towards the end (Fig 5). I would expect a better reading flow where the authors first specify how they pretrain the model and then talk about finetuning. It is just immediately assumed that we have a foundation model that we can evaluate, which is not ideal given that the paper is about the foundation model training.
- **Hard to follow details in some areas:** the paper e.g., highlights that selection based on table 1 for the second best prior isn't optimal without mentioning anything else which makes it really confusing to the reader if they are missing something. They clarify this later in Table 6, but I would expect that the restructuring should make the flow of the paper much smoother to read in general.

---

> ### Author Rebuttal · Authors · 2025-07-31
>
> We sincerely thank the reviewer for the thoughtful and constructive feedback. We appreciate that the reviewer finds (1) the paper covers an **important** problem, (2) the experiments are **comprehensive**, and (3) the paper is **well-written**. We address the reviewer’s concerns as follows.
>
> > Reasons of performance gains after ensemble and fine-tuning.
>
> We explain below why Mitra shows performance gains after ensemble and fine-tuning, which is a direct consequence of our design choices. Our central goal is to build a foundation optimized for maximum adaptability, providing the most practical path to state-of-the-art performance with efficient adaptation. Mitra’s ICL performance is comparable to TabICL [1] (concurrent work) as shown in Table 2, and by fine-tuning from a strong and efficiently adaptable foundation, Mitra ultimately achieves the state-of-the-art performance.
>
> 1. **Performance gap re: ICL vs ensemble + fine-tuning**: We acknowledge the performance gap between ICL and ensemble. We attribute this to (a) Mitra’s pretraining uses fewer features, and (b) TabPFNv2’s use of comprehensive feature preprocessing such as quantile transform and SVD that diversifies its ensemble members. Preliminary attempts to naively increase Mitra’s pretraining features or adopt TabPFNv2’s preprocessing did not yield gains out-of-the-box, highlighting a complex interplay between prior mixture, architecture, and training dynamics. We leave a systematic exploration of this as future work. Note that our primary focus is getting the best performance with minimal adaptation, to that end, Mitra sets a new state-of-the-art with fast fine-tuning, significantly outperforming TabPFNv2 (+e/+ef) by **~30-50 Elo** in Table 2 with std only +4/-4 and **~100** over the top ICL performance.
> 2. **Performance gap due to more efficient feature space**:
>     1. Mitra was pre-trained with a maximum of 16 features for pretraining computational efficiency, whereas TabPFNv2 was pre-trained with an order of magnitude more features (up to 160 features). Despite this discrepancy, Mitra achieves ICL performance comparable to TabPFNv2 on downstream datasets with up to 100 features. This demonstrates the strong **generalization ability** and sample efficiency of our proposed prior mixture, allowing the model to learn more robust patterns that generalize better to unseen, larger feature spaces.
>     2. To further validate that the performance gap is not an effect of our prior mixture, we evaluate Mitra and TabPFNv2 ICL on small datasets with <= 16 features and <= 2k rows. **On these datasets, Mitra shows better ICL performance than TabPFNv2**. There are concurrent papers on exploring architectural advances to scale up rows and features such as TabICL [1] and TabFlex [2]. Our focus in this work is on developing an effective mixture of priors, and we plan to integrate these scalable architectures as future work to further improve scalability.
>     3. Mitra also has better **sample efficiency** as demonstrated in Section 4.4 and Table 5, making it highly effective in real-world scenarios with limited labeled data and an overall best choice across both low- and high-label regimes.
>
> |Model|AUC|ACC|CE|
> |-|-|-|-|
> |Mitra(+ef)|0.851±0.137|0.805±0.184|0.433±0.387|
> |TabPFNv2(+e)|0.849±0.139|0.802±0.183|0.447±0.384|
> |TabPFNv2(+ef)|0.851±0.137|0.804±0.181|0.450±0.381|
> |Mitra(+e)|0.847±0.138|0.800±0.185|0.453±0.388|
> |**Mitra**|0.837±0.145|0.791±0.187|0.460±0.390|
> |TabICL(+e)|0.828±0.163|0.784±0.188|0.470±0.385|
> |**TabPFNv2**|0.831±0.153|0.787±0.188|0.463±0.390|
> |TabICL|0.826±0.159|0.778±0.192|0.477±0.384|
>
> |Model|Avg. Rank|Elo|Winrate|RAcc|Champ Delta|
> |:-|-:|:-|-:|-:|-:|
> |Mitra(+ef)|3.9|1057(+13/-12)|0.59|0.67|13.5|
> |TabPFNv2(+e)|4.0|1047(+12/-12)|0.58|0.66|15.1|
> |TabPFNv2(+ef)|4.0|1041(+11/-11)|0.57|0.65|15.3|
> |Mitra(+e)|4.6|990(+12/-11)|0.48|0.55|20.1|
> |**Mitra**|4.7|982(+12/-12)|0.47|0.54|22.1|
> |TabICL(+e)|4.8|976(+12/-12)|0.46|0.53|23.1|
> |**TabPFNv2**|4.9|968(+13/-12)|0.45|0.52|19.8|
> |TabICL|5.2|938(+12/-12)|0.40|0.43|25.8|
>
> 3. **Pretraining on our priors enables effective fine-tuning**: Our mixture of priors provides a critical foundation that enables effective fine-tuning. To assess the impact of pretraining, we compare Mitra (+ef) with an identical model trained directly on the downstream dataset without any pretraining (denoted as Mitra-no-pretrain (+ef)). For Mitra-no-pretrain (+ef), we conducted a hyper-parameter search over training epochs {50, 100, 200} and maximum learning rates {0.01, 0.001, 0.0001}, reporting its best performance as follows. Training from scratch performs significantly worse, highlighting the importance of pretraining on our proposed mixture of priors.
>
> |Model|AUC|ACC|CE|
> |-|-|-|-|
> |Mitra|0.868±0.133|0.822±0.168|0.403±0.369|
> |Mitra(+e)|0.877±0.126|0.831±0.164|0.394±0.367|
> |**Mitra(+ef)**|0.882±0.124|0.837±0.164|0.370±0.367|
> |**Mitra-no-pretrain(+ef)**|0.672±0.127|0.626±0.194|0.810±0.462|
>
> 4. **Diverse priors create a better model foundation for efficient fine-tuning**: The synthetic priors, particularly tree-based ones, are designed to capture decision boundaries. As suggested by prior works (e.g., TabForestPFN [3], Attic [4]), these are not perfect replicas of real-world datasets but are ideal for creating a foundation that can be effectively adapted. The diversity of our mixture gives rise to a better model foundation than TabPFNv2, and the gap could not be closed by TabPFNv2 even after extensive fine-tuning, as noted by its fine-tuning performance in Table 2 and confirmed with its authors. More importantly, this aligns with what users care about: achieving the best predictive performance on their datasets efficiently. Mitra fine-tuning is not only highly effective but also computationally efficient, running **~3.5× faster than CatBoost and faster than RealMLP as shown in Figure 14**, delivering state-of-the-art performance with minimal adaptation cost.
> 5. **Additional results on TabArena also show Mitra’s SOTA performance**: We have run additional comparison on a recent benchmark TabArena [5]. **Due to character limit, we only show the top methods. The full table can be referred to in Response to Reviewer Uza4**. We show that
>     1. **Mitra achieves Pareto efficiency both in training time and inference time**, with TabPFNv2 (HPO + ensemble) being significantly slower.
>     2. **Mitra is the strongest single model**, outperforming all methods even when they perform hyperparameter tuning for 200 iterations. Mitra is only outperformed once the hyperparameter configurations of TabPFNv2 are ensembled together. We are still working on constructing a search space for Mitra, and thus do not have a HPO and HPO + ensemble result at this time.
>     3. **Mitra performs favorably to all non-foundational methods**, including TabM and RealMLP.
>
> |Method|Avg. Rank|Elo|Winrate|Champ Delta|
> |:-|-:|-:|-:|-:|
> |TabPFNv2 (HPO+ensemble)|6.2|1745|0.89|0.05|
> |**Mitra (single)**|7.4|1699|0.86|0.07|
> |TabM (HPO+ensemble)|9.9|1620|0.8|0.1|
> |TabICL (single)|10.1|1617|0.8|0.07|
> |RealMLP (HPO+ensemble)|10.9|1597|0.78|0.09|
> |TabPFNv2 (HPO)|10.9|1596|0.78|0.08|
> |AutoGluon1.3 (4h)|12.6|1551|0.74|0.1|
> |TabPFNv2 (single)|12.7|1548|0.74|0.1|
> |LightGBM (HPO+ensemble)|13.7|1525|0.72|0.12|
> |TabM (HPO)|14.2|1512|0.71|0.11|
> |LightGBM (HPO)|16.2|1470|0.66|0.12|
> |CatBoost (HPO+ensemble)|16.5|1465|0.66|0.12|
> |CatBoost (HPO)|17.3|1444|0.64|0.12|
> |TabM (single)|17.8|1437|0.63|0.14|
> |CatBoost (single)|17.8|1434|0.63|0.14|
> |ModernNCA (HPO)|18|1428|0.62|0.12|
> |XGBoost (HPO+ensemble)|18.5|1420|0.61|0.13|
> ...
> |KNN (single)|45.2|415|0.02|0.59|
>
> > Evaluation consistency.
>
> Table 3 evaluates 1D attention model. The TabForestPFN baseline does not offer native ensemble logic, and the TabPFNv1 baseline does not offer native fine-tuning logic. For these reasons, we report results under the capabilities available in the respective baselines to ensure a fair comparison. We present comparisons that are meaningful and applicable to each method, and are not hiding any results. We will make this distinction on the 1D vs. 2D models clearer in the final version.
>
> > Training details.
>
> We discussed the dataset size, model size, epochs in Figure 5 and more details in Appendix B.3. We pretrain Mitra for 22,000 steps on 45 million synthetically generated datasets. Mitra is built on a Transformer with 12 layers, 512 embedding size and 4 attention heads. The resulting model contains 72M parameters. We will adjust the paper flow to introduce these pretraining details earlier before discussing fine-tuning and evaluation.
>
> > On the second-best prior.
>
> We have explained and compared the second and the third best prior in Line 164 - Line 169. The major reason is that the third-best prior ET shows better diversity and distinctiveness compared to the second-best prior DSRF. We also revisited this in the ablation study section in Line 277 - Line 279. We will make this more explicit in the final version.
>
> > Prior importance color in Table 1.
>
> We use a color gradient in Table 1 to visually indicate the relative quality of each prior. The best-performing priors are shown in the darkest green, while the worst-performing priors are shown in the darkest red. Specifically, for diversity score on the diagonal of the generalizability matrix, lower values are better, with the lowest being the darkest green and the highest being the darkest red. For performance vector scores, higher values are better, with the highest being the darkest green and the lowest being the darkest red. We will make this clearer in the final version.
>
> [1] TabICL: A Tabular Foundation Model for In-Context Learning on Large Data
>
> [2] TabFlex: Scaling Tabular Learning to Millions with Linear Attention
>
> [3] Fine-tuned In-Context Learning Transformers are Excellent Tabular Data Classifiers
>
> [4] Attic: A New Architecture for Tabular In-Context Learning Transformers
>
> [5] TabArena: TabArena: A Living Benchmark for Machine Learning on Tabular Data

---

> > ### Comment · Reviewer_fccq · 2025-08-05
> > **Official comment by reviewer fccq**
> >
> > I am very thankful to the authors for taking the time in writing such a detailed response. I am satisfied with the responses, and would be happy to see the paper through. I would certainly suggest including at least the basic training details in the main paper to the authors in order to make sure that the reader has the right context to understand. Perhaps the distinction with the features is also worth mentioning since the reader should immediately notice the drop in performance with vanilla ICL.

---

> > > ### Author Response · Authors · 2025-08-06
> > >
> > > We sincerely thank the reviewer for the support on our work! We will make sure to include the training details and mention the feature distinction in the final version.

---

### Official Review · Reviewer_Uza4 · 2025-07-02

**Clarity:** 4
**Significance:** 4
**Originality:** 3
**Rating:** 5
**Confidence:** 3

**Summary:**

With the emergence of tabular foundational models (TFMs) trained on vast collections of synthetic data, this work aims to systematically identify the best combination and type of synthetic priors for effective real-world data generalisation. The authors propose three qualities of an effective prior: standalone performance on real-world data, diversity (not overfitting on itself), and distinctiveness (not covered by other proposed priors). Guided by these criteria, they assemble curated sets of synthetic priors and train MITRA, a new TFM that performs comparably or outperforms state-of-the-art TFMs across a range of classification and regression benchmarks.

**Questions:**

1. In L915 in A.2.1 you mention that you design the classification datasets in order to encourage a balanced distribution of labels across classes. Have you tested the model in cases where there is significant class imbalance too? Would that be out-of-distribution for the model given the design choice for the priors?
2. Could you please give more details about the exact parameterisation of the training loss, as well as any unmentioned pre-processing techniques that might have been applied to the data?
3. Could you please comment on how normalisation is performed on the test sets? What happens if some of the real-data shows some very large values that would be out-of-distribution if any min-max normalisation based on the training data were performed?
4. It seems that the 2D attention is needed for SOTA performance. Up to how many nodes and rows do the authors think it is feasible to scale?

**Ethical Concerns:**

["NO or VERY MINOR ethics concerns only"]

**Final Justification:**

I maintain my view that this paper is of interest to the community, supported by substantial empirical evidence and thorough ablations, and therefore I retain my rating of 5 (Accept). While further work is needed to convincingly demonstrate that pre-training on lower-dimensional datasets and fine-tuning at test time is more advantageous than directly amortising over higher-dimensional datasets, I believe this paper has the potential to open an important discussion on this topic.

**Limitations:**

Perhaps one additional limitation not mentioned by the authors is the increased runtime of the best performing MITRA model (which requires fine-tuning on the training set of the target data). This might become limiting at high throughput rates.

**Paper Formatting Concerns:**

No paper formatting concerns.

**Quality:**

3

**Strengths And Weaknesses:**

**Strengths**
1. **Comprehensive, Systematic Evaluation** - The work involves a wide range of trained models on different combinations of priors to empirically sustain the claims regarding the effectiveness of each prior. The study is systematic and thoroughly conducted, with a wide range of ablations.
2. **Robust Baseline Comparisons** - MITRA is benchmarked against a wide array of state-of-the-art baselines—including TabPFNv2—using multiple metrics, ensuring a fair and thorough evaluation.
3. **Broad, Insightful Analysis** - Beyond addressing the choice of priors, the study examines sample efficiency, scaling laws (model and data size), and the effects of fine-tuning and ensembling. Taken together, these help understand what contributes to the success of the model the most.
4. **Clear Motivation and Practical Impact** - The paper is well-written and directly addresses a relevant question in the light of the rise of TFMs: how to select synthetic data sources that genuinely improve real-world tabular model performance.

**Weaknesses**
1. **Incomplete model specification** - Although the experiments are very well explained and detailed, the model specification is not as thorough. Section B.3.1 gives some details about the transformer hyperparameters, but the authors do not give too much detail about:
   - More detailed data pre-processing steps (e.g. are the categorical variables encoded using one-hot-encoding?),
   - Test-set normalisation procedure, given that training uses dataset-specific min–max scaling on the support set.
   - Exact form of the likelihood mentioned in L176 (distributional assumptions, factorisation across query points, etc.).
2. **Reliance on Post-Training Techniques** - Performance of vanilla MITRA (no fine-tuning, no ensembling) underperforms many baselines, implying that selecting strong synthetic priors is not enough to reach SOTA. To get there, the choice of priors need to be combined with well-tuned post-training techniques. These in turn increase the average runtime of the model (which is at least 20x higher than for TabPFNv2 (+e)), as shown in Figure 14, while bringing only marginal performance improvements.
3. **Marginal Gains vs. Compute Trade-Off** - While MITRA (+ef) is consistently the best, the 95\% confidence intervals for Elo sometimes overlap (e.g. in Table 2), and the std of the metrics are quite high. Other considerations that are worth taking into account when developing TFMs are training costs, and runtime performance. Given that MITRA’s runtime is quite high (because of the need for fine-tuning as a consequence of maximum dimension of 16 in the training data), and given that the differences between models are sometimes small, some other models may offer better accuracy–efficiency trade-offs.
4. **Minor figure formatting issues** - Figures 3, 4, and 5 are fairly small. Perhaps for Figure 5 at least the authors could include only the zoomed in versions?

---

> ### Author Rebuttal · Authors · 2025-07-31
>
> We sincerely thank the reviewer for the thoughtful and constructive feedback. We appreciate that the reviewer finds the paper (1) **comprehensive and systematic** in its evaluation; (2) **robust** in baseline comparisons; (3) **broad and insightful** in its analysis; and (4) **clear in motivation** and **practical impact** and **well-written**. We address the reviewer’s concerns as follows.
>
> > Running time of post-training techniques.
>
> While the running time of Mitra with ensemble and fine-tuning is higher than TabPFNv2 and Mitra in ICL mode, it is still reasonable and practical for tabular prediction models. For instance, it is **~3.5 times faster than CatBoost and also faster than RealMLP** as shown in **Figure 14**. Mitra is targeted especially for smaller tabular datasets which makes the practical running time fast. In addition, Mitra (bagging) is faster than TabPFNv2 with post-hoc ensemble, while achieving the best performance with advanced ensemble techniques (**Figure 3**).
>
>
> > On performance improvement.
>
> We added a new experiment showing that Mitra offers substantial rather than marginal performance gains on a recent tabular benchmark TabArena [1]. We show that
>
> 1. **Mitra achieves Pareto efficiency** both in training time and inference time, with TabPFNv2 (HPO + ensemble) being significantly slower.
> 2. **Mitra is the strongest single model**, outperforming all methods even when they perform hyperparameter tuning for 200 iterations. Mitra is only outperformed once the hyperparameter configurations of TabPFNv2 are ensembled together. We are still working on constructing a search space for Mitra, and thus do not have a HPO and HPO + ensemble result at this time.
> 3. **Mitra performs favorably to all non-foundational methods**, including TabM and RealMLP.
>
> |Method|Avg. Rank|Elo|Winrate|Champ Delta|
> |:-|-:|:-|-:|-:|
> |TabPFNv2 (HPO+ensemble)|6.2|1745(+41/-39)|0.89|0.05|
> |**Mitra (single)**|7.4|1699(+46/-38)|0.86|0.07|
> |TabM (HPO+ensemble)|9.9|1620(+42/-41)|0.8|0.1|
> |TabICL (single)|10.1|1617(+46/-43)|0.8|0.07|
> |RealMLP (HPO+ensemble)|10.9|1597(+39/-40)|0.78|0.09|
> |TabPFNv2 (HPO)|10.9|1596(+38/-46)|0.78|0.08|
> |AutoGluon1.3 (4h)|12.6|1551(+45/-39)|0.74|0.1|
> |TabPFNv2 (single)|12.7|1548(+41/-43)|0.74|0.1|
> |LightGBM (HPO+ensemble)|13.7|1525(+33/-46)|0.72|0.12|
> |TabM (HPO)|14.2|1512(+38/-35)|0.71|0.11|
> |LightGBM (HPO)|16.2|1470(+40/-44)|0.66|0.12|
> |CatBoost (HPO+ensemble)|16.5|1465(+36/-38)|0.66|0.12|
> |CatBoost (HPO)|17.3|1444(+37/-38)|0.64|0.12|
> |TabM (single)|17.8|1437(+37/-40)|0.63|0.14|
> |CatBoost (single)|17.8|1434(+30/-37)|0.63|0.14|
> |ModernNCA (HPO)|18|1428(+41/-33)|0.62|0.12|
> |XGBoost (HPO+ensemble)|18.5|1420(+32/-36)|0.61|0.13|
> |EBM (HPO+ensemble)|20|1390(+32/-34)|0.58|0.16|
> |XGBoost (HPO)|20.1|1386(+34/-32)|0.58|0.14|
> |RealMLP (HPO)|20.2|1383(+33/-33)|0.57|0.13|
> |ModernNCA (HPO+ensemble)|20.4|1381(+35/-32)|0.57|0.13|
> |ModernNCA (single)|20.8|1368(+31/-32)|0.56|0.15|
> |TorchMLP (HPO+ensemble)|20.9|1370(+33/-38)|0.56|0.14|
> |FastaiMLP (HPO+ensemble)|21.1|1366(+36/-42)|0.55|0.16|
> |TabDPT (single)|22.7|1329(+38/-36)|0.52|0.15|
> |EBM (HPO)|23.1|1323(+41/-36)|0.51|0.17|
> |EBM (single)|23.8|1307(+33/-30)|0.49|0.18|
> |RealMLP (single)|25.6|1270(+36/-37)|0.45|0.16|
> |FastaiMLP (HPO)|25.6|1273(+31/-41)|0.45|0.17|
> |ExtraTrees (HPO+ensemble)|26.1|1261(+31/-31)|0.44|0.18|
> |TorchMLP (HPO)|27|1241(+33/-34)|0.42|0.16|
> |XGBoost (single)|28.2|1214(+35/-31)|0.39|0.17|
> |ExtraTrees (HPO)|28.6|1205(+39/-39)|0.39|0.2|
> |RandomForest (HPO+ensemble)|30.5|1159(+39/-38)|0.34|0.2|
> |LightGBM (single)|30.7|1159(+34/-40)|0.34|0.18|
> |RandomForest (HPO)|33.1|1094(+47/-36)|0.29|0.21|
> |TorchMLP (single)|33.4|1087(+43/-32)|0.28|0.22|
> |FastaiMLP (single)|34.6|1056(+32/-32)|0.25|0.23|
> |Linear (HPO+ensemble)|35.2|1032(+43/-35)|0.24|0.29|
> |Linear (HPO)|36.2|1005(+42/-50)|0.22|0.29|
> |RandomForest (single)|36.3|1000(+0/-0)|0.21|0.26|
> |Linear (single)|36.8|985(+49/-32)|0.21|0.31|
> |ExtraTrees (single)|37.7|952(+42/-37)|0.19|0.28|
> |KNN (HPO+ensemble)|42.6|716(+42/-52)|0.08|0.48|
> |KNN (HPO)|43.7|623(+49/-69)|0.05|0.5|
> |KNN (single)|45.2|415(+77/-96)|0.02|0.59|
>
> > Evaluate under class imbalance settings.
>
> To evaluate robustness to class imbalance, we computed the imbalance ratio (max class : min class) across TabRepo datasets and selected 14 datasets with a ratio > 10 as imbalanced datasets. Mitra **remains state-of-the-art performance on the class-imbalanced datasets**, as shown below, for both aggregate and ranking metrics.
>
> | Model                         | AUC          | ACC         | CE         |
> |-------------------------------|------------------|------------------|------------------|
> | **Mitra (+ef)**                   | 0.875 ± 0.128  | 0.871 ± 0.132  | 0.320 ± 0.306  |
> | TabPFNv2 (+e)                 | 0.872 ± 0.132  | 0.870 ± 0.134  | 0.328 ± 0.311  |
> | TabPFNv2 (+ef)                | 0.869 ± 0.130  | 0.863 ± 0.132  | 0.349 ± 0.306  |
> | TabICL (+e)                   | 0.864 ± 0.136  | 0.859 ± 0.139  | 0.346 ± 0.314  |
> | Mitra (+e)                    | 0.868 ± 0.132  | 0.863 ± 0.137  | 0.343 ± 0.319  |
> | TabPFNv2                      | 0.865 ± 0.136  | 0.864 ± 0.138  | 0.338 ± 0.317  |
> | Mitra                         | 0.867 ± 0.130  | 0.861 ± 0.139  | 0.347 ± 0.320  |
> | TabICL                        | 0.859 ± 0.136  | 0.856 ± 0.140  | 0.353 ± 0.316  |
>
> | Model             | Avg. Rank | Elo       | Winrate | RAcc | Champ Delta |
> |:------------------|--------:|:------------------|--------:|-------------:|--------------:|
> | **Mitra (+ef)**       |     3.6 | 1081 (+13/-13)    |    0.63 |         0.74 |          11.5 |
> | TabPFNv2 (+e)     |     4.1 | 1039 (+14/-13)    |    0.56 |         0.68 |          14.4 |
> | TabPFNv2 (+ef)    |     4.3 | 1022 (+12/-13)    |    0.54 |         0.64 |          16.6 |
> | TabICL (+e)       |     4.4 | 1008 (+13/-13)    |    0.51 |         0.62 |          20.5 |
> | Mitra (+e)        |     4.7 |  979 (+12/-14)    |    0.46 |         0.58 |          19.6 |
> | TabPFNv2          |     4.9 |  964 (+13/-12)    |    0.44 |         0.55 |          18.0 |
> | Mitra             |     5.0 |  956 (+14/-13)    |    0.43 |         0.55 |          21.6 |
> | TabICL            |     5.0 |  951 (+13/-14)    |    0.42 |         0.50 |          23.6 |
>
> > Model specification.
>
> 1. **Categorical feature encoding**: We use ordinal encoding for categorical features.
> 2. **Normalization**: In Appendix B.3.2, we mentioned min-max normalization. We would like to clarify that this applies only to regression targets and not to input features. We will make this clear in the final version. To normalize features, we apply uniform quantile transform based on the support set, followed by standard normalization based on the mean and standard deviation from the support set. To normalize targets for regression task, we use min-max normalization based on the support set. The normalization is consistently applied for both pretraining datasets and inference datasets.
> 3. **Likelihood**: We will clarify the likelihood formulation in the final version: the population version of the likelihood takes a form of an expectation over the table $D$, with each table sampled from the prior mixture. Given the sampled table, the model assumes $q$ query rows are conditionally independent given the in-context examples, so that the log-likelihood within the expectation is decomposed into a sum of $q$ individual terms, one per query row. When the query label corresponds to a classification task, its (conditional) distribution is assumed to be categorical, making the training objective equivalent to minimizing the cross-entropy loss. When the query label corresponds to a regression task, its (conditional) distribution is assumed to be Gaussian and the training objective corresponds to minimizing the Mean Squared Error (MSE) loss.
>
> > Large values in test data.
>
> As described in model specification part, to normalize features, we apply uniform quantile transform based on the support set, followed by standard normalization based on the mean and standard deviation from the support set. To normalize targets for regression task, we use min-max normalization based on the support set. The normalization is consistently applied for both pretraining datasets and inference datasets.
>
> We observed that the model is able to extrapolate and generate predictions beyond [0, 1] for regression predictions. For example, on Tabrepo regression 167210 across 10 folds, 6 entries are out of distribution. Mitra successfully extrapolates 4 of these 6 entries, achieving a 67% success rate, with a mean absolute error of 0.03 and mean squared error of 0.001 in the normalized space. We leave other preprocessing techniques for improved extrapolation as future work.
>
> > Up to how many rows and features to scale.
>
> There are concurrent papers on exploring architectural advances to scale up rows and features such as TabICL [2] and TabFlex [3]. Our focus in this work is on design of effective mixture of priors, and we plan to integrate these scalable architectures as future work to further improve scalability.
>
> > Revise figure format.
>
> Thank the reviewers for the comment. We will reformat Figure 3, 4, 5 to make them larger and include only the zoomed in versions in the final paper.
>
> [1] TabArena: A Living Benchmark for Machine Learning on Tabular Data
>
> [2] TabICL: A Tabular Foundation Model for In-Context Learning on Large Data
>
> [3] TabFlex: Scaling Tabular Learning to Millions with Linear Attention

---

> > ### Comment · Reviewer_Uza4 · 2025-08-02
> >
> > Thank you for the additional results and detailed explanations.
> >
> > Please ensure that the revised version of the paper includes the clarifications regarding normalization and likelihood computation. I believe these are essential for completeness.
> >
> > Regarding the following point:
> >
> > > We observed that the model is able to extrapolate and generate predictions beyond [0, 1] for regression predictions. For example, on Tabrepo regression 167210 across 10 folds, 6 entries are out of distribution. Mitra successfully extrapolates 4 of these 6 entries, achieving a 67% success rate
> >
> > Could you clarify what you mean by “successfully extrapolates” in 4 out of 6 entries? Specifically, what occurs in the remaining 2 cases? While I understand that this is not the main focus of the paper, understanding the nature of these failure cases would be valuable for identifying potential areas of improvement.

---

> > > ### Author Response · Authors · 2025-08-03
> > >
> > > We sincerely thank the reviewer for the thoughtful feedback. We will ensure the final version includes clarifications regarding normalization and likelihood computation.
> > >
> > > We define "successfully extrapolates" as the model producing predictions beyond the normalized range [0, 1], which is predicting a value greater than 1 when the ground truth exceeds 1, or a value less than 0 when the ground truth is below 0. On the TabRepo regression 167210 dataset across 10 folds, Mitra successfully extrapolates 4 out of 6 entries. This demonstrates that Mitra can generate predictions for out-of-distribution targets. For the remaining two entries, it predicts values within the normalization range [0, 1] when the ground truth lies outside this range. However, these two entries are only slightly outside the range, so it is reasonable to predict values within [0, 1], and the resulting errors remain small, with a mean absolute error of 0.02 and a mean squared error of 0.0004 in the normalized space. We will clarify this in the final version and explore other preprocessing techniques to further improve extrapolation in out-of-distribution scenarios as future work. We sincerely thank the reviewer again for the insightful feedback.

---

> > > > ### Comment · Reviewer_Uza4 · 2025-08-05
> > > >
> > > > Thank you for the clarification, please do include it in the revised version of the paper too.
> > > >
> > > > I maintain my opinion that this is a paper of interest to the community, with significant empirical evidence and ablations, and hence maintain my rating of Accept. There is still some way to convince the community that using lower-dimensional datasets in pre-training stage, and fine-tuning at test time is more advantageous than amortising over higher-dimensional datasets to begin with, but I think this paper can be the one that opens up this interesting discussion.

---

> > > > > ### Author Response · Authors · 2025-08-06
> > > > >
> > > > > We sincerely thank the reviewer for the support and recognition of the impact of our work! We will ensure to include the clarifications in the final version. We also appreciate the reviewer’s thoughtful remark. We agree that this is an important open direction that we will continue to explore, and are encouraged that our work can help initiate this discussion in the community.

---

### Official Review · Reviewer_gtzp · 2025-07-05

**Clarity:** 4
**Significance:** 1
**Originality:** 3
**Rating:** 3
**Confidence:** 4

**Summary:**

The paper proposes an approach to select synthetic data priors for tabular foundation model pre-training. Specifically authors propose three criteria for prior selection - performance, diversity and distinctiveness - and design an empirical procedure to select priors that optimise these criteria. Empirical results show that when a foundation model is pre-trained with a proposed mixture better downstream generalisation results can be achieved.

**Questions:**

-Do you have any additional explanation for why Mitra is not beating TabPFNv2 in non-finetuned settings? For current results it does look like TabPFNv2's synthetic data strategy is superior and leads to a better "foundation" model base.
-The procedure to select prior mixture in Section 3.1 is quite expensive and requires training M models. I might have missed it in the appendix but do you have run time results for this procedure and stability across different restarts?

**Ethical Concerns:**

["NO or VERY MINOR ethics concerns only"]

**Limitations:**

Yes

**Quality:**

4

**Strengths And Weaknesses:**

Strengths:
-The paper is very well written an easy to follow. The proposed criteria for prior selection is innovative and conceptually sound.
-The procedure to derive the mixture is novel although it does look computationally expensive.
-Extensive empirical evaluation is provided on real-world datasets with detailed ablation study.

Weaknesses:
-I have a major concern with the empirical results. In the main result table TabPFNv2 base/e/ef would be the closest baseline to compare to for Mitra base/e/ef. However, looking at direct pairwise comparisons TabPFNv2 vs Mitra, TabPFNv2 (+e) vs Mitra (+e), and TabPFNv2 (+ef) vs Mitra (+ef), only the fine-tuned Mitra (+ef) performs better. This really calls into question the premise of the paper that the proposed procedure for prior selection leads to better downstream generalisation. The gap between non-finetuned TabPFNv2 and Mitra is consistent and quite large, particularly for the ensembled versions.

---

> ### Author Rebuttal · Authors · 2025-07-31
>
> We sincerely thank the reviewer for the thoughtful and constructive feedback. We appreciate that the reviewer finds (1) the proposed criteria for prior selection **innovative and conceptually sound**; (2) the procedure to derive the mixture **novel**; (3) the empirical evaluation **extensive with detailed** ablation study; (3) the paper **very well written and easy to follow**. We address the reviewer’s concerns as follows.
>
> > Mitra’s ICL performance relative to TabPFNv2 does not undermine the quality of our proposed prior mixture but reflects our design choices.
>
> We clarify below why Mitra’s ICL performance relative to TabPFNv2 reflects our design choices rather than a limitation of the prior mixture. Our central goal is to build a foundation optimized for maximum adaptability, providing the most practical path to state-of-the-art performance with efficient adaptation. Mitra’s ICL performance is comparable to TabICL [1] (concurrent work) as shown in Table 2, and by fine-tuning from a strong and efficiently adaptable foundation, Mitra ultimately achieves the state-of-the-art performance.
>
> 1. **Performance gap re: ICL vs ensemble + fine-tuning**: We acknowledge the performance gap between ICL and ensemble. We attribute this to (a) Mitra’s pretraining uses fewer features, and (b) TabPFNv2’s use of comprehensive feature preprocessing such as quantile transform and SVD that diversifies its ensemble members. Preliminary attempts to naively increase Mitra’s pretraining features or adopt TabPFNv2’s preprocessing did not yield gains out-of-the-box, highlighting a complex interplay between prior mixture, architecture, and training dynamics. We leave a systematic exploration of this as future work. Note that our primary focus is getting the best performance with minimal adaptation, to that end, Mitra sets a new state-of-the-art with fast fine-tuning, significantly outperforming TabPFNv2 (+e/+ef) by **~30-50 Elo** in Table 2 with std only +4/-4 and **~100** over the top ICL performance.
> 2. **Performance gap due to more efficient feature space**:
>     1. Mitra was pre-trained with a maximum of 16 features for pretraining computational efficiency, whereas TabPFNv2 was pre-trained with an order of magnitude more features (up to 160 features). Despite this discrepancy, Mitra achieves ICL performance comparable to TabPFNv2 on downstream datasets with up to 100 features. This demonstrates the strong **generalization ability** and sample efficiency of our proposed prior mixture, allowing the model to learn more robust patterns that generalize better to unseen, larger feature spaces.
>     2. To further validate that the performance gap is not an effect of our prior mixture, we evaluate Mitra and TabPFNv2 ICL on small datasets with <= 16 features and <= 2k rows. **On these datasets, Mitra shows better ICL performance than TabPFNv2**. There are concurrent papers on exploring architectural advances to scale up rows and features such as TabICL [1] and TabFlex [2]. Our focus in this work is on developing an effective mixture of priors, and we plan to integrate these scalable architectures as future work to further improve scalability.
>     3. Mitra also has better **sample efficiency** as demonstrated in Section 4.4 and Table 5, making it highly effective in real-world scenarios with limited labeled data and an overall best choice across both low- and high-label regimes.
>
> |Model|AUC|ACC|CE|
> |-|-|-|-|
> |Mitra(+ef)|0.851±0.137|0.805±0.184|0.433±0.387|
> |TabPFNv2(+e)|0.849±0.139|0.802±0.183|0.447±0.384|
> |TabPFNv2(+ef)|0.851±0.137|0.804±0.181|0.450±0.381|
> |Mitra(+e)|0.847±0.138|0.800±0.185|0.453±0.388|
> |**Mitra**|0.837±0.145|0.791±0.187|0.460±0.390|
> |TabICL(+e)|0.828±0.163|0.784±0.188|0.470±0.385|
> |**TabPFNv2**|0.831±0.153|0.787±0.188|0.463±0.390|
> |TabICL|0.826±0.159|0.778±0.192|0.477±0.384|
>
> |Model|Avg. Rank|Elo|Winrate|RAcc|Champ Delta|
> |:-|-:|:-|-:|-:|-:|
> |Mitra(+ef)|3.9|1057(+13/-12)|0.59|0.67|13.5|
> |TabPFNv2(+e)|4.0|1047(+12/-12)|0.58|0.66|15.1|
> |TabPFNv2(+ef)|4.0|1041(+11/-11)|0.57|0.65|15.3|
> |Mitra(+e)|4.6|990(+12/-11)|0.48|0.55|20.1|
> |**Mitra**|4.7|982(+12/-12)|0.47|0.54|22.1|
> |TabICL(+e)|4.8|976(+12/-12)|0.46|0.53|23.1|
> |**TabPFNv2**|4.9|968(+13/-12)|0.45|0.52|19.8|
> |TabICL|5.2|938(+12/-12)|0.40|0.43|25.8|
>
> 3. **Pretraining on our priors enables effective fine-tuning**: Our mixture of priors provides a critical foundation that enables effective fine-tuning. To assess the impact of pretraining, we compare Mitra (+ef) with an identical model trained directly on the downstream dataset without any pretraining (denoted as Mitra-no-pretrain (+ef)). For Mitra-no-pretrain (+ef), we conducted a hyper-parameter search over training epochs {50, 100, 200} and maximum learning rates {0.01, 0.001, 0.0001}, reporting its best performance as follows. Training from scratch performs significantly worse, highlighting the importance of pretraining on our proposed mixture of priors.
>
> |Model|AUC|ACC|CE|
> |-|-|-|-|
> |Mitra|0.868±0.133|0.822±0.168|0.403±0.369|
> |Mitra(+e)|0.877±0.126|0.831±0.164|0.394±0.367|
> |**Mitra(+ef)**|0.882±0.124|0.837±0.164|0.370±0.367|
> |**Mitra-no-pretrain(+ef)**|0.672±0.127|0.626±0.194|0.810±0.462|
>
> 4. **Diverse priors create a better model foundation for efficient fine-tuning**: The synthetic priors, particularly tree-based ones, are designed to capture decision boundaries. As suggested by prior works (e.g., TabForestPFN [3], Attic [4]), these are not perfect replicas of real-world datasets but are ideal for creating a foundation that can be effectively adapted. The diversity of our mixture gives rise to a better model foundation than TabPFNv2, and the gap could not be closed by TabPFNv2 even after extensive fine-tuning, as noted by its fine-tuning performance in Table 2 and confirmed with its authors. More importantly, this aligns with what users care about: achieving the best predictive performance on their datasets efficiently. Mitra fine-tuning is not only highly effective but also computationally efficient, running **~3.5× faster than CatBoost and faster than RealMLP as shown in Figure 14**, delivering state-of-the-art performance with minimal adaptation cost.
> 5. **Additional results on TabArena also show Mitra’s SOTA performance**: We have run additional comparison on a recent benchmark TabArena [5] and show that
>     1. **Mitra achieves Pareto efficiency** both in training time and inference time, with TabPFNv2 (HPO + ensemble) being significantly slower.
>     2. **Mitra is the strongest single model**, outperforming all methods even when they perform hyperparameter tuning for 200 iterations. Mitra is only outperformed once the hyperparameter configurations of TabPFNv2 are ensembled together. We are still working on constructing a search space for Mitra, and thus do not have a HPO and HPO + ensemble result at this time.
>     3. **Mitra performs favorably to all non-foundational methods**, including TabM and RealMLP.
>
> |Method|Avg. Rank|Elo|Winrate|Champ Delta|
> |:-|-:|-:|-:|-:|
> |TabPFNv2 (HPO+ensemble)|6.2|1745|0.89|0.05|
> |**Mitra (single)**|7.4|1699|0.86|0.07|
> |TabM (HPO+ensemble)|9.9|1620|0.8|0.1|
> |TabICL (single)|10.1|1617|0.8|0.07|
> |RealMLP (HPO+ensemble)|10.9|1597|0.78|0.09|
> |TabPFNv2 (HPO)|10.9|1596|0.78|0.08|
> |AutoGluon1.3 (4h)|12.6|1551|0.74|0.1|
> |TabPFNv2 (single)|12.7|1548|0.74|0.1|
> |LightGBM (HPO+ensemble)|13.7|1525|0.72|0.12|
> |TabM (HPO)|14.2|1512|0.71|0.11|
> |LightGBM (HPO)|16.2|1470|0.66|0.12|
> |CatBoost (HPO+ensemble)|16.5|1465|0.66|0.12|
> |CatBoost (HPO)|17.3|1444|0.64|0.12|
> |TabM (single)|17.8|1437|0.63|0.14|
> |CatBoost (single)|17.8|1434|0.63|0.14|
> |ModernNCA (HPO)|18|1428|0.62|0.12|
> |XGBoost (HPO+ensemble)|18.5|1420|0.61|0.13|
> |EBM (HPO+ensemble)|20|1390|0.58|0.16|
> |XGBoost (HPO)|20.1|1386|0.58|0.14|
> |RealMLP (HPO)|20.2|1383|0.57|0.13|
> |ModernNCA (HPO+ensemble)|20.4|1381|0.57|0.13|
> |ModernNCA (single)|20.8|1368|0.56|0.15|
> |TorchMLP (HPO+ensemble)|20.9|1370|0.56|0.14|
> |FastaiMLP (HPO+ensemble)|21.1|1366|0.55|0.16|
> |TabDPT (single)|22.7|1329|0.52|0.15|
> |EBM (HPO)|23.1|1323|0.51|0.17|
> |EBM (single)|23.8|1307|0.49|0.18|
> |RealMLP (single)|25.6|1270|0.45|0.16|
> |FastaiMLP (HPO)|25.6|1273|0.45|0.17|
> |ExtraTrees (HPO+ensemble)|26.1|1261|0.44|0.18|
> |TorchMLP (HPO)|27|1241|0.42|0.16|
> |XGBoost (single)|28.2|1214|0.39|0.17|
> |ExtraTrees (HPO)|28.6|1205|0.39|0.2|
> |RandomForest (HPO+ensemble)|30.5|1159|0.34|0.2|
> |LightGBM (single)|30.7|1159|0.34|0.18|
> |RandomForest (HPO)|33.1|1094|0.29|0.21|
> |TorchMLP (single)|33.4|1087|0.28|0.22|
> |FastaiMLP (single)|34.6|1056|0.25|0.23|
> |Linear (HPO+ensemble)|35.2|1032|0.24|0.29|
> |Linear (HPO)|36.2|1005|0.22|0.29|
> |RandomForest (single)|36.3|1000|0.21|0.26|
> |Linear (single)|36.8|985|0.21|0.31|
> |ExtraTrees (single)|37.7|952|0.19|0.28|
> |KNN (HPO+ensemble)|42.6|716|0.08|0.48|
> |KNN (HPO)|43.7|623|0.05|0.5|
> |KNN (single)|45.2|415|0.02|0.59|
>
> > Computational complexity and stability for prior analysis.
>
> 1. **Computational complexity**: Mitra’s pretraining cost is comparable to TabPFNv2, taking around 60 hours on 8 Nvidia A100 GPUs. Our prior analysis (Table 1) and ablation study (Table 6) are used for analyzing the prior importance in our mixture of priors in Mitra and demonstrating the robustness of our principled method, and they are not part of the actual pretraining process itself. Compared with prior works such as TabICL [1], we are the first to perform such rigorous study on prior importance.
> 2. **Stability**: Mitra and its ablations are stable across runs. For example, the average AUC differences is within 0.003 across different restarts.
>
> [1] TabICL: A Tabular Foundation Model for In-Context Learning on Large Data
>
> [2] TabFlex: Scaling Tabular Learning to Millions with Linear Attention
>
> [3] Fine-tuned In-Context Learning Transformers are Excellent Tabular Data Classifiers
>
> [4] Attic: A New Architecture for Tabular In-Context Learning Transformers
>
> [5] TabArena: TabArena: A Living Benchmark for Machine Learning on Tabular Data

---

> > ### Author Response · Authors · 2025-08-06
> >
> > We are writing to follow up on our rebuttal, where we have done our best to address your concerns with additional experiments and detailed clarifications. As the discussion period closes soon, we sincerely hope that you have a moment to review our response. We are eager to address any remaining questions you might have and would be happy to discuss further. Thank you again for your valuable feedback!

---

### Note · Authors · 2025-08-12

Dear AC and reviewers,

We sincerely thank you for the thoughtful feedback and discussions.

**3 out of 4 Reviewers** have all finalized with **acceptance** recommendations. To summarize our paper, we conducted the first principled analysis on what makes effective priors, and constructed a diverse mixture of priors that leads to our state-of-the-art tabular foundation model, Mitra. Our findings are **novel and insightful (Reviewer gtzp, Uza4, EUuF)**, supported by **comprehensive experiments and analysis (All four Reviewers)**, and our paper is **clear and well-written (All four Reviewers)**. Reviewers stated that “this is a paper of **interest to the community**, with **significant empirical evidence and ablations**” (Reviewer Uza4), “**satisfied** with the responses, and would be **happy to see the paper through**” (Reviewer fccq), “among the **most detailed investigations** so far in the domain regarding the mixture of priors”, “the proposed work is of **interest to the community**” (Reviewer EUuF).

Reviewer gtzp has not yet provided a response, their concerns are on ICL performance and the prior importance analysis, **which were similarly raised by the other three Reviewers who all expressed satisfaction with our clarifications as detailed above**. Furthermore, we have clarified that Mitra’s ICL performance relative to TabPFNv2 reflects our design choices rather than a limitation of the proposed prior mixture, supported by additional experiments validating our response. We also provided details on the computational considerations for prior importance analysis. We would greatly appreciate your consideration of these clarifications.

We are grateful for the time and effort all reviewers and the AC have put into the review process, and we would greatly appreciate your consideration of these points and your support.

Best regards,

The Authors

---

### Decision · Program_Chairs · 2025-09-17

**Decision:**

Accept (poster)

**Comment:**

Claims and findings: The paper studies what makes synthetic priors effective for tabular foundation models trained via in-context learning. It proposes a few criteria, such as a prior’s standalone performance on real data, diversity, and distinctiveness. These criteria are used to curate a mixture of priors. This mixture is used to train a tabular foundation models that perform quite well.

Strengths: It’s nice to have a systematic analysis of synthetic prior design for this tabular setting. The paper’s main problem is timely; tabular foundation models are increasingly popular. The results are generally pretty good, and the analysis is extensive.

Weaknesses: Probably the main weakness is that the basic formulation doesn’t always perform better than a strong baseline, and it becomes necessary to fine-tune for these gains.

Decision: Overall this is a nice combination of analysis, systematic  experimentation, and solid empirical results. It also sheds some light in an important area. Based on this, the paper is worth accepting.

Rebuttal discussion: The discussion was quite extensive. A bunch of points were clarified, which will improve the final version of the paper.